# Tight last-iterate convergence rates for no-regret learning in multi-player games

**Noah Golowich**
MIT CSAIL
nzg@mit.edu

**Sarath Pattathil**
MIT EECS
sarathp@mit.edu

**Constantinos Daskalakis**
MIT CSAIL
costis@csail.mit.edu

## Abstract

We study the question of obtaining last-iterate convergence rates for no-regret learning algorithms in multi-player games. We show that the optimistic gradient (OG) algorithm with a constant step-size, which is no-regret, achieves a last-iterate rate of $O(1/\sqrt{T})$ with respect to the gap function in smooth monotone games. This result addresses a question of Mertikopoulos & Zhou (2018), who asked whether extra-gradient approaches (such as OG) can be applied to achieve improved guarantees in the multi-agent learning setting. The proof of our upper bound uses a new technique centered around an adaptive choice of potential function at each iteration. We also show that the $O(1/\sqrt{T})$ rate is tight for all $p$-SCLI algorithms, which includes OG as a special case. As a byproduct of our lower bound analysis we additionally present a proof of a conjecture of Arjevani et al. (2015) which is more direct than previous approaches.

## 1 Introduction

In the setting of *multi-agent online learning* ([SS11, CBL06]), $K$ players interact with each other over time. At each time step $t$, each player $k \in \{1, \ldots, K\}$ chooses an *action* $\mathbf{z}_k^{(t)}$; $\mathbf{z}_k^{(t)}$ may represent, for instance, the bidding strategy of an advertiser at time $t$. Player $k$ then suffers a *loss* $\ell_t(\mathbf{z}_k^{(t)})$ that depends on both player $k$'s action $\mathbf{z}_k^{(t)}$ and the actions of all other players at time $t$ (which are absorbed into the loss function $\ell_t(\cdot)$). Finally, player $k$ receives some *feedback* informing them of how to improve their actions in future iterations. In this paper we study gradient-based feedback, meaning that the feedback is the vector $\mathbf{g}_k^{(t)} = \nabla_{\mathbf{z}_k} \ell_t(\mathbf{z}_k^{(t)})$.

A fundamental quantity used to measure the performance of an online learning algorithm is the *regret* of player $k$, which is the difference between the total loss of player $k$ over $T$ time steps and the loss of the best possible action in hindsight: formally, the regret at time $T$ is $\sum_{t=1}^T \ell_t(\mathbf{z}_k^{(t)}) - \min_{\mathbf{z}_k} \sum_{t=1}^T \ell_t(\mathbf{z}_k)$. An algorithm is said to be *no-regret* if its regret at time $T$ grows sub-linearly with $T$ for an adversarial choice of the loss functions $\ell_t$. If all agents playing a game follow no-regret learning algorithms to choose their actions, then it is well-known that the empirical frequency of their actions converges to a *coarse correlated equilibrium (CCE)* ([MV78, CBL06]). In turn, a substantial body of work (e.g., [CBL06, DP09, EDMN09, CD11, VZ13, KKDB15, BTHK15, MP17, MZ18, KBTB18]) has focused on establishing for which classes of games or learning algorithms this convergence to a CCE can be strengthened, such as to convergence to a *Nash equilibrium (NE)*.

However, the type of convergence guaranteed in these works generally either applies only to the time-average of the joint action profiles, or else requires the sequence of learning rates to converge to 0. Such guarantees leave substantial room for improvement: a statement about the average of the joint action profiles fails to capture the game dynamics over time ([MPP17]), and both types of guarantees use newly acquired information with decreasing weight, which, as remarked by [LZMJ20], is very

Table 1: Known last-iterate convergence rates for learning in smooth monotone games with perfect gradient feedback (i.e., *deterministic* algorithms). We specialize to the 2-player 0-sum case in presenting prior work, since some papers in the literature only consider this setting. Recall that a game $\mathcal{G}$ has a $\gamma$-*singular value lower bound* if for all $\mathbf{z}$, all singular values of $\partial F_{\mathcal{G}}(\mathbf{z})$ are $\geq \gamma$. $\ell, \Lambda$ are the Lipschitz constants of $F_{\mathcal{G}}, \partial F_{\mathcal{G}}$, respectively, and $c, C > 0$ are absolute constants where $c$ is sufficiently small and $C$ is sufficiently large. Upper bounds in the left-hand column are for the EG algorithm, and lower bounds are for a general form of 1-SCLI methods which include EG. Upper bounds in the right-hand column are for algorithms which are implementable as online no-regret learning algorithms (e.g., OG or online gradient descent), and lower bounds are shown for two classes of algorithms containing OG and online gradient descent, namely $p$-SCLI algorithms for general $p \geq 1$ (recall for OG, $p = 2$) as well as those satisfying a 2-step linear span assumption (see [IAGM19]). The reported upper and lower bounds are stated for the total gap function (Definition 3); leading constants and factors depending on distance between initialization and optimum are omitted.

| | Deterministic | |
| Game class | Extra gradient | Implementable as no-regret |
| --- | --- | --- |
| $\mu$-strongly monotone | *Upper:* $\ell\left(1 - \frac{c\mu}{\ell}\right)^T$ [MOP19b, EG] <br> *Lower:* $\mu\left(1 - \frac{C\mu}{\ell}\right)^T$ [AMLJG19, 1-SCLI] | *Upper:* $\ell\left(1 - \frac{c\mu}{\ell}\right)^T$ [MOP19b, OG] <br> *Lower:* $\mu\left(1 - \frac{C\mu}{\ell}\right)^T$ [IAGM19, 2-step lin. span] <br> *Lower:* $\mu\left(1 - \sqrt[p]{\frac{C\mu}{\ell}}\right)^T$ [ASSS15, IAGM19, $p$-SCLI] |
| Monotone, $\gamma$-sing. val. low. bnd. | *Upper:* $\ell\left(1 - \frac{c\gamma^2}{\ell^2}\right)^T$ [AMLJG19, EG] <br> *Lower:* $\gamma\left(1 - \frac{C\gamma^2}{\ell^2}\right)^T$ [AMLJG19, 1-SCLI] | *Upper:* $\ell\left(1 - \frac{c\gamma^2}{\ell^2}\right)^T$ [AMLJG19, OG] <br> *Lower:* $\gamma\left(1 - \frac{C\gamma}{\ell}\right)^T$ [IAGM19, 2-step lin. span] <br> *Lower:* $\gamma\left(1 - \sqrt[p]{\frac{C\gamma}{\ell}}\right)^T$ [ASSS15, IAGM19, $p$-SCLI] |
| $\lambda$-cocoercive | – | *Upper:* $\frac{1}{\lambda\sqrt{T}}$ [LZMJ20, Online grad. descent] |
| Monotone | *Upper:* $\frac{\ell+\Lambda}{\sqrt{T}}$ [GPDO20, EG] <br> *Lower:* $\frac{\ell}{\sqrt{T}}$ [GPDO20, 1-SCLI] | *Upper:* $\frac{\ell+\Lambda}{\sqrt{T}}$ (**Theorem 5**, OG) <br> *Lower:* $\frac{\ell}{\sqrt{T}}$ (**Theorem 7**, $p$-SCLI, lin. coeff. matrices) |

unnatural from an economic perspective.[1] Therefore, the following question is of particular interest ([MZ18, LZMJ20, MPP17, DISZ17]):

> *Can we establish last-iterate rates if all players act according to a no-regret learning algorithm with constant step size?*     ($\star$)

We measure the proximity of an action profile $\mathbf{z} = (\mathbf{z}_1, \ldots, \mathbf{z}_K)$ to equilibrium in terms of the *total gap function* at $\mathbf{z}$ (Definition 3): it is defined to be the sum over all players $k$ of the maximum decrease in cost player $k$ could achieve by deviating from its action $\mathbf{z}_k$. [LZMJ20] took initial steps toward addressing ($\star$), showing that if all agents follow the *online gradient descent* algorithm, then for all $\lambda$-*cocoercive games*, the action profiles $\mathbf{z}^{(t)} = (\mathbf{z}_1^{(t)}, \ldots, \mathbf{z}_K^{(t)})$ will converge to equilibrium in terms of the total gap function at a rate of $O(1/\sqrt{T})$. Moreover, linear last-iterate rates have been long known for smooth *strongly-monotone* games ([Tse95, GBV$^+$18, LS18, MOP19b, AMLJG19, ZMM$^+$20]), a sub-class of $\lambda$-cocoercive games. Unfortunately, even $\lambda$-cocoercive games exclude many important classes of games, such as bilinear games, which are the adaptation of matrix games to the unconstrained setting. Moreover, this shortcoming is not merely an artifact of the analysis of [LZMJ20]: it has been observed (e.g. [DISZ17, GBV$^+$18]) that in bilinear games, the players' actions in online gradient descent not only fail to converge, but diverge to infinity. Prior work on last-iterate convergence rates for these various subclasses of monotone games is summarized in Table 1 for the case of perfect gradient feedback; the setting for noisy feedback is summarized in Table 2 in Appendix A.4.

## 1.1 Our contributions

In this paper we answer ($\star$) in the affirmative for all *monotone games* (Definition 1) satisfying a mild smoothness condition, which includes smooth $\lambda$-cocoercive games and bilinear games. Many common and well-studied classes of games, such as zero-sum polymatrix games ([BF87, DP09, CCDP16]) and its generalization zero-sum socially-concave games ([EDMN09]) are monotone but are not in general $\lambda$-cocoercive. Hence our paper is the first to prove last-iterate convergence in the sense of ($\star$) for the unconstrained version of these games as well. In more detail, we establish the following:

- We show in Theorem 5 and Corollary 6 that the actions taken by learners following the *optimistic gradient (OG)* algorithm, which is no-regret, exhibit last-iterate convergence to a Nash equilibrium in smooth, monotone games at a rate of $O(1/\sqrt{T})$ in terms of the global gap function. The proof uses a new technique which we call *adaptive potential functions* (Section 3.1) which may be of independent interest.

- We show in Theorem 7 that the rate $O(1/\sqrt{T})$ cannot be improved for any algorithm belonging to the class of $p$-SCLI algorithms (Definition 5), which includes OG.

The OG algorithm is closely related to the *extra-gradient (EG)* algorithm ([Kor76, Nem04]),[2] which, at each time step $t$, assumes each player $k$ has an oracle $\mathcal{O}_k$ which provides them with an additional gradient at a slightly different action than the action $\mathbf{z}_k^{(t)}$ played at step $t$. Hence EG does not naturally fit into the standard setting of multi-agent learning. One could try to "force" EG into the setting of multi-agent learning by taking actions at odd-numbered time steps $t$ to simulate the oracle $\mathcal{O}_k$, and using the even-numbered time steps to simulate the actions $\mathbf{z}_k^{(t)}$ that EG actually takes. Although this algorithm exhibits last-iterate convergence at a rate of $O(1/\sqrt{T})$ in smooth monotone games when all players play according to it [GPDO20], it is straightforward to see that it is *not* a no-regret learning algorithm, i.e., for an adversarial loss function the regret can be linear in $T$ (see Proposition 10 in Appendix A.3).

Nevertheless, due to the success of EG at solving monotone variational inequalities, [MZ18] asked whether similar techniques to EG could be used to speed up last-iterate convergence to Nash equilibria. Our upper bound for OG answers this question in the affirmative: various papers ([CYL$^+$12, RS12, RS13, HIMM19]) have observed that OG may be viewed as an approximation of EG, in which the previous iteration's gradient is used to simulate the oracle $\mathcal{O}_k$. Moreover, our upper bound of $O(1/\sqrt{T})$ applies in many games for which the approach used in [MZ18], namely Nesterov's dual averaging ([Nes09]), either fails to converge (such as bilinear games) or only yields asymptotic rates with decreasing learning rate (such as smooth strictly monotone games). Proving last-iterate rates for OG has also been noted as an important open question in [HIMM19, Table 1]. At a technical level, the proof of our upper bound (Theorem 5) uses the proof technique in [GPDO20] for the last-iterate convergence of EG as a starting point. In particular, similar to [GPDO20], our proof proceeds by first noting that some iterate $\mathbf{z}^{(t^*)}$ of OG will have gradient gap $O(1/\sqrt{T})$ (see Definition 2; this is essentially a known result) and then showing that for all $t \geq t^*$ the gradient gap only increases by at most a constant factor. The latter step is the bulk of the proof, as was the case in [GPDO20]; however, since each iterate of OG depends on the previous two iterates and gradients, the proof for OG is significantly more involved than that for EG. We refer the reader to Section 3.1 and Appendix B for further details.

The proof of our lower bound for $p$-SCLI algorithms, Theorem 7, reduces to a question about the spectral radius of a family of polynomials. In the course of our analysis we prove a conjecture by [ASSS15] about such polynomials; though the validity of this conjecture is implied by each of several independent results in the literature (e.g., [AS16, Nev93]), our proof is more direct than previous ones.

Lastly, we mention that our focus in this paper is on the unconstrained setting, meaning that the players' losses are defined on all of Euclidean space. We leave the constrained setting, in which the players must project their actions onto a convex constraint set, to future work.

## 1.2 Related work

**Multi-agent learning in games.** In the constrained setting, many papers have studied conditions under which the action profile of no-regret learning algorithms, often variants of Follow-The-Regularized-Leader (FTRL), converges to equilibrium. However, these works all assume either a learning rate that decreases over time ([MZ18, ZMB$^+$17, ZMA$^+$18, ZMM$^+$17]), or else only apply to specific types of *potential games* ([KKDB15, KBTB18, PPP17, KPT09, CL16, BEDL06, PP14]), which significantly facilitates the analysis of last-iterate convergence.[3]

Such potential games are in general incomparable with monotone games, and do not even include finite-state two-player zero sum games (i.e., *matrix games*). In fact, [BP18] showed that the actions of players following FTRL in two-player zero-sum matrix games *diverge* from interior Nash equilibria. Many other works ([HMC03, MPP17, KLP11, DFP$^+$10, BCM12, PP16]) establish similar non-convergence results in both discrete and continuous time for various types of monotone games, including zero-sum polymatrix games. Such non-convergence includes chaotic behavior such as Poincaré recurrence, which showcases the insufficiency of on-average convergence (which holds in such settings) and so is additional motivation for the question ($\star$).

**Monotone variational inequalities & OG.** The problem of finding a Nash equilibrium of a monotone game is exactly that of finding a solution to a monotone variational inequality (VI). OG was originally introduced by [Pop80], who showed that its iterates converge to solutions of monotone VIs, without proving explicit rates.[4] It is also well-known that the *averaged* iterate of OG converges to the solution of a monotone VI at a rate of $O(1/T)$ ([HIMM19, MOP19a, RS13]), which is known to be optimal ([Nem04, OX19, ASM$^+$20]). Recently it has been shown ([DP18, LNPW20]) that a modification of OG known as optimistic multiplicative-weights update exhibits last-iterate convergence to Nash equilibria in two-player zero-sum monotone games, but as with the unconstrained case ([MOP19a]) non-asymptotic rates are unknown. To the best of our knowledge, the only work proving last-iterate convergence rates for general smooth monotone VIs was [GPDO20], which only treated the EG algorithm, which is not no-regret. There is a vast literature on solving VIs, and we refer the reader to [FP03] for further references.

## 2 Preliminaries

Throughout this paper we use the following notational conventions. For a vector $\mathbf{v} \in \mathbb{R}^n$, let $\|\mathbf{v}\|$ denote the Euclidean norm of $\mathbf{v}$. For $\mathbf{v} \in \mathbb{R}^n$, set $\mathcal{B}(\mathbf{v}, R) := \{\mathbf{z} \in \mathbb{R}^n : \|\mathbf{v} - \mathbf{z}\| \leq R\}$; when we wish to make the dimension explicit we write $\mathcal{B}_{\mathbb{R}^n}(\mathbf{v}, R)$. For a matrix $\mathbf{A} \in \mathbb{R}^{n \times n}$ let $\|\mathbf{A}\|_\sigma$ denote the spectral norm of $\mathbf{A}$.

We let the set of $K$ players be denoted by $\mathcal{K} := \{1, 2, \dots K\}$. Each player $k$'s actions $\mathbf{z}_k$ belong to their *action set*, denoted $\mathcal{Z}_k$, where $\mathcal{Z}_k \subseteq \mathbb{R}^{n_k}$ is a convex subset of Euclidean space. Let $\mathcal{Z} = \prod_{k=1}^K \mathcal{Z}_k \subseteq \mathbb{R}^n$, where $n = n_1 + \cdots + n_K$. In this paper we study the setting where the action sets are unconstrained (as in [LZMJ20]), meaning that $\mathcal{Z}_k = \mathbb{R}^{n_k}$, and $\mathcal{Z} = \mathbb{R}^n$, where $n = n_1 + \cdots + n_K$. The *action profile* is the vector $\mathbf{z} := (\mathbf{z}_1, \dots, \mathbf{z}_K) \in \mathcal{Z}$. For any player $k \in \mathcal{K}$, let $\mathbf{z}_{-k} \in \prod_{k' \neq k} \mathcal{Z}_{k'}$ be the vector of actions of all the other players. Each player $k \in \mathcal{K}$ wishes to minimize its *cost function* $f_k : \mathcal{Z} \to \mathbb{R}$, which is assumed to be twice continuously differentiable. The tuple $\mathcal{G} := (\mathcal{K}, (\mathcal{Z}_k)_{k=1}^K, (f_k)_{k=1}^K)$ is known as a *continuous game*.

At each time step $t$, each player $k$ plays an action $\mathbf{z}_k^{(t)}$; we assume the feedback to player $k$ is given in the form of the gradient $\nabla_{\mathbf{z}_k} f_k(\mathbf{z}_k^{(t)}, \mathbf{z}_{-k}^{(t)})$ of their cost function with respect to their action $\mathbf{z}_k^{(t)}$, given the actions $\mathbf{z}_{-k}^{(t)}$ of the other players at time $t$. We denote the concatenation of these gradients by $F_{\mathcal{G}}(\mathbf{z}) := (\nabla_{\mathbf{z}_1} f_1(\mathbf{z}), \dots, \nabla_{\mathbf{z}_K} f_K(\mathbf{z})) \in \mathbb{R}^n$. When the game $\mathcal{G}$ is clear, we will sometimes drop the subscript and write $F : \mathcal{Z} \to \mathbb{R}^n$.

**Equilibria & monotone games.** A *Nash equilibrium* in the game $\mathcal{G}$ is an action profile $\mathbf{z}^* \in \mathcal{Z}$ so that for each player $k$, it holds that $f_k(\mathbf{z}_k^*, \mathbf{z}_{-k}^*) \leq f_k(\mathbf{z}_k', \mathbf{z}_{-k}^*)$ for any $\mathbf{z}_k' \in \mathcal{Z}_k$. Throughout this paper we study *monotone* games:

**Definition 1** (Monotonicity; [Ros65])**.** The game $\mathcal{G} = (\mathcal{K}, (\mathcal{Z}_k)_{k=1}^K, (f_k)_{k=1}^K)$ is *monotone* if for all $\mathbf{z}, \mathbf{z}' \in \mathcal{Z}$, it holds that $\langle F_\mathcal{G}(\mathbf{z}') - F_\mathcal{G}(\mathbf{z}), \mathbf{z}' - \mathbf{z} \rangle \geq 0$. In such a case, we say also that $F_\mathcal{G}$ is a monotone operator.

The following classical result characterizes the Nash equilibria in monotone games:

**Proposition 1** ([FP03])**.** *In the unconstrained setting, if the game $\mathcal{G}$ is monotone, any Nash equilibrium $\mathbf{z}^*$ satisfies $F_\mathcal{G}(\mathbf{z}^*) = \mathbf{0}$. Conversely, if $F_\mathcal{G}(\mathbf{z}) = \mathbf{0}$, then $\mathbf{z}$ is a Nash equilibrium.*

In accordance with Proposition 1, one measure of the proximity to equilibrium of some $\mathbf{z} \in \mathcal{Z}$ is the norm of $F_\mathcal{G}(\mathbf{z})$:

**Definition 2** (Gradient gap function)**.** Given a monotone game $\mathcal{G}$ with its associated operator $F_\mathcal{G}$, the *gradient gap function* evaluated at $\mathbf{z}$ is defined to be $\|F_\mathcal{G}(\mathbf{z})\|$.

It is also common ([MOP19a, Nem04]) to measure the distance from equilibrium of some $\mathbf{z} \in \mathcal{Z}$ by adding the maximum decrease in cost that each player could achieve by deviating from their current action $\mathbf{z}_k$:

**Definition 3** (Total gap function)**.** Given a monotone game $\mathcal{G} = (\mathcal{K}, (\mathcal{Z}_k)_{k=1}^K, (f_k)_{k=1}^K)$, compact subsets $\mathcal{Z}_k' \subseteq \mathcal{Z}_k$ for each $k \in \mathcal{K}$, and a point $\mathbf{z} \in \mathcal{Z}$, define the *total gap function* at $\mathbf{z}$ with respect to the set $\mathcal{Z}' := \prod_{k=1}^K \mathcal{Z}_k'$ by $\mathrm{TGap}_\mathcal{G}^{\mathcal{Z}'}(\mathbf{z}) := \sum_{k=1}^K \left( f_k(\mathbf{z}) - \min_{\mathbf{z}_k' \in \mathcal{Z}_k'} f_k(\mathbf{z}_k', \mathbf{z}_{-k}) \right)$. At times we will slightly abuse notation, and for $F := F_\mathcal{G}$, write $\mathrm{TGap}_F^{\mathcal{Z}'}$ in place of $\mathrm{TGap}_\mathcal{G}^{\mathcal{Z}'}$.

As discussed in [GPDO20], it is in general impossible to obtain meaningful guarantees on the total gap function by allowing each player to deviate to an action in their entire space $\mathcal{Z}_k$, which necessitates defining the total gap function in Definition 3 with respect to the compact subsets $\mathcal{Z}_k'$. We discuss in Remark 4 how, in our setting, it is without loss of generality to shrink $\mathcal{Z}_k$ so that $\mathcal{Z}_k = \mathcal{Z}_k'$ for each $k$. Proposition 2 below shows that in monotone games, the gradient gap function upper bounds the total gap function:

**Proposition 2.** *Suppose $\mathcal{G} = (\mathcal{K}, (\mathcal{Z}_k)_{k=1}^K, (f_k)_{k=1}^K)$ is a monotone game, and compact subsets $\mathcal{Z}_k' \subset \mathcal{Z}_k$ are given, where the diameter of each $\mathcal{Z}_k'$ is upper bounded by $D > 0$. Then*

$$\mathrm{TGap}_\mathcal{G}^{\mathcal{Z}'}(\mathbf{z}) \leq D\sqrt{K} \cdot \|F_\mathcal{G}(\mathbf{z})\|.$$

For completeness, a proof of Proposition 2 is presented in Appendix A.

**Special case: convex-concave min-max optimization.** Since in a two-player zero-sum game $\mathcal{G} = (\{1, 2\}, (\mathcal{Z}_1, \mathcal{Z}_2), (f_1, f_2))$ we must have $f_1 = -f_2$, it is straightforward to show that $f_1(\mathbf{z}_1, \mathbf{z}_2)$ is convex in $\mathbf{z}_1$ and concave in $\mathbf{z}_2$. Moreover, it is immediate that Nash equilibria of the game $\mathcal{G}$ correspond to saddle points of $f_1$; thus a special case of our setting is that of finding saddle points of convex-concave functions ([FP03]). Such saddle point problems have received much attention recently since they can be viewed as a simplified model of generative adversarial networks (e.g., [GBV$^+$18, DISZ17, CGFLJ19, GHP$^+$18, YSX$^+$17]).

**Optimistic gradient (OG) algorithm.** In the *optimistic gradient (OG)* algorithm, each player $k$ performs the following update:

$$\mathbf{z}_k^{(t+1)} := \mathbf{z}_k^{(t)} - 2\eta_t \mathbf{g}_k^{(t)} + \eta_t \mathbf{g}_k^{(t-1)}, \tag{OG}$$

where $\mathbf{g}_k^{(t)} = \nabla_{\mathbf{z}_k} f_k(\mathbf{z}_k^{(t)}, \mathbf{z}_{-k}^{(t)})$ for $t \geq 0$. The following essentially optimal regret bound is well-known for the OG algorithm, when the actions of the other players $\mathbf{z}_{-k}^{(t)}$ (often referred to as the *environment*'s actions) are adversarial:

**Proposition 3.** *Assume that for all $\mathbf{z}_{-k}$ the function $\mathbf{z}_k \mapsto f_k(\mathbf{z}_k, \mathbf{z}_{-k})$ is convex. Then the regret of OG with learning rate $\eta_t = O(D/L\sqrt{t})$ is $O(DL\sqrt{T})$, where $L = \max_t \|\mathbf{g}_k^{(t)}\|$ and $D = \max\{\|\mathbf{z}_k^*\|, \max_t \|\mathbf{z}_k^{(t)}\|\}$.*

In Proposition 3, $\mathbf{z}_k^*$ is defined by $\mathbf{z}_k^* \in \arg\min_{\mathbf{z}_k \in \mathcal{Z}_k} \sum_{t'=0}^{t} f_k(\mathbf{z}_k, \mathbf{z}_{-k}^{(t')})$. The assumption in the proposition that $\|\mathbf{z}_k^{(t)}\| \leq D$ may be satisfied in the unconstrained setting by projecting the iterates onto the region $\mathcal{B}(0, D) \subset \mathbb{R}^{n_k}$, for some $D \geq \|\mathbf{z}_k^*\|$, without changing the regret bound. The implications of this modification to (OG) are discussed further in Remark 4.

## 3 Last-iterate rates for OG via adaptive potential functions

In this section we show that in the unconstrained setting (namely, that where $\mathcal{Z}_k = \mathbb{R}^{n_k}$ for all $k \in \mathcal{K}$), when all players act according to OG, their iterates exhibit last-iterate convergence to a Nash equilibrium. Our convergence result holds for games $\mathcal{G}$ for which the operator $F_{\mathcal{G}}$ satisfies the following smoothness assumption:

**Assumption 4** (Smoothness). *For a monotone operator $F : \mathcal{Z} \to \mathbb{R}^n$, assume that the following first and second-order Lipschitzness conditions hold, for some $\ell, \Lambda > 0$:*

$$\forall \mathbf{z}, \mathbf{z}' \in \mathcal{Z}, \qquad \|F(\mathbf{z}) - F(\mathbf{z}')\| \leq \ell \cdot \|\mathbf{z} - \mathbf{z}'\| \tag{1}$$

$$\forall \mathbf{z}, \mathbf{z}' \in \mathcal{Z}, \qquad \|\partial F(\mathbf{z}) - \partial F(\mathbf{z}')\|_\sigma \leq \Lambda \cdot \|\mathbf{z} - \mathbf{z}'\|. \tag{2}$$

*Here $\partial F : \mathcal{Z} \to \mathbb{R}^{n \times n}$ denotes the Jacobian of $F$.*

Condition (1) is entirely standard in the setting of solving monotone variational inequalities ([Nem04]); condition (2) is also very mild, being made for essentially all second-order methods (e.g., [ALW19, Nes06]).

By the definition of $F_{\mathcal{G}}(\cdot)$, when all players in a game $\mathcal{G}$ act according to (OG) with constant step size $\eta$, then the action profile $\mathbf{z}^{(t)}$ takes the form

$$\mathbf{z}^{(-1)}, \mathbf{z}^{(0)} \in \mathbb{R}^n, \qquad \mathbf{z}^{(t+1)} = \mathbf{z}^{(t)} - 2\eta F_{\mathcal{G}}(\mathbf{z}^{(t)}) + \eta F_{\mathcal{G}}(\mathbf{z}^{(t-1)}) \ \forall t \geq 0. \tag{3}$$

The main theorem of this section, Theorem 5, shows that under the OG updates (3), the iterates converge at a rate of $O(1/\sqrt{T})$ to a Nash equilibrium with respect to the gradient gap function:

**Theorem 5** (Last-iterate convergence of OG). *Suppose $\mathcal{G}$ is a monotone game so that $F_{\mathcal{G}}$ satisfies Assumption 4. For some $\mathbf{z}^{(-1)}, \mathbf{z}^{(0)} \in \mathbb{R}^n$, suppose there is $\mathbf{z}^* \in \mathbb{R}^n$ so that $F_{\mathcal{G}}(\mathbf{z}^*) = 0$ and $\|\mathbf{z}^* - \mathbf{z}^{(-1)}\| \leq D, \|\mathbf{z}^* - \mathbf{z}^{(0)}\| \leq D$. Then the iterates $\mathbf{z}^{(T)}$ of OG (3) for any $\eta \leq \min\left\{\frac{1}{150\ell}, \frac{1}{1711 D \Lambda}\right\}$ satisfy:*

$$\|F_{\mathcal{G}}(\mathbf{z}^{(T)})\| \leq \frac{60D}{\eta\sqrt{T}} \tag{4}$$

By Proposition 2, we immediately get a bound on the total gap function at each time $T$:

**Corollary 6** (Total gap function for last iterate of OG). *In the setting of Theorem 5, let $\mathcal{Z}_k' := \mathcal{B}(\mathbf{z}_k^{(0)}, 3D)$ for each $k \in \mathcal{K}$. Then, with $\mathcal{Z}' = \prod_{k \in \mathcal{K}} \mathcal{Z}_k'$,*

$$\text{TGap}_{\mathcal{G}}^{\mathcal{Z}'}(\mathbf{z}^{(T)}) \leq \frac{180KD^2}{\eta\sqrt{T}}. \tag{5}$$

We made no attempt to optimize the consants in Theorem 5 and Corollary 6, and they can almost certainly be improved.

**Remark 4** (Bounded iterates). Recall from the discussion following Proposition 3 that it is necessary to project the iterates of OG onto a compact ball to achieve the no-regret property. As our guiding question ($\star$) asks for last-iterate rates achieved by a no-regret algorithm, we should ensure that such projections are compatible with the guarantees in Theorem 5 and Corollary 6. For this we note that [MOP19a, Lemma 4(b)] showed that for the dynamics (3) without constraints, for all $t \geq 0$, $\|\mathbf{z}^{(t)} - \mathbf{z}^*\| \leq 2\|\mathbf{z}^{(0)} - \mathbf{z}^*\|$. Therefore, as long as we make the very mild assumption of a known a priori upper bound $\|\mathbf{z}^*\| \leq D/2$ (as well as $\|\mathbf{z}_k^{(-1)}\| \leq D/2, \|\mathbf{z}_k^{(0)}\| \leq D/2$), if all players act according to (3), then the updates (3) remain unchanged if we project onto the constraint sets $\mathcal{Z}_k := \mathcal{B}(\mathbf{0}, 3D)$ at each time step $t$. This observation also serves as motivation for the compact sets $\mathcal{Z}_k'$ used in Corollary 6: the natural choice for $\mathcal{Z}_k'$ is $\mathcal{Z}_k$ itself, and by restricting $\mathcal{Z}_k$ to be compact, this choice becomes possible.

### 3.1 Proof overview: adaptive potential functions

In this section we sketch the idea of the proof of Theorem 5; full details of the proof may be found in Appendix B. First we note that it follows easily from results of [HIMM19] that OG exhibits *best-iterate* convergence, i.e., in the setting of Theorem 5 we have, for each $T > 0$, $\min_{1 \le t \le T} \|F_{\mathcal{G}}(\mathbf{z}^{(t)})\| \le O(1/\sqrt{T})$.[5] The main contribution of our proof is then to show the following: if we choose $t^*$ so that $\|F_{\mathcal{G}}(\mathbf{z}^{(t^*)})\| \le O(1/\sqrt{T})$, then for all $t' \ge t^*$, we have $\|F_{\mathcal{G}}(\mathbf{z}^{(t')})\| \le O(1) \cdot \|F_{\mathcal{G}}(\mathbf{z}^{(t^*)})\|$. This was the same general approach taken in [GPDO20] to prove that the extragradient (EG) algorithm has last-iterate convergence. In particular, they showed the stronger statement that $\|F_{\mathcal{G}}(\mathbf{z}^{(t)})\|$ may be used as an approximate potential function in the sense that it only increases by a small amount each step:

$$\|F_{\mathcal{G}}(\mathbf{z}^{(t'+1)})\| \underbrace{\le}_{t' \ge 0} (1 + \|F(\mathbf{z}^{(t')})\|^2) \cdot \|F_{\mathcal{G}}(\mathbf{z}^{(t')})\| \underbrace{\le}_{t' \ge t^*} (1 + O(1/T)) \cdot \|F_{\mathcal{G}}(\mathbf{z}^{(t')})\|. \quad (6)$$

However, their approach relies crucially on the fact that for the EG algorithm, $\mathbf{z}^{(t+1)}$ depends only on $\mathbf{z}^{(t)}$. For the OG algorithm, it is possible that (6) fails to hold, even when $F_{\mathcal{G}}(\mathbf{z}^{(t)})$ is replaced by the more natural choice of $(F_{\mathcal{G}}(\mathbf{z}^{(t)}), F_{\mathcal{G}}(\mathbf{z}^{(t-1)}))$.[6]

Instead of using $\|F_{\mathcal{G}}(\mathbf{z}^{(t)})\|$ as a potential function in the sense of (6), we propose instead to track the behavior of $\|\tilde{F}^{(t)}\|$, where

$$\tilde{F}^{(t)} := F_{\mathcal{G}}(\mathbf{z}^{(t)} + \eta F_{\mathcal{G}}(\mathbf{z}^{(t-1)})) + \mathbf{C}^{(t-1)} \cdot F_{\mathcal{G}}(\mathbf{z}^{(t-1)}) \in \mathbb{R}^n, \quad (7)$$

and the matrices $\mathbf{C}^{(t-1)} \in \mathbb{R}^{n \times n}$ are defined recursively *backwards*, i.e., $\mathbf{C}^{(t-1)}$ depends directly on $\mathbf{C}^{(t)}$, which depends directly on $\mathbf{C}^{(t+1)}$, and so on. For an appropriate choice of the matrices $\mathbf{C}^{(t)}$, we show that $\tilde{F}^{(t+1)} = (I - \eta \mathbf{A}^{(t)} + \mathbf{C}^{(t)}) \cdot \tilde{F}^{(t)}$, for some matrix $\mathbf{A}^{(t)} \approx \partial F_{\mathcal{G}}(\mathbf{z}^{(t)})$. We then show that for $t \ge t^*$, it holds that $\|I - \eta \mathbf{A}^{(t)} + \mathbf{C}^{(t)}\|_\sigma \le 1 + O(1/T)$, from which it follows that $\|\tilde{F}^{(t+1)}\| \le (1 + O(1/T)) \cdot \|\tilde{F}^{(t)}\|$. This modification of (6) is enough to show the desired upper bound of $\|F_{\mathcal{G}}(\mathbf{z}^{(T)})\| \le O(1/\sqrt{T})$.

To motivate the choice of $\tilde{F}^{(t)}$ in (7) it is helpful to consider the simple case where $F(\mathbf{z}) = \mathbf{A}\mathbf{z}$ for some $\mathbf{A} \in \mathbb{R}^{n \times n}$, which was studied by [LS18]. Simple algebraic manipulations using (3) (detailed in Appendix B) show that, for the matrix $\mathbf{C} := \frac{(I+(2\eta\mathbf{A})^2)^{1/2}-I}{2}$, we have $\tilde{F}^{(t+1)} = (I - \eta\mathbf{A} + \mathbf{C})\tilde{F}^{(t)}$ for all $t$. It may be verified that we indeed have $\mathbf{A}^{(t)} = \mathbf{A}$ and $\mathbf{C}^{(t)} = \mathbf{C}$ for all $t$ in this case, and thus (7) may be viewed as a generalization of these calculations to the nonlinear case.

**Adaptive potential functions.** In general, a *potential function* $\Phi(F_{\mathcal{G}}, \mathbf{z})$ depends on the problem instance, here taken to be $F_{\mathcal{G}}$, and an element $\mathbf{z}$ representing the current state of the algorithm. Many convergence analyses from optimization (e.g., [BG17, WRJ18], and references therein) have as a crucial element in their proofs a statement of the form $\Phi(F_{\mathcal{G}}, \mathbf{z}^{(t+1)}) \lesssim \Phi(F_{\mathcal{G}}, \mathbf{z}^{(t)})$. For example, for the iterates $\mathbf{z}^{(t)}$ of the EG algorithm, [GPDO20] (see (6)) used the potential function $\Phi(F_{\mathcal{G}}, \mathbf{z}^{(t)}) := \|F_{\mathcal{G}}(\mathbf{z}^{(t)})\|$.

Our approach of controlling the the norm of the vectors $\tilde{F}^{(t)}$ defined in (7) can also be viewed as an instation of the potential function approach: since each iterate of OG depends on the previous two iterates, the state is now given by $\mathbf{v}^{(t)} := (\mathbf{z}^{(t-1)}, \mathbf{z}^{(t)})$. The potential function is given by $\Phi_{\mathrm{OG}}(F_{\mathcal{G}}, \mathbf{v}^{(t)}) := \|\tilde{F}^{(t)}\|$, where $\tilde{F}^\top$ is defined in (7) and indeed only depends on $\mathbf{v}^{(t)}$ once $F_{\mathcal{G}}$ is fixed since $\mathbf{v}^{(t)}$ determines $\mathbf{z}^{(t')}$ for all $t' \ge t$ (as OG is deterministic), which in turn determine $\mathbf{C}^{(t-1)}$. However, the potential function $\Phi_{\mathrm{OG}}$ is quite unlike most other choices of potential functions in optimization (e.g., [BG17]) in the sense that it depends *globally* on $F_{\mathcal{G}}$: For any $t' > t$, a local change in $F_{\mathcal{G}}$ in the neighborhood of $\mathbf{v}^{(t')}$ may cause a change in $\Phi_{\mathrm{OG}}(F_{\mathcal{G}}, \mathbf{v}^{(t)})$, *even if $\|\mathbf{v}^{(t)} - \mathbf{v}^{(t')}\|$ is arbitrarily large*. Because $\Phi_{\mathrm{OG}}(F_{\mathcal{G}}, \mathbf{v}^{(t)})$ adapts to the behavior of $F_{\mathcal{G}}$ at iterates later on in the optimization sequence, we call it an *adaptive potential function*. We are not aware of any prior works

using such adaptive potential functions to prove last-iterate convergence results, and we believe this technique may find additional applications.

# 4  Lower bound for convergence of $p$-SCLIs

The main result of this section is Theorem 7, stating that the bounds on last-iterate convergence in Theorem 5 and Corollary 6 are tight when we require the iterates $\mathbf{z}^{(T)}$ to be produced by an optimization algorithm satisfying a particular formal definition of "last-iterate convergence". Notice that that we cannot hope to prove that they are tight for *all* first-order algorithms, since the averaged iterates $\bar{\mathbf{z}}^{(T)} := \frac{1}{T} \sum_{t=1}^{T} \mathbf{z}^{(t)}$ of OG satisfy $\mathrm{TGap}_{\mathcal{G}}^{\mathcal{Z}'}(\bar{\mathbf{z}}^{(T)}) \leq O\left(\frac{D^2}{\eta T}\right)$ [MOP19a, Theorem 2]. Similar to [GPDO20], we use *$p$-stationary canonical linear iterative methods ($p$-SCLIs)* to formalize the notion of "last-iterate convergence". [GPDO20] only considered the special case $p = 1$ to establish a similar lower bound to Theorem 7 for a family of last-iterate algorithms including the extragradient algorithm. The case $p > 1$ leads to new difficulties in our proof since even for $p = 2$ we must rule out algorithms such as Nesterov's accelerated gradient descent ([Nes75]) and Pólya's heavy-ball method ([Pol87]), a situation that did not arise for $p = 1$.

**Definition 5** ($p$-SCLIs [ASSS15, ASM+20]). An algorithm $\mathcal{A}$ is a *first-order $p$-stationary canonical linear iterative algorithm ($p$-SCLI)* if, given a monotone operator $F$, and an arbitrary set of $p$ initialization points $\mathbf{z}^{(0)}, \mathbf{z}^{(-1)}, \ldots, \mathbf{z}^{(-p+1)} \in \mathbb{R}^n$, it generates iterates $\mathbf{z}^{(t)}$, $t \geq 1$, for which

$$\mathbf{z}^{(t)} = \sum_{j=0}^{p-1} \alpha_j \cdot F(\mathbf{z}^{(t-p+j)}) + \beta_j \cdot \mathbf{z}^{(t-p+j)}, \tag{8}$$

for $t = 1, 2, \ldots$, where $\alpha_j, \beta_j \in \mathbb{R}$ are any scalars.[7]

From (3) it is evident that OG with constant step size $\eta$ is a 2-SCLI with $\beta_1 = 1, \beta_0 = 0, \alpha_1 = -2\eta, \alpha_0 = \eta$. Many standard algorithms for convex function minimization, including gradient descent, Nesterov's accelerated gradient descent (AGD), and Pólya's Heavy Ball method, are of the form (8) as well. We additionally remark that several variants of SCLIs (and their non-stationary counterpart, CLIs) have been considered in recent papers proving lower bounds for min-max optimization ([AMLJG19, IAGM19, ASM+20]).

For simplicity, we restrict our attention to monotone operators $F$ arising as $F = F_{\mathcal{G}} : \mathbb{R}^n \to \mathbb{R}^n$ for a two-player zero-sum game $\mathcal{G}$ (i.e., the setting of min-max optimization). For simplicity suppose that $n$ is even and for $\mathbf{z} \in \mathbb{R}^n$ write $\mathbf{z} = (\mathbf{x}, \mathbf{y})$ where $\mathbf{x}, \mathbf{y} \in \mathbb{R}^{n/2}$. Define $\mathcal{F}_{n,\ell,D}^{\mathrm{bil}}$ to be the set of $\ell$-Lipschitz operators $F : \mathbb{R}^n \to \mathbb{R}^n$ of the form $F(\mathbf{x}, \mathbf{y}) = (\nabla_{\mathbf{x}} f(\mathbf{x}, \mathbf{y}), -\nabla_{\mathbf{y}} f(\mathbf{x}, \mathbf{y}))^\top$ for some bilinear function $f : \mathbb{R}^{n/2} \times \mathbb{R}^{n/2} \to \mathbb{R}$, with a unique equilibrium point $\mathbf{z}^* = (\mathbf{x}^*, \mathbf{y}^*)$, which satisfies $\mathbf{z}^* \in \mathcal{D}_D := \mathcal{B}_{\mathbb{R}^{n/2}}(\mathbf{0}, D) \times \mathcal{B}_{\mathbb{R}^{n/2}}(\mathbf{0}, D)$. The following Theorem 7 uses functions in $\mathcal{F}_{n,\ell,D}^{\mathrm{bil}}$ as "hard instances" to show that the $O(1/\sqrt{T})$ rate of Corollary 5 cannot be improved by more than an *algorithm-dependent* constant factor.

**Theorem 7** (Algorithm-dependent lower bound for $p$-SCLIs). *Fix $\ell, D > 0$, let $\mathcal{A}$ be a $p$-SCLI, and let $\mathbf{z}^{(t)}$ denote the $t$th iterate of $\mathcal{A}$. Then there are constants $c_{\mathcal{A}}, T_{\mathcal{A}} > 0$ so that the following holds: For all $T \geq T_{\mathcal{A}}$, there is some $F \in \mathcal{F}_{n,\ell,D}^{\mathrm{bil}}$ so that for some initialization $\mathbf{z}^{(0)}, \ldots, \mathbf{z}^{(-p+1)} \in \mathcal{D}_D$ and $T' \in \{T, T+1, \ldots, T+p-1\}$, it holds that $\mathrm{TGap}_F^{\mathcal{D}_{2D}}(\mathbf{z}^{(T')}) \geq \frac{c_{\mathcal{A}} \ell D^2}{\sqrt{T}}$.*

We remark that the order of quantifiers in Theorem 7 is important: if instead we first fix a monotone operator $F \in \mathcal{F}_{n,\ell,D}^{\mathrm{bil}}$ corresponding to some bilinear function $f(\mathbf{x}, \mathbf{y}) = \mathbf{x}^\top \mathbf{M} \mathbf{y}$, then as shown in [LS18, Theorem 3], the iterates $\mathbf{z}^{(T)} = (\mathbf{x}^{(T)}, \mathbf{y}^{(T)})$ of the OG algorithm will converge at a rate of $e^{-O\left(\frac{\sigma_{\min}(\mathbf{M})^2}{\sigma_{\max}(\mathbf{M})^2} \cdot T\right)}$, which eventually becomes smaller than the sublinear rate of $1/\sqrt{T}$.[8] Such "instance-specific" bounds are complementary to the minimax perspective taken in this paper.

We briefly discuss the proof of Theorem 7; the full proof is deferred to Appendix C. As in prior work proving lower bounds for $p$-SCLIs ([ASSS15, IAGM19]), we reduce the problem of proving a lower bound on $\text{TGap}_{\mathcal{G}}^{\mathcal{D}_D}(\mathbf{z}^{(t)})$ to the problem of proving a lower bound on the supremum of the spectral norms of a family of polynomials (which depends on $\mathcal{A}$). Recall that for a polynomial $p(z)$, its *spectral norm* $\rho(p(z))$ is the maximum norm of any root. We show:

**Proposition 8.** *Suppose $q(z)$ is a degree-$p$ monic real polynomial such that $q(1) = 0$, $r(z)$ is a polynomial of degree $p-1$, and $\ell > 0$. Then there is a constant $C_0 > 0$, depending only on $q(z), r(z)$ and $\ell$, and some $\mu_0 \in (0, \ell)$, so that for any $\mu \in (0, \mu_0)$,*

$$\sup_{\nu \in [\mu, \ell]} \rho(q(z) - \nu \cdot r(z)) \geq 1 - C_0 \cdot \frac{\mu}{\ell}.$$

The proof of Proposition 8 uses elementary tools from complex analysis. The fact that the constant $C_0$ in Proposition 8 depends on $q(z), r(z)$ leads to the fact that the constants $c_{\mathcal{A}}, T_{\mathcal{A}}$ in Theorem 7 depend on $\mathcal{A}$. Moreover, we remark that this dependence cannot be improved from Proposition 8, so removing it from Theorem 7 will require new techniques:

**Proposition 9** (Tightness of Proposition 8). *For any constant $C_0 > 0$ and $\mu_0 \in (0, \ell)$, there is some $\mu \in (0, \mu_0)$ and polynomials $q(z), r(z)$ so that $\sup_{\nu \in [\mu, \ell]} \rho(q(z) - \nu \cdot r(z)) < 1 - C_0 \cdot \mu$. Moreover, the choice of the polynomials is given by*

$$q(z) = \ell(z - \alpha)(z - 1), \qquad r(z) = -(1 + \alpha)z + \alpha \qquad \textit{for} \qquad \alpha := \frac{\sqrt{\ell} - \sqrt{\mu}}{\sqrt{\ell} + \sqrt{\mu}}. \qquad (9)$$

The choice of polynomials $q(z), r(z)$ in (9) are exactly the polynomials that arise in the $p$-SCLI analysis of Nesterov's AGD [ASSS15]; as we discuss further in Appendix C, Proposition 8 is tight, then, even for $p = 2$, because acceleration is possible with a 2-SCLI. As byproducts of our lower bound analysis, we additionally obtain the following:

- Using Proposition 8, we show that any $p$-SCLI algorithm must have a rate of at least $\Omega_{\mathcal{A}}(1/T)$ for smooth convex function minimization (again, with an algorithm-dependent constant).[9] This is slower than the $O(1/T^2)$ error achievable with Nesterov's AGD with a time-varying learning rate.

- We give a direct proof of the following statement, which was conjectured by [ASSS15]: for polynomials $q, r$ in the setting of Proposition 8, for any $0 < \mu < \ell$, there exists $\nu \in [\mu, \ell]$ so that $\rho(q(z) - \nu \cdot r(z)) \geq \frac{\sqrt{\ell/\mu} - 1}{\sqrt{\ell/\mu} + 1}$. Using this statement, for the setting of Theorem 7, we give a proof of an *algorithm-independent* lower bound $\text{TGap}_F^{\mathcal{D}_D}(\mathbf{z}^{(t)}) \geq \Omega(\ell D^2/T)$. Though the algorithm-independent lower bound of $\Omega(\ell D^2/T)$ has already been established in the literature, even for non-stationary CLIs (e.g., [ASM$^+$20, Proposition 5]), we give an alternative proof from existing approaches.

## 5 Discussion

In this paper we proved tight last-iterate convergence rates for smooth monotone games when all players act according to the optimistic gradient algorithm, which is no-regret. We believe that there are many fruitful directions for future research. First, it would be interesting to obtain last-iterate rates in the case that each player's actions is constrained to the simplex and they use the *optimistic multiplicative weights update (OMWU)* algorithm. [DP18, LNPW20] showed that OMWU exhibits last-iterate convergence, but non-asymptotic rates remain unknown even for the case that $F_{\mathcal{G}}(\cdot)$ is linear, which includes finite-action polymatrix games. Next, it would be interesting to determine whether Theorem 5 holds if (2) is removed from Assumption 4; this problem is open even for the EG algorithm ([GPDO20]). Finally, it would be interesting to extend our results to the setting where players receive noisy gradients (i.e., the stochastic case). As for lower bounds, it would be interesting to determine whether an algorithm-independent lower bound of $\Omega(1/\sqrt{T})$ in the context of Theorem 7 could be proven for stationary $p$-SCLIs. As far as we are aware, this question is open even for convex minimization (where the rate would be $\Omega(1/T)$).

## Broader impact

As this is a theoretical paper, we expect that the direct ethical and societal impacts of this work will be limited. As the setting of multi-agent learning in games describes many systems with potential for practical impact, such as GANs, we believe that the insights developed in this paper may eventually aid the improvement of such technologies. If not deployed and regulated carefully, technologies such as GANs could lead to harmful outcomes, such as through the proliferation of false media ("deepfakes"). We hope that, through a combination of legal and technological measures, such negative impacts of GANs can be limited and the positive applications, such as drug discovery and image analysis in the medical field, may be realized.

## Acknowledgments and Disclosure of Funding

We thank Yossi Arjevani for a helpful conversation.

N.G. is supported by a Fannie & John Hertz Foundation Fellowship and an NSF Graduate Fellowship. C.D. is supported by NSF Awards IIS-1741137, CCF-1617730 and CCF-1901292, by a Simons Investigator Award, and by the DOE PhILMs project (No. DE-AC05-76RL01830).

## Footnotes

[1]In fact, even in the adversarial setting, standard no-regret algorithms such as FTRL ([SS11]) need to be applied with decreasing step-size in order to achieve sublinear regret.

[2]EG is also known as *mirror-prox*, which specifically refers to its generalization to general Bregman divergences.

[3]In *potential games*, there is a canonical choice of potential function whose local minima are equivalent to being at a Nash equilibrium. The lack of existence of a natural potential function in general monotone games is a significant challenge in establishing last-iterate convergence.

[4]Technically, the result of [Pop80] only applies to two-player zero-sum monotone games (i.e., finding the saddle point of a convex-concave function). The proof readily extends to general monotone VIs ([HIMM19]).

[5]In this discussion we view $\eta, D$ as constants.

[6]For a trivial example, suppose that $n = 1$, $F_{\mathcal{G}}(\mathbf{z}) = \mathbf{z}$, $\mathbf{z}^{(t')} = \delta > 0$, and $\mathbf{z}^{(t'-1)} = 0$. Then $\|(F_{\mathcal{G}}(\mathbf{z}^{(t')}), F_{\mathcal{G}}(\mathbf{z}^{(t'-1)}))\| = \delta$ but $\|(F_{\mathcal{G}}(\mathbf{z}^{(t'+1)}), F_{\mathcal{G}}(\mathbf{z}^{(t')}))\| > \delta\sqrt{2 - 4\eta}$.

[7]We use slightly different terminology from [ASSS15]; technically, the $p$-SCLIs considered in this paper are those in [ASSS15] with *linear coefficient matrices*.

[8]$\sigma_{\min}(\mathbf{M})$ and $\sigma_{\max}(\mathbf{M})$ denote the minimum and maximum singular values of $\mathbf{M}$, respectively. The matrix $\mathbf{M}$ is assumed in [LS18] to be a square matrix of full rank (which holds for the construction used to prove Theorem 7).

[9][AS16] claimed to prove a similar lower bound for stationary algorithms in the setting of smooth convex function minimization; however, as we discuss in Appendix C, their results only apply to the strongly convex case, where they show a linear lower bound.

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
