[Supplementary Material]

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

[10]In particular, they are equivalent in the unconstrained setting when the learning rate $\eta_t$ is constant.

[11]Technically, [LS18] only considered the case where $\mathbf{A} = \begin{pmatrix} \mathbf{0} & \mathbf{M} \\ -\mathbf{M}^\top & \mathbf{0} \end{pmatrix}$ for some matrix $\mathbf{M}$, which corresponds to min-max optimization for bilinear functions, but their proof readily extends to the case we consider in this section.

[12]The invertibility of $\mathbf{M}^{(t)}$, and thus the well-definedness of $\mathbf{C}^{(t-1)}$, is established in Lemma 15.

[13] As we have already noted, this observation establishes also that the preconditions of Lemmas 15 and 16 hold.

[14]More generally, $\mathcal{A}$ may be any algorithm satisfying the conditions of Observation 22.

[15] In fact, $G$ is a conformal mapping, though we will not need this.

[16] In particular, this modified version can be established by only using functions for which the condition number $L/\mu$ is a constant. In more detail, one runs into the following issue when using the machinery of [AS16] to attempt to prove that the iteration complexity of a $p$-SCLI cannot be $O(\kappa^\alpha \ln(1/\epsilon))$ for any $\alpha < 1$: at the end of the proof of [AS16, Theorem 2], Lemma 4 of [AS16] is used to conclude the existence of some $\eta \in (L/2, L)$ satisfying a certain inequality. However, $L/\eta$ represents the condition number $\kappa$ of the problem, and so choosing $\eta \in (L/2, L)$ forces the condition number $\kappa$ of the function to be a constant.

[17] Recall that $f$ is $\ell$-smooth iff its gradient is $\ell$-Lipschitz.

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

# A  Additional preliminaries

## A.1  Proof of Proposition 2

*Proof of Proposition 2.* Fix a game $\mathcal{G}$, and let $F = F_{\mathcal{G}} : \mathcal{Z} \to \mathbb{R}^n$. Monoticity of $F$ gives that for any fixed $\mathbf{z}_{-k} \in \prod_{k' \neq k} \mathcal{Z}_{k'}$, for any $\mathbf{z}_k, \mathbf{z}'_k \in \mathcal{Z}_k$, we have

$$\langle F(\mathbf{z}'_k, \mathbf{z}_{-k}) - F(\mathbf{z}_k, \mathbf{z}_{-k}), (\mathbf{z}'_k, \mathbf{z}_{-k}) - (\mathbf{z}_k, \mathbf{z}_{-k}) \rangle = \langle \nabla_{\mathbf{z}_k} f_k(\mathbf{z}'_k, \mathbf{z}_{-k}) - \nabla_{\mathbf{z}_k}(\mathbf{z}_k, \mathbf{z}_{-k}), \mathbf{z}'_k - \mathbf{z}_k \rangle \geq 0.$$

Since $f_k$ is continuously differentiable, [Nes75, Theorem 2.1.3] gives that $f_k$ is convex. Thus

$$f_k(\mathbf{z}_k, \mathbf{z}_{-k}) - \min_{\mathbf{z}'_k \in \mathcal{Z}'_k} f_k(\mathbf{z}'_k, \mathbf{z}_{-k}) \leq \langle \nabla_{\mathbf{z}_k} f_k(\mathbf{z}_k, \mathbf{z}_{-k}), \mathbf{z}_k - \mathbf{z}'_k \rangle \leq \|\nabla_{\mathbf{z}_k} f_k(\mathbf{z}_k, \mathbf{z}_{-k})\| \cdot D.$$

Summing the above for $k \in \mathcal{K}$ and using the definition of the total and gradient gap functions, as well as Cauch-Schwarz, gives that $\mathrm{TGap}_{\mathcal{G}}^{\mathcal{Z}'}(\mathbf{z}) \leq D \cdot \sum_{k=1}^{K} \|\nabla_{\mathbf{z}_k} f_k(\mathbf{z})\| \leq D\sqrt{K}\|F(\mathbf{z})\|.$  □

## A.2  Optimistic gradient algorithm

In this section we review some additional background about the optimistic gradient algorithm in the setting of no-regret learning. The starting point is *online gradient descent*; player $k$ following online gradient descent produces iterates $\mathbf{z}_k^{(t)} \in \mathcal{Z}_k$ defined by $\mathbf{z}_k^{(t+1)} = \mathbf{z}_k^{(t)} - \eta_t \mathbf{g}_k^{(t)}$, where $\mathbf{g}_k^{(t)} = \nabla_{\mathbf{z}_k} f_k(\mathbf{z}_k^{(t)}, \mathbf{z}_{-k}^{(t)})$ is player $k$'s gradient given its action $\mathbf{z}_k^{(t)}$ and the other players' actions $\mathbf{z}_{-k}^{(t)}$ at time $t$. Online gradient descent is a no-regret algorithm (in particular, it satisfies the same regret bound as OG in Proposition 3); it is also closely related to the *follow-the-regularized-leader (FTRL)* ([SS11]) algorithm from online learning.[10]

The *optimistic gradient (OG)* algorithm ([RS13, DISZ17]) is a modification of online gradient descent, for which player $k$ performs the following update:

$$\mathbf{z}_k^{(t+1)} := \mathbf{z}_k^{(t)} - 2\eta_t \mathbf{g}_k^{(t)} + \eta_t \mathbf{g}_k^{(t-1)}, \tag{OG}$$

where again $\mathbf{g}_k^{(t)} = \nabla_{\mathbf{z}_k} f_k(\mathbf{z}_k^{(t)}, \mathbf{z}_{-k}^{(t)})$ for $t \geq 0$. As way of intuition behind the updates (OG), [DISZ17] observed that OG is closely related to the *optimistic follow-the-regularized-leader (OFTRL)* algorithm from online learning: OFTRL augments the standard FTRL update by using the gradient $\mathbf{g}_k^{(t)}$ at time $t$ as a prediction for the gradient at time $t+1$. When the actions $\mathbf{z}_{-k}^{(t)}$ of the other players are predictable in the sense that they do not change quickly over time, then such a prediction using $\mathbf{g}_k^{(t)}$ is reasonably accurate and can improve the speed of convergence to an equilibrium ([RS13]).

## A.3  Linear regret for extragradient algorithm

In this section we review the definition of the extragradient (EG) algorithm, and show that if one attempts to implement it in the setting of online multi-agent learning, then it is not a no-regret algorithm. Given a monotone game $\mathcal{G}$ and its corresponding monotone operator $F_{\mathcal{G}} : \mathcal{Z} \to \mathbb{R}^n$ and an initial point $\mathbf{u}^{(0)} \in \mathbb{R}^n$ the EG algorithm attempts to find a Nash equilibrium $\mathbf{z}^*$ (i.e., a point satisfying $F_{\mathcal{G}}(\mathbf{z}^*) = 0$) by performing the updates:

$$\mathbf{u}^{(t)} = \Pi_{\mathcal{Z}}(\mathbf{u}^{(t-1)} - \eta F_{\mathcal{G}}(\mathbf{z}^{(t-1)})), \qquad t \geq 1 \tag{10}$$

$$\mathbf{z}^{(t)} = \Pi_{\mathcal{Z}}(\mathbf{u}^{(t)} - \eta F_{\mathcal{G}}(\mathbf{u}^{(t)})), \qquad t \geq 0, \tag{11}$$

where $\Pi_{\mathcal{Z}}(\cdot)$ denotes Euclidean projection onto the convex set $\mathcal{Z}$. Assuming $\mathcal{Z}$ contains a sufficiently large ball centered at $\mathbf{z}^*$, this projection step has no effect for the updates shown above when all players perform EG updates (see Remark 4); the projection is typically needed, however, for the adversarial setting that we proceed to discuss in this section (e.g., as in Proposition 3).

It is easy to see that the updates (10) and (11) can be rewritten as $\mathbf{u}^{(t)} = \Pi_{\mathcal{Z}}(\mathbf{u}^{(t-1)} - \eta F_{\mathcal{G}}(\Pi_{\mathcal{Z}}(\mathbf{u}^{(t-1)} - \eta F_{\mathcal{G}}(\mathbf{u}^{(t-1)}))))$. Note that these updates are somewhat similar to those of OG when expressed as (23) and (24), with $\mathbf{w}^{(t)}$ in (23) and (24) playing a similar role to $\mathbf{u}^{(t)}$ in (10)

and (11). A key difference is that the iterate $\mathbf{u}^{(t)}$ is needed to update $\mathbf{z}^{(t)}$ in (11), whereas this is not true for the update to $\mathbf{z}^{(t)}$ in (23). Since in the standard setting of online multi-agent learning, agents can only see gradients corresponding to actions they play, in order to implement the above EG updates in this setting, we need two timesteps for every timestep of EG. In particular, the agents will play actions $\mathbf{v}^{(t)}$, $t \geq 0$, where $\mathbf{v}^{(2t)} = \mathbf{u}^{(t)}$ and $\mathbf{v}^{(2t+1)} = \mathbf{z}^{(t)}$ for all $t \geq 0$. Recalling that $F_{\mathcal{G}}(\mathbf{z}) = (\nabla_{\mathbf{z}_1} f_1(\mathbf{z}), \ldots, \nabla_{\mathbf{z}_K} f_K(\mathbf{z}))$, this means that player $k \in [K]$ performs the updates

$$\mathbf{v}_k^{(2t)} = \Pi_{\mathcal{Z}_k}(\mathbf{v}_k^{(2t-2)} - \eta \nabla_{\mathbf{z}_k} f_k(\mathbf{v}_k^{(2t-1)}, \mathbf{z}_{-k}^{(2t-1)})), \qquad t \geq 1 \tag{12}$$

$$\mathbf{v}_k^{(2t+1)} = \Pi_{\mathcal{Z}_k}(\mathbf{v}_k^{(2t)} - \eta \nabla_{\mathbf{z}_k} f_k(\mathbf{v}_k^{(2t)}, \mathbf{v}_{-k}^{(2t)})), \qquad t \geq 0, \tag{13}$$

where $\mathbf{v}_k^{(0)} = \mathbf{u}_k^{(0)}$. Unfortunately, as we show in Proposition 10 below, in the setting when the other players' actions $\mathbf{z}_{-k}^{(t)}$ are adversarial (i.e., players apart from $k$ do not necessarily play according to EG), the algorithm for player $k$ given by the EG updates (12) and (13) can have linear regret, i.e., is not a no-regret algorithm. Thus the EG algorithm is insufficient for answering our motivating question $(\star)$.

**Proposition 10.** *There is a set $\mathcal{Z} = \prod_{k=1}^{K} \mathcal{Z}_k$ together with a convex, 1-Lipschitz, and 1-smooth function $f_1 : \mathcal{Z} \to \mathbb{R}$ so that for an adversarial choice of $\mathbf{z}_{-k}^{(t)}$, the EG updates (12) and (13) produce a sequence $\mathbf{v}_k^{(t)}$, $0 \leq t \leq T$ with regret $\Omega(T)$ with respect to the sequence of functions $\mathbf{v}_k \mapsto f_k(\mathbf{v}_k, \mathbf{v}_{-k}^{(t)})$ for any $T > 0$.*

*Proof.* We take $K = 1, k = 1, n = 2, \mathcal{Z}_1 = \mathcal{Z}_2 = [-1, 1]$, and $f_1 : \mathcal{Z}_1 \times \mathcal{Z}_2 \to \mathbb{R}$ to be $f_1(\mathbf{v}_1, \mathbf{v}_2) = \mathbf{v}_1 \cdot \mathbf{v}_2$, where $\mathbf{v}_1, \mathbf{v}_2 \in [-1, 1]$. Consider the following sequence of actions $\mathbf{v}_2^{(t)}$ of player 2:

$$\mathbf{v}_2^{(t)} = 1 \text{ for } t \text{ even}; \qquad \mathbf{v}_2^{(t)} = 0 \text{ for } t \text{ odd}.$$

Suppose that player 1 initializes at $\mathbf{v}_1^{(0)} = 0$. Then for all $t \geq 0$, we have

$$\nabla_{\mathbf{z}_1} f_1(\mathbf{v}_1^{(2t-1)}, \mathbf{v}_2^{(2t-1)}) = \mathbf{v}_2^{(2t-1)} = 0 \qquad \forall t \geq 1$$

$$\nabla_{\mathbf{z}_1} f_1(\mathbf{z}_1^{(2t)}, \mathbf{v}_2^{(2t)}) = \mathbf{v}_2^{(2t)} = 1 \qquad \forall t \geq 0.$$

It follows that for $t \geq 0$ we have $\mathbf{v}_1^{(2t)} = 0$ and $\mathbf{v}_1^{(2t+1)} = \max\{-\eta, -1\}$. Hence for any $T \geq 0$ we have $\sum_{t=0}^{T-1} f_1(\mathbf{v}_1^{(t)}, \mathbf{v}_2^{(t)}) = 0$ whereas

$$\min_{\mathbf{v}_1 \in \mathcal{Z}_1} \sum_{t=0}^{T-1} f_1(\mathbf{v}_1, \mathbf{v}_2^{(t)}) = -\lceil T/2 \rceil,$$

(with the optimal point $\mathbf{v}_1$ being $\mathbf{v}_1^* = -1$) so the regret is $\lceil T/2 \rceil$. $\qquad \square$

### A.4 Prior work on last-iterate rates for noisy feedback

In this section we present Table 2, which exhibits existing last-iterate convergence rates for gradient-based learning algorithms in the case of noisy gradient feedback (i.e., it is an analogue of Table 1 for noisy feedback, leading to stochastic algorithms). We briefly review the setting of noisy feedback: at each time step $t$, each player $k$ plays an action $\mathbf{z}_k^{(t)}$, and receives the feedback

$$\mathbf{g}_k^{(t)} := \nabla_{\mathbf{z}_k} f_k(\mathbf{z}_k^{(t)}, \mathbf{z}_{-k}^{(t)}) + \xi_k^{(t)},$$

where $\xi_k^{(t)} \in \mathbb{R}^{n_k}$ is a random variable satisfying:

$$\mathbb{E}[\xi_k^{(t)} | \mathcal{F}^{(t)}] = \mathbf{0}, \tag{14}$$

where $\mathcal{F} = (\mathcal{F}^{(t)})_{t \geq 0}$ is the filtration given by the sequence of $\sigma$-algebras $\mathcal{F}^{(t)} := \sigma(\mathbf{z}^{(0)}, \mathbf{z}^{(1)}, \ldots, \mathbf{z}^{(t)})$ generated by $\mathbf{z}^{(0)}, \ldots, \mathbf{z}^{(t)}$. Additionally, it is required that the variance of $\xi_k^{(t)}$ be bounded; we focus on the following two possible boundedness assumptions:

$$\mathbb{E}[\|\xi_k^{(t)}\|^2 | \mathcal{F}^{(t)}] \leq \sigma_t^2 \tag{Abs}$$

$$\text{or} \qquad \mathbb{E}[\|\xi_k^{(t)}\|^2 | \mathcal{F}^{(t)}] \leq \tau_t \|F_{\mathcal{G}}(\mathbf{z}^{(t)})\|^2, \tag{Rel}$$

where $\sigma_t > 0$ and $\tau_t > 0$ are sequences of positive reals (typically taken to be decreasing with $t$). Often it is assumed that $\sigma_t$ is the same for all $t$, in which case we write $\sigma = \sigma_t$. Noise model (Abs) is known as *absolute random noise*, and (Rel) is known as *relative random noise* [LZMJ20]. The latter is only of use in the unconstrained setting in which the goal is to find $\mathbf{z}^*$ with $F_{\mathcal{G}}(\mathbf{z}^*) = \mathbf{0}$. While we restrict Table 2 to 1st order methods, we refer the reader also to the recent work of [LBJM$^+$20], which provides last-iterate rates for stochastic Hamiltonian gradient descent, a 2nd order method, in "sufficiently bilinear" games.

As can be seen in Table 2, there is no work to date proving last-iterate rates for general smooth monotone games. We view the problem of extending the results of this paper and of [GPDO20] to the stochastic setting (i.e., the bottom row of Table 2) as an interesting direction for future work.

Table 2: Known upper bounds on last-iterate convergence rates for learning in smooth monotone games with noisy gradient feedback (i.e., *stochastic* algorithms). Rows of the table are as in Table 1; $\ell, \Lambda$ are the Lipschitz constants of $F_{\mathcal{G}}, \partial F_{\mathcal{G}}$, respectively, and $c > 0$ is a sufficiently small absolute constant. The right-hand column contains algorithms implementable as online no-regret learning algorithms: stochastic optimistic gradient (Stoch. OG) or stochastic gradient descent (SGD). The left-hand column contains algorithms not implementable as no-regret algorithms, which includes stochastic extragradient (Stoch. EG), stochastic forward-backward (FB) splitting, double stepsize extragradient (DSEG), and stochastic variance reduced extragradient (SVRE). SVRE only applies in the *finite-sum setting*, which is a special case of (Abs) in which $f_k$ is a sum of $m$ individual loss functions $f_{k,i}$, and a noisy gradient is obtained as $\nabla f_{k,i}$ for a random $i \in [m]$. Due to the stochasticity, many prior works make use of a step size $\eta_t$ that decreases with $t$; we make note of whether this is the case ("$\eta_t$ *decr.*") or whether the step size $\eta_t$ can be constant ("$\eta_t$ *const.*"). For simplicity of presentation we assume $\Omega(1/t) \leq \{\tau_t, \sigma_t\} \leq O(1)$ for all $t \geq 0$ in all cases for which $\sigma_t, \tau_t$ vary with $t$. Reported bounds are stated for the total gap function (Definition 3); leading constants and factors depending on distance between initialization and optimum are omitted.

| Game class | Stochastic | |
| --- | --- | --- |
| | Not implementable as no-regret | Implementable as no-regret |
| $\mu$-strongly monotone | *(Abs):* $\frac{\sigma\ell}{\mu\sqrt{T}}$ [PB16, Stoch. FB splitting, $\eta_t$ decr.] (See also [RVV16, MKS$^+$19]) <br><br> *(Abs):* $\frac{\ell(\sigma+\ell)}{\mu\sqrt{T}}$ [KUS19, Stoch. EG, $\eta_t$ decr.] <br><br> *Finite-sum:* $\ell\left(1 - c\min\{\frac{1}{m}, \frac{\mu}{\ell}\}\right)^T$ [CGFLJ19, SVRE, $\eta_t$ const.] (See also [PB16]) | *(Abs):* $\frac{\sigma\ell}{\mu\sqrt{T}}$ [HIMM19, Stoch. OG, $\eta_t$ decr.] (See also [FOP20]) |
| Monotone, $\gamma$-sing. val. low. bnd. | *(Abs), (Rel):* Stoch. EG may not convg. [CGFLJ19, HIMM20] <br><br> *(Abs):* $\frac{\ell^2\sigma}{\gamma^{3/2}\sqrt[6]{T}}$ [HIMM20, DSEG, $\eta_t$ decr.] | Open |
| $\lambda$-cocoercive | – | *(Rel):* $\frac{1}{\lambda\sqrt{T}} + \sqrt{\frac{\sum_{t\leq T}\tau_t}{T}}$ [LZMJ20, SGD, $\eta_t$ const.] <br><br> *(Abs):* $\frac{\sqrt{\sum_{t\leq t}(t+1)\sigma_t^2}}{\lambda\sqrt{T}}$ [LZMJ20, SGD, $\eta_t$ const.] |
| Monotone | Open | Open |

# B  Proofs for Section 3

In this section we prove Theorem 5. In Section B.1 we show that OG exhibits *best-iterate convergence*, which is a simple consequence of prior work. In Section B.1 we begin to work towards the main contribution of this work, namely showing that best-iterate convergence implies last iterate convergence, treating the special case of linear monotone operators $F(\mathbf{z}) = \mathbf{A}\mathbf{z}$. In Section B.3 we introduce the adaptive potential function for the case of general smooth monotone operators $F$, and finally in Section B.4, using this choice of adaptive potential function, we prove Theorem 5. Some minor lemmas used throughout the proof are deferred to Section B.5.

## B.1 Best-iterate convergence

Throughout this section, fix a monotone game $\mathcal{G}$ satisfying Assumption 4, and write $F = F_{\mathcal{G}}$, so that $F$ is a monotone operator (Definition 1). Recall that the OG algorithm with constant step size $\eta > 0$ is given by:

$$\mathbf{z}^{(-1)}, \mathbf{z}^{(0)} \in \mathbb{R}^n, \qquad \mathbf{z}^{(t+1)} = \mathbf{z}^{(t)} - 2\eta F(\mathbf{z}^{(t)}) + \eta F(\mathbf{z}^{(t-1)}) \ \forall t \geq 0. \tag{15}$$

In Lemma 11 we observe that *some* iterate $\mathbf{z}^{(t^*)}$ of OG has small gradient gap.

**Lemma 11.** *Suppose $F : \mathbb{R}^n \to \mathbb{R}^n$ is a monotone operator that is $\ell$-Lipschitz. Fix some $\mathbf{z}^{(0)}, \mathbf{z}^{(-1)} \in \mathbb{R}^n$, and suppose there is $\mathbf{z}^* \in \mathbb{R}^n$ so that $F(\mathbf{z}^*) = 0$ and $\max\{\|\mathbf{z}^* - \mathbf{z}^{(0)}\|, \|\mathbf{z}^* - \mathbf{z}^{(-1)}\|\} \leq D$. Then the iterates $\mathbf{z}^{(t)}$ of OG for any $\eta < \frac{1}{\ell\sqrt{10}}$ satisfy:*

$$\min_{0 \leq t \leq T-1} \|F(\mathbf{z}^{(t)})\| \leq \frac{4D}{\eta\sqrt{T} \cdot \sqrt{1 - 10\eta^2\ell^2}}. \tag{16}$$

*More generally, we have, for any $S \geq 0$ with $S < T/3$,*

$$\min_{0 \leq t \leq T-S} \max_{0 \leq s < S} \|F(\mathbf{z}^{(t+s)})\| \leq \frac{6D}{\eta\sqrt{T/S} \cdot \sqrt{1 - 10\eta^2\ell^2}}. \tag{17}$$

*Proof.* For all $t \geq 1$, define $\mathbf{w}^{(t)} = \mathbf{z}^{(t)} + \eta F(\mathbf{z}^{(t-1)})$. Equation (B.4) of [HIMM19] gives that for each $t \geq 0$, $\mathbf{z} \in \mathbb{R}^n$

$$\|\mathbf{w}^{(t+1)} - \mathbf{z}\|^2 \leq \|\mathbf{w}^{(t)} - \mathbf{z}\|^2 - 2\eta\langle F(\mathbf{z}^{(t)}), \mathbf{z}^{(t)} - \mathbf{z}\rangle + \eta^2\ell^2\|\mathbf{z}^{(t)} - \mathbf{z}^{(t-1)}\|^2 - \|\eta F(\mathbf{z}^{(t-1)})\|^2.$$

Choosing $\mathbf{z} = \mathbf{z}^*$, using that $\langle F(\mathbf{z}^{(t)}), \mathbf{z}^{(t)} - \mathbf{z}^*\rangle \geq 0$, and applying Young's inequality gives that for $t \geq 1$,

$$\|\mathbf{w}^{(t+1)} - \mathbf{z}^*\|^2 \leq \|\mathbf{w}^{(t)} - \mathbf{z}^*\|^2 + \eta^2\ell^2\|2\eta F(\mathbf{z}^{(t-1)}) - \eta F(\mathbf{z}^{(t-2)})\|^2 - \|\eta F(\mathbf{z}^{(t-1)})\|^2$$

$$\leq \|\mathbf{w}^{(t)} - \mathbf{z}^*\|^2 + (\eta^2\ell^2) \cdot 8\eta^2\|F(\mathbf{z}^{(t-1)})\|^2 + (\eta^2\ell^2) \cdot 2\eta^2\|F(\mathbf{z}^{(t-2)})\|^2 - \eta^2\|F(\mathbf{z}^{(t-1)})\|^2.$$

Summing the above equation for $1 \leq t \leq T-1$ gives

$$\eta^2 \cdot \left( (1 - 8\eta^2\ell^2) \sum_{t=0}^{T-2} \|F(\mathbf{z}^{(t)})\|^2 - 2\eta^2\ell^2 \sum_{t=-1}^{T-3} \|F(\mathbf{z}^{(t)})\|^2 \right) \leq \|\mathbf{w}^{(1)} - \mathbf{z}^*\|^2 - \|\mathbf{w}^{(T-1)} - \mathbf{z}^*\|^2.$$

Since $\|\mathbf{w}^{(1)} - \mathbf{z}^*\| \leq 3D$, $\|F(\mathbf{z}^{(-1)})\| \leq D\ell$, and $2\eta^2\ell^2 \leq 1$, it follows that

$$\min_{0 \leq t \leq T-2} \|F(\mathbf{z}^{(t)})\| \leq \frac{4D}{\eta\sqrt{T-1} \cdot \sqrt{1 - 10\eta^2\ell^2}}.$$

The desired result (16) follows by substituting $T + 1$ for $T$.

To obtain (17), we break $\{0, 1, \ldots, T-2\}$ into $\lfloor (T-1)/S \rfloor$ windows of $S$ consecutive time steps each. Then there must be some $t \in \{0, \ldots, T - 2 - (S - 1)\}$ so that

$$\sum_{s=0}^{S-1} \|F(\mathbf{z}^{(t+s)})\|^2 \leq \frac{(4D)^2}{\eta^2(1 - 10\eta^2\ell^2)\lfloor (T-1)/S \rfloor},$$

from which (17) follows since $S < T/3$. $\qquad\square$

In the remainder of this section we present our main technical contribution in the context of Theorem 5, showing that for a fixed $T$, the last iterate $\mathbf{z}^{(T)}$ does not have gradient gap $\|F(\mathbf{z}^{(T)})\|$ much larger than $\min_{1 \leq t \leq T} \max_{0 \leq s \leq 2} \|F(\mathbf{z}^{(t+s)})\|$.

## B.2 Warm-up: different perspective on the linear case

Before treating the case where $F$ is a general smooth monotone operator, we first explain our proof technique for the case that $F(\mathbf{z}) = \mathbf{A}\mathbf{z}$ for some matrix $\mathbf{A} \in \mathbb{R}^{n \times n}$. This case is covered by [LS18, Theorem 3][11]; the discussion here can be viewed as an alternative perspective on this prior work.

Assume that $F(\mathbf{z}) = \mathbf{A}\mathbf{z}$ for some $\mathbf{A} \in \mathbb{R}^{n \times n}$ throughout this section. Let $\mathbf{z}^{(t)}$ be the iterates of OG, and define

$$\mathbf{w}^{(t)} = \mathbf{z}^{(t)} + \eta F(\mathbf{z}^{(t-1)}) = \mathbf{z}^{(t)} + \eta \mathbf{A}\mathbf{z}^{(t-1)}. \tag{18}$$

Thus the updates of OG can be written as

$$\mathbf{z}^{(t)} = \mathbf{w}^{(t)} - \eta F(\mathbf{z}^{(t-1)}) = \mathbf{w}^{(t)} - \eta \mathbf{A}\mathbf{z}^{(t-1)} \tag{19}$$

$$\mathbf{w}^{(t+1)} = \mathbf{w}^{(t)} - \eta F(\mathbf{z}^{(t)}) = \mathbf{w}^{(t)} - \eta \mathbf{A}\mathbf{z}^{(t)}. \tag{20}$$

The *extra-gradient* (EG) algorithm is the same as the updates (19), (20), except that in (19), $F(\mathbf{z}^{(t-1)})$ is replaced with $F(\mathbf{w}_t)$. As such, OG in this context is often referred to as *past extragradient (PEG)* [HIMM19]. Many other works have also made use of this interpretation of OG, e.g., [RS12, RS13, Pop80].

Now define

$$\mathbf{C} = \frac{(I + (2\eta\mathbf{A})^2)^{1/2} - I}{2} = \eta^2 \mathbf{A}^2 + O((\eta\mathbf{A})^4), \tag{21}$$

where the square root of $I + (2\eta\mathbf{A})^2$ may be defined via the power series $\sqrt{I - \mathbf{X}} := \sum_{j=0}^{\infty} \mathbf{X}^k (-1)^k \binom{1/2}{k}$. It is easy to check that $\mathbf{C}$ is well-defined as long as $\eta \leq O(1/\ell) \leq O(1/\|\mathbf{A}\|_\sigma)$, and that $\mathbf{C}\mathbf{A} = \mathbf{A}\mathbf{C}$. Also note that $\mathbf{C}$ satisfies

$$\mathbf{C}^2 + \mathbf{C} = \eta^2 \mathbf{A}^2. \tag{22}$$

Finally set

$$\tilde{\mathbf{w}}^{(t)} = \mathbf{w}^{(t)} + \mathbf{C}\mathbf{z}^{(t-1)},$$

so that $\tilde{\mathbf{w}}^{(t)}$ corresponds (under the PEG interpretation of OG) to the iterates $\mathbf{w}^{(t)}$ of EG, plus an "adjustment" term, $\mathbf{C}\mathbf{z}^{(t)}$, which is $O((\eta\mathbf{A})^2)$. Though this adjustment term is small, it is crucial in the following calculation:

$$
\begin{aligned}
\tilde{\mathbf{w}}^{(t+1)} &= \mathbf{w}^{(t+1)} + \mathbf{C}\mathbf{z}^{(t)} \\
&\overset{(20)}{=} \mathbf{w}^{(t)} - \eta\mathbf{A}\mathbf{z}^{(t)} + \mathbf{C}\mathbf{z}^{(t)} \\
&\overset{(19)}{=} \mathbf{w}^{(t)} + (\mathbf{C} - \eta\mathbf{A})(\mathbf{w}^{(t)} - \eta\mathbf{A}\mathbf{z}^{(t-1)}) \\
&= (I - \eta\mathbf{A} + \mathbf{C})\mathbf{w}^{(t)} + (\eta^2\mathbf{A}^2 - \eta\mathbf{A}\mathbf{C})\mathbf{z}^{(t-1)} \\
&\overset{(22)}{=} (I - \eta\mathbf{A} + \mathbf{C})(\mathbf{w}^{(t)} + \mathbf{C}\mathbf{z}^{(t-1)}) \\
&= (I - \eta\mathbf{A} + \mathbf{C})\tilde{\mathbf{w}}^{(t)}.
\end{aligned}
$$

Since $\mathbf{C}, \mathbf{A}$ commute, the above implies that $F(\tilde{\mathbf{w}}^{(t+1)}) = (I - \eta\mathbf{A} + \mathbf{C})F(\tilde{\mathbf{w}}^{(t)})$. Monotonicity of $F$ implies that for $\eta = O(1/\ell)$, we have $\|I - \eta\mathbf{A} + \mathbf{C}\|_\sigma \leq 1$. It then follows that $\|F(\tilde{\mathbf{w}}^{(t+1)})\| \leq \|F(\tilde{\mathbf{w}}^{(t)})\|$, which establishes that the last iterate is the best iterate.

## B.3 Setting up the adaptive potential function

We next extend the argument of the previous section to the smooth convex-concave case, which will allow us to prove Theorem 5 in its full generality. Recall the PEG formulation of OG introduced in the previous section:

$$\mathbf{z}^{(t)} = \mathbf{w}^{(t)} - \eta F(\mathbf{z}^{(t-1)}) \tag{23}$$

$$\mathbf{w}^{(t+1)} = \mathbf{w}^{(t)} - \eta F(\mathbf{z}^{(t)}), \tag{24}$$

where again $\mathbf{z}^{(t)}$ denote the iterates of OG (15).

As discussed in Section 3.1, the adaptive potential function is given by $\|\tilde{F}^{(t)}\|$, where

$$\tilde{F}^{(t)} := F(\mathbf{w}^{(t)}) + \mathbf{C}^{(t-1)} \cdot F(\mathbf{z}^{(t-1)}) \in \mathbb{R}^n, \tag{25}$$

for some matrices $\mathbf{C}^{(t)} \in \mathbb{R}^{n \times n}$, $-1 \le t \le T$, to be chosen later. Then:

$$\begin{aligned}
\tilde{F}^{(t+1)} &= F(\mathbf{w}^{(t+1)}) + \mathbf{C}^{(t)} \cdot F(\mathbf{z}^{(t)}) \\
&\overset{(24)}{=} F(\mathbf{w}^{(t)} - \eta F(\mathbf{z}^{(t)})) + \mathbf{C}^{(t)} \cdot F(\mathbf{z}^{(t)}) \\
&= F(\mathbf{w}^{(t)}) - \eta \mathbf{A}^{(t)} F(\mathbf{z}^{(t)}) + \mathbf{C}^{(t)} \cdot F(\mathbf{z}^{(t)}) \\
&\overset{(23)}{=} F(\mathbf{w}^{(t)}) + (\mathbf{C}^{(t)} - \eta \mathbf{A}^{(t)}) \cdot F(\mathbf{w}^{(t)} - \eta F(\mathbf{z}^{(t-1)})) \\
&= F(\mathbf{w}^{(t)}) + (\mathbf{C}^{(t)} - \eta \mathbf{A}^{(t)}) \cdot (F(\mathbf{w}^{(t)}) - \eta \mathbf{B}^{(t)} F(\mathbf{z}^{(t-1)})) \\
&= (I - \eta \mathbf{A}^{(t)} + \mathbf{C}^{(t)}) \cdot F(\mathbf{w}^{(t)}) + \eta(\eta \mathbf{A}^{(t)} - \mathbf{C}^{(t)})\mathbf{B}^{(t)} \cdot F(\mathbf{z}^{(t-1)}), \tag{26}
\end{aligned}$$

where

$$\mathbf{A}^{(t)} := \int_0^1 \partial F(\mathbf{w}^{(t)} - (1 - \alpha)\eta F(\mathbf{z}^{(t)}))d\alpha$$

$$\mathbf{B}^{(t)} := \int_0^1 \partial F(\mathbf{w}^{(t)} - (1 - \alpha)\eta F(\mathbf{z}^{(t-1)}))d\alpha.$$

(Recall that $\partial F(\cdot)$ denotes the Jacobian of $F$.) We state the following lemma for later use:

**Lemma 12.** *For each $t$, $\mathbf{A}^{(t)} + (\mathbf{A}^{(t)})^\top, \mathbf{B}^{(t)} + (\mathbf{B}^{(t)})^\top$ are PSD, and $\|\mathbf{A}^{(t)}\|_\sigma \le \ell, \|\mathbf{B}^{(t)}\|_\sigma \le \ell$. Moreover, it holds that*

$$\|\mathbf{A}^{(t)} - \mathbf{B}^{(t)}\|_\sigma \le \frac{\eta\Lambda}{2}\|F(\mathbf{z}^{(t)}) - F(\mathbf{z}^{(t-1)})\|$$

$$\|\mathbf{A}^{(t)} - \mathbf{A}^{(t+1)}\|_\sigma \le \Lambda\|\mathbf{w}^{(t)} - \mathbf{w}^{(t+1)}\| + \frac{\eta\Lambda}{2}\|F(\mathbf{z}^{(t)}) - F(\mathbf{z}^{(t+1)})\|$$

$$\|\mathbf{B}^{(t)} - \mathbf{B}^{(t+1)}\|_\sigma \le \Lambda\|\mathbf{w}^{(t)} - \mathbf{w}^{(t+1)}\| + \frac{\eta\Lambda}{2}\|F(\mathbf{z}^{(t-1)}) - F(\mathbf{z}^{(t)})\|.$$

*Proof.* For all $\mathbf{z} \in \mathbb{R}^n$, monotonicity of $F$ gives that $\partial F(\mathbf{z}) + \partial F(\mathbf{z})^\top$ is PSD, which means that so are $\mathbf{A}^{(t)} + (\mathbf{A}^{(t)})^\top, \mathbf{B}^{(t)} + (\mathbf{B}^{(t)})^\top$. Similarly, (1) gives that for all $\mathbf{z} \in \mathbb{R}^n$, $\|\partial F(\mathbf{z})\|_\sigma \le \ell$, from which we get $\|\mathbf{A}^{(t)}\|_\sigma \le \ell, \|\mathbf{B}^{(t)}\|_\sigma \le \ell$ by the triangle inequality.

The remaining three inequalities are an immediate consequence of the triangle inequality and the fact that $\partial F$ is $\Lambda$-Lipschitz (Assumption 4). $\square$

Now define the following $n \times n$ matrices:

$$\mathbf{M}^{(t)} := I - \eta \mathbf{A}^{(t)} + \mathbf{C}^{(t)}$$

$$\mathbf{N}^{(t)} := \eta(\eta \mathbf{A}^{(t)} - \mathbf{C}^{(t)})\mathbf{B}^{(t)}.$$

Moreover, for a positive semidefinite (PSD) matrix $\mathbf{S} \in \mathbb{R}^{n \times n}$ and a vector $\mathbf{v} \in \mathbb{R}^n$, write $\|\mathbf{v}\|_{\mathbf{S}}^2 := \mathbf{v}^\top \mathbf{S} \mathbf{v}$, so that for a matrix $\mathbf{M} \in \mathbb{R}^{n \times n}$ and a vector $\mathbf{v} \in \mathbb{R}^n$, we have

$$\|\mathbf{v}\|_{\mathbf{M}^\top \mathbf{M}}^2 := \mathbf{v}^\top \mathbf{M}^\top \mathbf{M} \mathbf{v} = \|\mathbf{M}\mathbf{v}\|_2^2.$$

Then by (26),

$$\begin{aligned}
\|\tilde{F}^{(t+1)}\|^2 &= \|\mathbf{M}^{(t)} \cdot F(\mathbf{w}^{(t)}) + \mathbf{N}^{(t)} \cdot F(\mathbf{z}^{(t-1)})\|^2 \\
&= \|F(\mathbf{w}^{(t)}) + (\mathbf{M}^{(t)})^{-1}\mathbf{N}^{(t)} \cdot F(\mathbf{z}^{(t-1)})\|_{(\mathbf{M}^{(t)})^\top \mathbf{M}^{(t)}}^2. \tag{27}
\end{aligned}$$

Next we define $\mathbf{C}^{(T)} = \mathbf{0}$ and for $-1 \le t < T$,[12]

$$\mathbf{C}^{(t-1)} := (\mathbf{M}^{(t)})^{-1}\mathbf{N}^{(t)}. \tag{28}$$

Notice that the definition of $\mathbf{C}^{(t-1)}$ in (28) depends on $\mathbf{C}^{(t)}$, which depends on $\mathbf{C}^{(t+1)}$, and so on. By (27) and (25), it follows that

$$\|\tilde{F}^{(t+1)}\|^2 = \|F(\mathbf{w}^{(t)}) + \mathbf{C}^{(t-1)} \cdot F(\mathbf{z}^{(t-1)})\|^2_{(\mathbf{M}^{(t)})^\top \mathbf{M}^{(t)}} \qquad (29)$$

$$= \|\tilde{F}^{(t)}\|^2_{(\mathbf{M}^{(t)})^\top \mathbf{M}^{(t)}}$$

$$= \|(I - \eta\mathbf{A}^{(t)} + \mathbf{C}^{(t)})\tilde{F}^{(t)}\|^2$$

$$\leq \|I - \eta\mathbf{A}^{(t)} + \mathbf{C}^{(t)}\|^2_\sigma \|\tilde{F}^{(t)}\|^2. \qquad (30)$$

Our goal from here on is two-fold: (1) to prove an upper bound on $\|I - \eta\mathbf{A}^{(t)} + \mathbf{C}^{(t)}\|_\sigma$, which will ensure, by (30), that $\|\tilde{F}^{(t+1)}\| \lesssim \|\tilde{F}^{(t)}\|$, and (2) to ensure that $\|\tilde{F}^{(t)}\|$ is an (approximate) upper bound on $\|F(\mathbf{z}^{(t)})\|$ for all $t$, so that in particular upper bounding $\|\tilde{F}^{(T)}\|$ suffices to upper bound $\|F(\mathbf{z}^{(T)})\|$. These tasks will be performed in the following section; we first make a few remarks on the choice of $\mathbf{C}^{(t-1)}$ in (28):

**Remark 6** (Specialization to the linear case & experiments)**.** In the case that the monotone operator $F$ is linear, i.e., $F(\mathbf{z}) = \mathbf{A}\mathbf{z}$, it is straightforward to check that the matrices $\mathbf{C}^{(t-1)}$ as defined in (28) are all equal to the matrix $\mathbf{C}$ defined in (21) and $\mathbf{A}^{(t)} = \mathbf{B}^{(t)} = \mathbf{A}$ for all $t$. A special case of a linear operator $F$ is that corresponding to a two-player zero-sum matrix game, i.e., where the payoffs of the players given actions $\mathbf{x}, \mathbf{y}$, are $\pm\mathbf{x}^\top\mathbf{M}\mathbf{y}$. In experiments we conducted for random instances of such matrix games, we observe that the adaptive potential function $\tilde{F}^{(t)}$ closely tracks $F(\mathbf{z}^{(t)})$, and both are monotonically decreasing with $t$. It seems that any "interesting" behavior whereby $F(\mathbf{z}^{(t)})$ grows by (say) a constant factor over the course of one or more iterations, but where $\tilde{F}^{(t)}$ grows only by much less, must occur for more complicated monotone operators (if at all). We leave a detailed experimental evaluation of such possibilities to future work.

**Remark 7** (Alternative choice of $\mathbf{C}^{(t)}$)**.** It is not necessary to choose $\mathbf{C}^{(t-1)}$ as in (28). Indeed, in light of the fact that it is the spectral norms $\|I - \eta\mathbf{A}^{(t-1)} + \mathbf{C}^{(t-1)}\|_\sigma$ that control the increase in $\|\tilde{F}^{(t-1)}\|$ to $\|\tilde{F}^{(t)}\|$, it is natural to try to set

$$\tilde{\mathbf{C}}^{(t-1)} = \arg\min_{\mathbf{C}\in\mathbb{R}^{n\times n}} \left[ \|I - \eta\mathbf{A}^{(t-1)} + \mathbf{C}\|_\sigma \,\Big|\, \left\{\|\mathbf{C}\|_\sigma \leq \frac{1}{10}\right\} \text{ and } * \right], \qquad (31)$$

where

$$* = \left\{ \|F(\mathbf{w}^{(t)}) + \mathbf{C} \cdot F(\mathbf{z}^{(t-1)})\|^2_{(\mathbf{M}^{(t)})^\top \mathbf{M}^{(t)}} \geq \|F(\mathbf{w}^{(t)}) + (\mathbf{M}^{(t)})^{-1}\mathbf{N}^{(t)} \cdot F(\mathbf{z}^{(t-1)})\|^2_{(\mathbf{M}^{(t)})^\top \mathbf{M}^{(t)}} \right\}. \qquad (32)$$

The reason for the constraint $*$ defined in (32) is to ensure that $\|\tilde{F}^{(t+1)}\|^2 \leq \|\tilde{F}^{(t)}\|^2_{(\mathbf{M}^{(t)})^\top \mathbf{M}^{(t)}}$ (so that (29) is replaced with an inequality). The reason for the constraint $\|\mathbf{C}\|_\sigma \leq 1/10$ is to ensure that $\|F(\mathbf{z}^{(T)})\| \leq O\left(\|\tilde{F}^{(T)}\|\right)$. Though the asymptotic rate of $O(1/\sqrt{T})$ established by the choice of $\mathbf{C}^{(t-1)}$ in (28) is tight in light of Theorem 7, it is possible that a choice of $\mathbf{C}^{(t-1)}$ as in (31) could lead to an improvement in the absolute constant. We leave an exploration of this possibility to future work.

## B.4 Proof of Theorem 5

In this section we prove Theorem 5 using the definition of $\tilde{F}^{(t)}$ in (25), where $\mathbf{C}^{(t-1)}$ is defined in (28). We begin with a few definitions: for positive semidefinie matrices $\mathbf{S}, \mathbf{T}$, write $\mathbf{S} \preceq \mathbf{T}$ if $\mathbf{T} - \mathbf{S}$ is positive semidefinite (this is known as the *Loewner ordering*). We also define

$$\mathbf{D}^{(t)} := -\eta\mathbf{C}^{(t)}\mathbf{B}^{(t)} + (I - \eta\mathbf{A}^{(t)} + \mathbf{C}^{(t)})^{-1}(\eta\mathbf{A}^{(t)} - \mathbf{C}^{(t)})^2\eta\mathbf{B}^{(t)} \qquad \forall t \leq T - 1. \qquad (33)$$

To understand the definition of the matrices $\mathbf{D}^{(t)}$ in (33), note that, in light of the equality

$$(I - \mathbf{X})^{-1}\mathbf{X} = \mathbf{X} + (I - \mathbf{X})^{-1}\mathbf{X}^2 \qquad (34)$$

for a square matrix $\mathbf{X}$ for which $I - \mathbf{X}$ is invertible, we have, for $t \leq T$,

$$
\begin{aligned}
&I - \eta\mathbf{A}^{(t-1)} + \mathbf{C}^{(t-1)} \\
={}&I - \eta\mathbf{A}^{(t-1)} + (I - \eta\mathbf{A}^{(t)} + \mathbf{C}^{(t)})^{-1}(\eta\mathbf{A}^{(t)} - \mathbf{C}^{(t)})\eta\mathbf{B}^{(t)} \\
={}&I - \eta\mathbf{A}^{(t-1)} + \eta^2\mathbf{A}^{(t)}\mathbf{B}^{(t)} + \left(-\eta\mathbf{C}^{(t)}\mathbf{B}^{(t)} + (I - \eta\mathbf{A}^{(t)} + \mathbf{C}^{(t)})^{-1}(\eta\mathbf{A}^{(t)} - \mathbf{C}^{(t)})^2\eta\mathbf{B}^{(t)}\right) \\
={}&I - \eta\mathbf{A}^{(t-1)} + \eta^2\mathbf{A}^{(t)}\mathbf{B}^{(t)} + \mathbf{D}^{(t)}.
\end{aligned}
\tag{35}
$$

Thus, to upper bound $\|I - \mathbf{A}^{(t-1)} + \mathbf{C}^{(t-1)}\|$, it will suffice to use the below lemma, which generalizes [GPDO20, Lemma 12] and can be used to give an upper bound on the spectral norm of $I - \eta\mathbf{A}^{(t-1)} + \eta^2\mathbf{A}^{(t)}\mathbf{B}^{(t)} + \mathbf{D}^{(t)}$ for each $t$:

**Lemma 13.** *Suppose* $\mathbf{A}_1, \mathbf{A}_2, \mathbf{B}, \mathbf{D} \in \mathbb{R}^{n \times n}$ *are matrices and* $K, L_0, L_1, L_2, \delta > 0$ *so that:*

- $\mathbf{A}_1 + \mathbf{A}_2^\top$, $\mathbf{A}_2 + \mathbf{A}_2^\top$, *and* $\mathbf{B} + \mathbf{B}^\top$ *are PSD;*

- $\|\mathbf{A}_1\|_\sigma, \|\mathbf{A}_2\|_\sigma, \|\mathbf{B}\|_\sigma \leq L_0 \leq 1/106$;

- $\mathbf{D} + \mathbf{D}^\top \preceq L_1 \cdot \left(\mathbf{B}^\top\mathbf{B} + \mathbf{A}_1\mathbf{A}_1^\top\right) + K\delta^2 \cdot I.$

- $\mathbf{D}^\top\mathbf{D} \preceq L_2 \cdot \mathbf{B}^\top\mathbf{B}.$

- $10L_0 + \frac{4L_2}{L_0^2} + 5L_1 \leq 24/50.$

- *For any two matrices* $\mathbf{X}, \mathbf{Y} \in \{\mathbf{A}_1, \mathbf{A}_2, \mathbf{B}\}$, $\|\mathbf{X} - \mathbf{Y}\|_\sigma \leq \delta$.

*It follows that*

$$
\|I - \mathbf{A}_1 + \mathbf{A}_2\mathbf{B} + \mathbf{D}\|_\sigma \leq \sqrt{1 + (K + 400)\,\delta^2}.
$$

*Proof of Lemma 13.* We wish to show that

$$
(I - \mathbf{A}_1 + \mathbf{A}_2\mathbf{B} + \mathbf{D})^\top(I - \mathbf{A}_1 + \mathbf{A}_2\mathbf{B} + \mathbf{D}) \preceq \left(1 + (K + 400)\cdot\delta^2\right)I,
$$

or equivalently

$$
\begin{aligned}
&(\mathbf{A}_1 + \mathbf{A}_1^\top) - (\mathbf{B}^\top\mathbf{A}_2^\top + \mathbf{A}_2\mathbf{B}) - \mathbf{A}_1^\top\mathbf{A}_1 + (\mathbf{B}^\top\mathbf{A}_2^\top\mathbf{A}_1 + \mathbf{A}_1^\top\mathbf{A}_2\mathbf{B}) - \mathbf{B}^\top\mathbf{A}_2^\top\mathbf{A}_2\mathbf{B} \\
&- (\mathbf{D}^\top + \mathbf{D}) + (\mathbf{D}^\top\mathbf{A}_1 + \mathbf{A}_1^\top\mathbf{D}) - (\mathbf{D}^\top\mathbf{A}_2\mathbf{B} + \mathbf{B}^\top\mathbf{A}_2^\top\mathbf{D}) - \mathbf{D}^\top\mathbf{D} \succeq -(K + 400)\cdot\delta^2 I.
\end{aligned}
\tag{36}
$$

For $i \in \{1, 2\}$, let us write $\mathbf{J}_i = (\mathbf{A}_i - \mathbf{A}_i^\top)/2$, $\mathbf{R}_i = (\mathbf{A}_i + \mathbf{A}_i^\top)/2$, and $\mathbf{K} = (\mathbf{B} - \mathbf{B}^\top)/2$, $\mathbf{S} = (\mathbf{B} + \mathbf{B}^\top)/2$, so that $\mathbf{R}_1, \mathbf{R}_2, \mathbf{S}$ are positive semidefinite and $\mathbf{J}_1, \mathbf{J}_2, \mathbf{K}$ are anti-symmetric.

Next we will show (in (42) below) that the sum of all terms in (36) apart from the first four are preceded by a constant (depending on $L_0, L_1$) times $\mathbf{B}^\top\mathbf{B}$ in the Loewner ordering. To show this we begin as follows: for any $\epsilon, \epsilon_1 > 0$, we have:

$$
\text{(Lemma 18)} \qquad \mathbf{A}_1^\top\mathbf{A}_1 \preceq (1 + \epsilon_1)\cdot\mathbf{B}^\top\mathbf{B} + \left(1 + \frac{1}{\epsilon_1}\right)\delta^2 I \tag{37}
$$

$$
\text{(Lemma 17)} \qquad -\mathbf{B}^\top\mathbf{A}_2^\top\mathbf{A}_1 - \mathbf{A}_1^\top\mathbf{A}_2\mathbf{B} \preceq \epsilon\cdot\mathbf{B}^\top\mathbf{B} + \frac{1}{\epsilon}\cdot\mathbf{A}_1^\top\mathbf{A}_2\mathbf{A}_2^\top\mathbf{A}_1
$$

$$
\text{(Lemma 20)} \qquad \preceq \epsilon\cdot\mathbf{B}^\top\mathbf{B} + \frac{L_0^2}{\epsilon}\cdot\mathbf{A}_1^\top\mathbf{A}_1
$$

$$
\text{(Lemma 18)} \qquad \preceq \left(\epsilon + \frac{2L_0^2}{\epsilon}\right)\cdot\mathbf{B}^\top\mathbf{B} + \frac{2L_0^2}{\epsilon}\delta^2 I \tag{38}
$$

$$
\text{(Lemma 20)} \qquad \mathbf{B}^\top\mathbf{A}_2^\top\mathbf{A}_2\mathbf{B} \preceq L_0^2\mathbf{B}^\top\mathbf{B}. \tag{39}
$$

Note in particular that (37), (38), and (39) imply that

$$
(\mathbf{A}_1 - \mathbf{A}_2\mathbf{B})^\top(\mathbf{A}_1 - \mathbf{A}_2\mathbf{B}) \preceq \left(1 + \epsilon + \epsilon_1 + \frac{2L_0^2}{\epsilon} + L_0^2\right)\cdot\mathbf{B}^\top\mathbf{B} + \left(1 + \frac{2L_0^2}{\epsilon} + \frac{1}{\epsilon_1}\right)\delta^2 I,
$$

and choosing $\epsilon = L_0 \leq 1$ (whereas $\epsilon_1$ is left as a free parameter to be specified below) gives

$$(\mathbf{A}_1 - \mathbf{A}_2\mathbf{B})^\top (\mathbf{A}_1 - \mathbf{A}_2\mathbf{B}) \preceq (1 + 4L_0 + \epsilon_1) \cdot \mathbf{B}^\top\mathbf{B} + \left(1 + 2L_0 + \frac{1}{\epsilon_1}\right) \cdot \delta^2 I. \qquad (40)$$

It follows from (40) and Lemma 17 that

$$(\mathbf{A}_2\mathbf{B} - \mathbf{A}_1)^\top \mathbf{D} + \mathbf{D}^\top (\mathbf{A}_2\mathbf{B} - \mathbf{A}_1)$$
$$\preceq \min_{\epsilon > 0, \epsilon_1 > 0} \epsilon \cdot \left((1 + 4L_0 + \epsilon_1) \cdot \mathbf{B}^\top\mathbf{B} + \left(1 + 2L_0 + \frac{1}{\epsilon_1}\right) \cdot \delta^2 I\right) + \frac{1}{\epsilon} \cdot L_2 \mathbf{B}^\top\mathbf{B}$$
$$\preceq \left(2L_0^2 + \frac{L_2}{L_0^2}\right) \cdot \mathbf{B}^\top\mathbf{B} + (2L_0^2 + L_0) \cdot \delta^2 I, \qquad (41)$$

where the last line results from the choice $\epsilon = L_0^2, \epsilon_1 = L_0$.
By (40) and (41) we have, for any $\epsilon_1 > 0$,

$$(\mathbf{A}_1 - \mathbf{A}_2\mathbf{B})^\top (\mathbf{A}_1 - \mathbf{A}_2\mathbf{B}) + (\mathbf{A}_2\mathbf{B} - \mathbf{A}_1)^\top \mathbf{D} + \mathbf{D}^\top (\mathbf{A}_2\mathbf{B} - \mathbf{A}_1) + (\mathbf{D}^\top + \mathbf{D}) + \mathbf{D}^\top\mathbf{D}$$
$$\preceq \left(1 + 4L_0 + \epsilon_1 + 2L_0^2 + \frac{L_2}{L_0^2} + L_1 + L_2\right) \cdot \mathbf{B}^\top\mathbf{B} + L_1 \cdot \mathbf{A}_1\mathbf{A}_1^\top + \left(K + 1 + 2L_0 + \frac{1}{\epsilon_1} + 2L_0^2 + L_0\right) \cdot \delta^2 I$$
$$\preceq \left(1 + 5L_0 + \epsilon_1 + \frac{2L_2}{L_0^2} + L_1\right) \cdot \mathbf{B}^\top\mathbf{B} + L_1 \cdot \mathbf{A}_1\mathbf{A}_1^\top + \left(K + 1 + 4L_0 + \frac{1}{\epsilon_1}\right) \cdot \delta^2 I. \qquad (42)$$

Next, for any $\epsilon > 0$, it holds that

$$\mathbf{B}^\top\mathbf{A}_2^\top + \mathbf{A}_2\mathbf{B}$$
$$= -(\mathbf{K}^\top\mathbf{J}_2 + \mathbf{J}_2^\top\mathbf{K}) + (\mathbf{SR}_2 + \mathbf{R}_2\mathbf{S}) + (\mathbf{SJ}_2^\top + \mathbf{J}_2\mathbf{S}) + (\mathbf{K}^\top\mathbf{R}_2 + \mathbf{R}_2\mathbf{K})$$
$$\text{(Lemma 17)} \preceq -(\mathbf{K}^\top\mathbf{J}_2 + \mathbf{J}_2^\top\mathbf{K}) + (\mathbf{SR}_2 + \mathbf{R}_2\mathbf{S}) + \frac{1}{\epsilon} \cdot (\mathbf{S}^2 + \mathbf{R}_2^2) + \epsilon \cdot (\mathbf{J}_2\mathbf{J}_2^\top + \mathbf{K}^\top\mathbf{K})$$
$$\text{(Lemma 19)} \preceq -(\mathbf{K}^\top\mathbf{J}_2 + \mathbf{J}_2^\top\mathbf{K}) + 3\mathbf{S}^2 + \frac{1}{\epsilon} \cdot (\mathbf{S}^2 + \mathbf{R}_2^2) + \epsilon \cdot (\mathbf{J}_2\mathbf{J}_2^\top + \mathbf{K}^\top\mathbf{K}) + 2\delta^2 I$$
$$\text{(Lemma 18)} \preceq -(\mathbf{K}^\top\mathbf{J}_2 + \mathbf{J}_2^\top\mathbf{K}) + \left(3 + \frac{3}{\epsilon}\right)\mathbf{S}^2 + 3\epsilon \cdot \mathbf{K}^\top\mathbf{K} + \left(2 + \frac{2}{\epsilon} + 2\epsilon\right)\delta^2 I. \qquad (43)$$

Next, we have for any $\epsilon > 0$,

$$\mathbf{A}_1\mathbf{A}_1^\top = \mathbf{R}_1\mathbf{R}_1^\top + (\mathbf{J}_1\mathbf{R}_1^\top + \mathbf{R}_1\mathbf{J}_1^\top) + \mathbf{J}_1\mathbf{J}_1^\top$$
$$\text{(Lemma 17)} \quad \preceq 2\mathbf{R}_1\mathbf{R}_1^\top + 2\mathbf{J}_1\mathbf{J}_1^\top$$
$$= 2\mathbf{R}_1\mathbf{R}_1^\top + 2\mathbf{J}_1^\top\mathbf{J}_1$$
$$\text{(Lemma 18)} \quad \preceq (2 + 2\epsilon)\mathbf{S}^2 + (2 + 2\epsilon)\mathbf{J}_2^\top\mathbf{J}_2 + \frac{4}{\epsilon} \cdot \delta^2 I. \qquad (44)$$

By (43) and (44), for any $\mu, \nu \in (0,1)$ and $\epsilon > 0$ with $2\nu + 10\epsilon + \mu \cdot (2 + 2\epsilon) \le 1$,

$$\mathbf{B}^\top \mathbf{A}_2^\top + \mathbf{A}_2 \mathbf{B} + (1+\nu)\mathbf{B}^\top \mathbf{B} + \mu \mathbf{A}\mathbf{A}^\top$$

$$\preceq -(\mathbf{K}^\top \mathbf{J}_2 + \mathbf{J}_2^\top \mathbf{K}) + 3\epsilon \mathbf{K}^\top \mathbf{K} + (1+\nu)\mathbf{K}^\top \mathbf{K} + \mu \cdot (2+2\epsilon)\mathbf{J}_2^\top \mathbf{J}_2 + \left(4 + \nu + \frac{3}{\epsilon} + \mu \cdot (2+2\epsilon)\right) \mathbf{S}^2$$

$$+ (1+\nu)(\mathbf{K}^\top \mathbf{S} + \mathbf{S}\mathbf{K}) + \left(2 + \frac{2}{\epsilon} + 2\epsilon + \frac{4\mu}{\epsilon}\right)\delta^2 I$$

$$\preceq -(\mathbf{K}^\top \mathbf{J}_2 + \mathbf{J}_2^\top \mathbf{K}) + (1 + \nu + 3\epsilon + (1+\nu)\epsilon)\,\mathbf{K}^\top \mathbf{K} + \mu \cdot (2+2\epsilon)\mathbf{J}_2^\top \mathbf{J}_2$$

$$+ \left(4 + \nu + \frac{3}{\epsilon} + \frac{1+\nu}{\epsilon} + \mu \cdot (2+2\epsilon)\right)\mathbf{S}^2 + \left(2 + \frac{2+4\mu}{\epsilon} + 2\epsilon\right)\delta^2 I \tag{45}$$

$$\preceq -(\mathbf{K}^\top \mathbf{J}_2 + \mathbf{J}_2^\top \mathbf{K}) + \mathbf{K}^\top \mathbf{K} + (2\nu + 10\epsilon + \mu(2+2\epsilon))\mathbf{J}_2^\top \mathbf{J}_2$$

$$+ \left(5 + \frac{5}{\epsilon} + \mu \cdot (2+2\epsilon)\right)\mathbf{S}^2 + \left(4 + \frac{2+4\mu}{\epsilon} + 2\epsilon\right)\delta^2 I \tag{46}$$

$$\preceq (\mathbf{J}_2 - \mathbf{K})^\top (\mathbf{J}_2 - \mathbf{K}) + \left(6 + \frac{5}{\epsilon}\right)\mathbf{S}^2 + \left(4 + \frac{4}{\epsilon} + 2\epsilon\right)\delta^2 I \tag{47}$$

$$\preceq \left(12 + \frac{10}{\epsilon}\right)\mathbf{R}_1^2 + \left(17 + \frac{14}{\epsilon} + 2\epsilon\right)\delta^2 I \tag{48}$$

$$\preceq \left(12 + \frac{10}{\epsilon}\right)L_0 \mathbf{R}_1 + \left(17 + \frac{14}{\epsilon} + 2\epsilon\right)\delta^2 I. \tag{49}$$

where (45) follows from Lemma 17, (46) follows from Lemma 18 and $\nu + 5\epsilon \le 1$, (47) follows from $2\nu + 10\epsilon + \mu \cdot (2+2\epsilon) \le 1$, (48) follows from $\|\mathbf{J}_2 - \mathbf{K}\|_\sigma \le \delta$ as well as Lemma 18, and (49) follows from Lemma 20 together with $\|\mathbf{R}_1^{1/2}\|_\sigma \le \sqrt{L_0}$.

By (42) and (49), by choosing $\epsilon_1 = 1/100, \epsilon = 1/20, \nu = 5L_0 + \epsilon_1 + \frac{2L_2}{L_0^2} + L_1$, and $\mu = L_1$, which satisfy

$$10\epsilon + 2\nu + (2+2\epsilon)\mu = 10\epsilon + 2\cdot\left(5L_0 + 1/100 + \frac{2L_2}{L_0^2} + L_1\right) + 3L_1 \le 1/2 + 1/50 + \left(10L_0 + \frac{4L_2}{L_0^2} + 5L_1\right) \le 1,$$

it holds that for the above choices of $\epsilon, \epsilon_1$,

$$(\mathbf{B}^\top \mathbf{A}_2^\top + \mathbf{A}_2 \mathbf{B}) + (\mathbf{A}_1 - \mathbf{A}_2 \mathbf{B})^\top (\mathbf{A}_1 - \mathbf{A}_2 \mathbf{B}) + (\mathbf{A}_2 \mathbf{B} - \mathbf{A}_1)^\top \mathbf{D} + \mathbf{D}^\top (\mathbf{A}_2 \mathbf{B} - \mathbf{A}_1) + (\mathbf{D}^\top + \mathbf{D}) + \mathbf{D}^\top \mathbf{D}$$

$$\preceq L_0/2 \cdot \left(12 + \frac{10}{\epsilon}\right)(\mathbf{A}_1^\top + \mathbf{A}_1) + \left(K + 18 + 4L_0 + \frac{1}{\epsilon_1} + \frac{14}{\epsilon} + 2\epsilon\right)\delta^2 I$$

$$\preceq 106 L_0 \cdot (\mathbf{A}_1^\top + \mathbf{A}_1) + (K + 400)\,\delta^2 I$$

$$\preceq \mathbf{A}_1^\top + \mathbf{A}_1 + (K + 400) \cdot \delta^2 I,$$

establishing (36). □

The next several lemmas ensure that the matrices $\mathbf{D}^{(t)}$ satisfy the conditions of the matrix $\mathbf{D}$ of Lemma 13. First, Lemma 14 shows that $\|F(\mathbf{z}^{(t)})\|$ only grows by a constant factor over the course of a constant number of time steps.

**Lemma 14.** *Suppose that for some $t \ge 1$, we have $\max\{\|F(\mathbf{z}^{(t)})\|, \|F(\mathbf{z}^{(t-1)})\|\} \le \delta$. Then for any $s \ge 1$, we have $\|F(\mathbf{z}^{(t+s)})\| \le \delta \cdot (1 + 3\eta\ell)^s$.*

*Proof.* We prove the claimed bound by induction. Since $F$ is $\ell$-Lipschitz, we get

$$\|F(\mathbf{z}^{(t+s)}) - F(\mathbf{z}^{(t+s-1)})\| \le 3\eta\ell \max\{\|F(\mathbf{z}^{(t+s-1)})\|, \|F(\mathbf{z}^{(t+s-2)})\|\}$$

for each $s \ge 1$, and so if $\delta_s := \max\{\|F(\mathbf{z}^{(t+s-1)})\|, \|F(\mathbf{z}^{(t+s-2)})\|\}$, the triangle inequality gives

$$\|F(\mathbf{z}^{(t+s)})\| \le \delta_s(1 + 3\eta\ell).$$

It follows by induction that $\|F(\mathbf{z}^{(t+s)})\| \le \delta \cdot (1 + 3\eta\ell)^s$.

□

Lemma 15 uses backwards induction (on $t$) to establish bounds on the matrices $\mathbf{C}^{(t)}$.

**Lemma 15** (Backwards induction lemma). *Suppose that there is some $L_0 > 0$ so that for all $t \leq T$, we have $\max\{\eta\|\mathbf{A}^{(t)}\|_\sigma, \eta\|\mathbf{B}^{(t)}\|_\sigma\} \leq L_0 \leq \sqrt{1/200}$ and $\eta\ell \leq 2/3$. Then:*

1. *$\|\mathbf{C}^{(t)}\|_\sigma \leq 2L_0^2$ for each $t \in [T]$.*

2. *The matrices $\mathbf{C}^{(t)}$ are well-defined, i.e., $I - \eta\mathbf{A}^{(t)} + \mathbf{C}^{(t)}$ is invertible for each $t \in [T]$, and the spectral norm of its inverse is bounded above by $\sqrt{2}$.*

3. *$\|\eta\mathbf{A}^{(t)} - \mathbf{C}^{(t)}\|_\sigma \leq 2L_0$ and $\|I - \eta\mathbf{A}^{(t)} + \mathbf{C}^{(t)}\|_\sigma \leq 1 + 2L_0$ for each $t \in [T]$.*

4. *For all $t < T$, it holds that*

$$(I - \eta\mathbf{A}^{(t+1)} + \mathbf{C}^{(t+1)})^{-1}(\eta\mathbf{A}^{(t+1)} - \mathbf{C}^{(t+1)})(\eta(\mathbf{A}^{(t+1)})^\top - (\mathbf{C}^{(t+1)})^\top)(I - \eta\mathbf{A}^{(t+1)} + \mathbf{C}^{(t+1)})^{-\top}$$
$$\preceq 3 \cdot \left( (\eta\mathbf{A}^{(t+1)})(\eta\mathbf{A}^{(t+1)})^\top + \mathbf{C}^{(t+1)}(\mathbf{C}^{(t+1)})^\top \right).$$

5. *Let $\delta^{(t)} := \max\{\|F(\mathbf{z}^{(t)})\|, \|F(\mathbf{z}^{(t-1)})\|\}$ for all $t \leq T$. For $t < T$, it holds that*

$$\mathbf{C}^{(t)}(\mathbf{C}^{(t)})^\top \preceq J_1 \cdot \eta\mathbf{A}^{(t)}(\eta\mathbf{A}^{(t)})^\top + J_2 \cdot (\delta^{(t)})^2 \cdot I,$$

*for $J_1 = 8L_0^2$ and $J_2 = 30L_0^2\eta^2(\eta\Lambda)^2$.*

*Proof.* The proof proceeds by backwards induction on $t$. The base case $t = T$ clearly holds since $\mathbf{C}^{(T)} = 0$. As for the inductive step, suppose that items 1 through 4 hold at time step $t$, for some $t \leq T$. Then by (28) and $L_0 \leq \frac{\sqrt{2}-1}{2}$,

$$\|\mathbf{C}^{(t-1)}\|_\sigma \leq L_0 \cdot (L_0 + \|\mathbf{C}^{(t)}\|_\sigma) \cdot \|(I - \eta\mathbf{A}^{(t)} + \mathbf{C}^{(t)})^{-1}\| \leq \sqrt{2}L_0 \cdot (L_0 + 2L_0^2) \leq 2L_0^2,$$

establishing item 1 at time $t - 1$.

Next, note that $\|\eta\mathbf{A}^{(t-1)} - \mathbf{C}^{(t-1)}\| \leq L_0 + 2L_0^2 \leq 2L_0$. Thus, by Equation (5.8.2) of [HJ12] and $L_0 \leq \frac{1}{2} - \frac{1}{2\sqrt{2}}$, it follows that

$$\|(I - \eta\mathbf{A}^{(t-1)} + \mathbf{C}^{(t-1)})^{-1}\|_\sigma \leq \frac{1}{1 - 2L_0} \leq \sqrt{2},$$

which establishes item 2 at time $t - 1$. It is also immediate that $\|I - \eta\mathbf{A}^{(t-1)} + \mathbf{C}^{(t-1)}\|_\sigma \leq 1 + 2L_0$, establishing item 3 at time $t - 1$.

Next we establish items 4 and 5 at time $t - 1$. First, we have

$$\|\mathbf{A}^{(t)} - \mathbf{A}^{(t-1)}\|_\sigma$$
$$\text{(Lemma 12)} \quad \leq \Lambda\|\mathbf{w}^{(t)} - \mathbf{w}^{(t-1)}\| + \frac{\eta\Lambda}{2}\|F(\mathbf{z}^{(t)}) - F(\mathbf{z}^{(t-1)})\|$$
$$\leq \eta\Lambda\|F(\mathbf{z}^{(t-1)})\| + \frac{\eta\Lambda}{2}\left( 2\eta\ell\|F(\mathbf{z}^{(t-1)})\| + \eta\ell\|F(\mathbf{z}^{(t-2)})\| \right)$$
$$\leq \delta^{(t-1)} \cdot 2\eta\Lambda, \tag{50}$$

where the final inequality uses $\eta\ell \leq 2/3$.

Next, by definition of $\mathbf{C}^{(t-1)}$ in (28),

$$\mathbf{C}^{(t-1)}(\mathbf{C}^{(t-1)})^\top$$

$$=\eta^2(I - \eta\mathbf{A}^{(t)} + \mathbf{C}^{(t)})^{-1}(\eta\mathbf{A}^{(t)} - \mathbf{C}^{(t)})\mathbf{B}^{(t)}(\mathbf{B}^{(t)})^\top(\eta(\mathbf{A}^{(t)})^\top - (\mathbf{C}^{(t)})^\top)(I - \eta\mathbf{A}^{(t)} + \mathbf{C}^{(t)})^{-\top}$$

$$\preceq L_0^2(I - \eta\mathbf{A}^{(t)} + \mathbf{C}^{(t)})^{-1}(\eta\mathbf{A}^{(t)} - \mathbf{C}^{(t)})(\eta(\mathbf{A}^{(t)})^\top - (\mathbf{C}^{(t)})^\top)(I - \eta\mathbf{A}^{(t)} + \mathbf{C}^{(t)})^{-\top} \tag{51}$$

$$\preceq \frac{L_0^2}{(1 - \|\eta\mathbf{A}^{(t)} - \mathbf{C}^{(t)}\|_\sigma)^2} \cdot (\eta\mathbf{A}^{(t)} - \mathbf{C}^{(t)})(\eta(\mathbf{A}^{(t)})^\top - (\mathbf{C}^{(t)})^\top) \tag{52}$$

$$\preceq \frac{2L_0^2}{(1 - 2L_0)^2} \cdot \left((\eta\mathbf{A}^{(t)})(\eta\mathbf{A}^{(t)})^\top + \mathbf{C}^{(t)}(\mathbf{C}^{(t)})^\top\right) \tag{53}$$

$$\preceq 3L_0^2 \cdot \left((\eta\mathbf{A}^{(t)})(\eta\mathbf{A}^{(t)})^\top + \mathbf{C}^{(t)}(\mathbf{C}^{(t)})^\top\right), \tag{54}$$

$$\preceq 3L_0^2 \cdot \left((\eta\mathbf{A}^{(t)})(\eta\mathbf{A}^{(t)})^\top \cdot (1 + J_1) + J_2 \cdot (\delta^{(t)})^2 \cdot I\right) \tag{55}$$

$$\preceq 6L_0^2(1 + J_1) \cdot (\eta\mathbf{A}^{(t-1)})(\eta\mathbf{A}^{(t-1)})^\top + 6L_0^2\eta^2(1 + J_1) \cdot \|\mathbf{A}^{(t-1)} - \mathbf{A}^{(t)}\|_\sigma^2 + 3L_0^2J_2 \cdot (\delta^{(t)})^2 \cdot I \tag{56}$$

$$\preceq 6L_0^2(1 + J_1) \cdot (\eta\mathbf{A}^{(t-1)})(\eta\mathbf{A}^{(t-1)})^\top + 24L_0^2\eta^2(1 + J_1)(\eta\Lambda)^2 \cdot (\delta^{(t-1)})^2 + 3L_0^2J_2 \cdot (\delta^{(t)})^2 \cdot I \tag{57}$$

$$\preceq 6L_0^2(1 + J_1) \cdot (\eta\mathbf{A}^{(t-1)})(\eta\mathbf{A}^{(t-1)})^\top + (\delta^{(t-1)})^2 \cdot \left(24L_0^2\eta^2(1 + J_1)(\eta\Lambda)^2 + 3L_0^2J_2(1 + 3\eta\ell)\right) \cdot I \tag{58}$$

where:

- (51) follows by Lemma 20;

- (52) is by Lemma 21 with $\mathbf{X} = \eta\mathbf{A}^{(t)} - \mathbf{C}^{(t)}$;

- (53) uses Lemma 17 and item 3 at time $t$;

- (54) follows from $L_0 \leq \frac{1 - \sqrt{2/3}}{2}$;

- (55) follows from the inductive hypothesis that item 5 holds at time $t$;

- (56) follows from Lemma 18;

- (57) follows from (50);

- (58) follows from the fact that $\delta^{(t)} \leq (1 + 3\eta\ell)\delta^{(t-1)}$, which is a consequence of Lemma 14.

Inequalities (51) through (54) establish item 4 at time $t - 1$. In order for item 5 to hold at time $t - 1$, we need that

$$6L_0^2(1 + J_1) \leq J_1 \tag{59}$$

$$24L_0^2\eta^2(1 + J_1)(\eta\Lambda)^2 + 3L_0^2J_2(1 + 3\eta\ell) \leq J_2. \tag{60}$$

By choosing $J_1 = 8L_0^2$ we satisfy (59) since $L_0 < \sqrt{1/24}$. By choosing $J_2 = 30L_0^2\eta^2(\eta\Lambda)^2$ we satisfy (60) since

$$24L_0^2\eta^2(1 + 8L_0^2)(\eta\Lambda)^2 + 3L_0^2 \cdot J_2(1 + 3\eta\ell) \leq 25L_0^2\eta^2(\eta\Lambda)^2 + 9L_0^2J_2 \leq J_2,$$

where we use $L_0 \leq \sqrt{1/192}$ and $\eta\ell \leq 2/3$. This completes the proof that item 5 holds at time $t - 1$. $\qquad\square$

**Lemma 16.** *Suppose that the pre-conditions of Lemma 15 (namely, those in its first sentence) hold. Then for each $t \in [T]$, we have*

$$\mathbf{D}^{(t)} + (\mathbf{D}^{(t)})^\top \preceq 6L_0\eta^2(\mathbf{B}^{(t)})^\top\mathbf{B}^{(t)} + 4L_0\eta^2\mathbf{A}^{(t)}(\mathbf{A}^{(t)})^\top + \left(4L_0 + \frac{1}{3L_0}\right)\mathbf{C}^{(t)}(\mathbf{C}^{(t)})^\top. \tag{61}$$

*and*

$$(\mathbf{D}^{(t)})^\top\mathbf{D}^{(t)} \preceq 60L_0^4\eta^2(\mathbf{B}^{(t)})^\top\mathbf{B}^{(t)}. \tag{62}$$

*Proof.* By Lemma 17, for any $\epsilon > 0$,

$$- \mathbf{C}^{(t)} \eta \mathbf{B}^{(t)} - \eta (\mathbf{B}^{(t)})^\top (\mathbf{C}^{(t)})^\top$$
$$\preceq \epsilon \cdot \eta^2 (\mathbf{B}^{(t)})^\top \mathbf{B}^{(t)} + \frac{1}{\epsilon} \cdot \mathbf{C}^{(t)} (\mathbf{C}^{(t)})^\top.$$

Also, for any $\epsilon > 0$,

$$(I - \eta \mathbf{A}^{(t)} + \mathbf{C}^{(t)})^{-1} (\eta \mathbf{A}^{(t)} - \mathbf{C}^{(t)})^2 \eta \mathbf{B}^{(t)} + \eta (\mathbf{B}^{(t)})^\top (\eta (\mathbf{A}^{(t)})^\top - (\mathbf{C}^{(t)})^\top)^2 (I - \eta \mathbf{A}^{(t)} + \mathbf{C}^{(t)})^{-\top}$$

$$\preceq \frac{1}{\epsilon} (I - \eta \mathbf{A}^{(t)} + \mathbf{C}^{(t)})^{-1} (\eta \mathbf{A}^{(t)} - \mathbf{C}^{(t)})^2 (\eta (\mathbf{A}^{(t)})^\top - (\mathbf{C}^{(t)})^\top)^2 (I - \eta \mathbf{A}^{(t)} + \mathbf{C}^{(t)})^{-\top} + \epsilon \eta^2 (\mathbf{B}^{(t)})^\top \mathbf{B}^{(t)} \tag{63}$$

$$\preceq \frac{4 L_0^2}{\epsilon} (I - \eta \mathbf{A}^{(t)} + \mathbf{C}^{(t)})^{-1} (\eta \mathbf{A}^{(t)} - \mathbf{C}^{(t)}) (\eta (\mathbf{A}^{(t)})^\top - (\mathbf{C}^{(t)})^\top) (I - \eta \mathbf{A}^{(t)} + \mathbf{C}^{(t)})^{-\top} + \epsilon \eta^2 (\mathbf{B}^{(t)})^\top \mathbf{B}^{(t)} \tag{64}$$

$$\preceq \frac{12 L_0^2}{\epsilon} \cdot \left( \eta^2 \mathbf{A}^{(t)} (\mathbf{A}^{(t)})^\top + \mathbf{C}^{(t)} (\mathbf{C}^{(t)})^\top \right) + \epsilon \eta^2 (\mathbf{B}^{(t)})^\top \mathbf{B}^{(t)}. \tag{65}$$

where (63) uses Lemma 17, (64) uses item 3 of Lemma 15 and Lemma 20, and (65) uses item 4 of Lemma 15.

Choosing $\epsilon = 3 L_0$ and using the definition of $\mathbf{D}^{(t)}$ in (33), it follows from the above displays that

$$\mathbf{D}^{(t)} + (\mathbf{D}^{(t)})^\top \preceq 6 L_0 \eta^2 (\mathbf{B}^{(t)})^\top \mathbf{B}^{(t)} + 4 L_0 \eta^2 \mathbf{A}^{(t)} (\mathbf{A}^{(t)})^\top + \left( 4 L_0 + \frac{1}{3 L_0} \right) \mathbf{C}^{(t)} (\mathbf{C}^{(t)})^\top,$$

which establishes (61).

To prove (62) we first note that

$$\left\| (I - \eta \mathbf{A}^{(t)} + \mathbf{C}^{(t)})^{-1} (\eta \mathbf{A}^{(t)} - \mathbf{C}^{(t)})^2 - \mathbf{C}^{(t)} \right\|_\sigma$$

(Lemma 15, item 2) $\quad \leq \sqrt{2} \| \eta \mathbf{A}^{(t)} - \mathbf{C}^{(t)} \|_\sigma^2 + \| \mathbf{C}^{(t)} \|_\sigma$

(Lemma 15, items 1 & 3) $\quad \leq \sqrt{2} \cdot 4 L_0^2 + 2 L_0^2 = (2 + 4\sqrt{2}) L_0^2.$

By Lemma 20, it follows that

$$(\mathbf{D}^{(t)})^\top \mathbf{D}^{(t)} \preceq 60 L_0^4 \cdot \eta^2 (\mathbf{B}^{(t)})^\top \mathbf{B}^{(t)},$$

establishing (62). $\qquad \square$

Finally we are ready to prove Theorem 5; for convenience we restate it here.

**Theorem 5** (restated). *Suppose $F : \mathbb{R}^n \to \mathbb{R}^n$ is a monotone operator that is $\ell$-Lipschitz and is such that $\partial F(\cdot)$ is $\Lambda$-Lipschitz. For some $\mathbf{z}^{(-1)}, \mathbf{z}^{(0)} \in \mathbb{R}^n$, suppose there is $\mathbf{z}^* \in \mathbb{R}^n$ so that $F_\mathcal{G}(\mathbf{z}^*) = 0$ and $\| \mathbf{z}^* - \mathbf{z}^{(-1)} \| \leq D, \| \mathbf{z}^* - \mathbf{z}^{(0)} \| \leq D$. Then the iterates $\mathbf{z}^{(T)}$ of the OG algorithm (3) for any $\eta \leq \min \left\{ \frac{1}{150\ell}, \frac{1}{1711 D \Lambda} \right\}$ satisfy:*

$$\| F_\mathcal{G}(\mathbf{z}^{(T)}) \| \leq \frac{60 D}{\eta \sqrt{T}} \tag{66}$$

*Proof of Theorem 5.* By Lemma 11 with $S = 3$, we have that for some $t^* \in \{0, 1, 2, \ldots, T\}$,

$$\max \{ \| F(\mathbf{z}^{(t^*)}) \|, \| F(\mathbf{z}^{(t^*-1)}) \|, \| F(\mathbf{z}^{(t^*-2)}) \| \} \leq \frac{6 \sqrt{3} D}{\eta \sqrt{T} \cdot \sqrt{1 - 10 \eta^2 \ell^2}} \leq \frac{12 D}{\eta \sqrt{T}} =: \delta_0. \tag{67}$$

Set $L_0 := \eta \ell \leq 1/150$ and $\Lambda_0 := \eta \Lambda$. By Lemma 12 we have that $\| \eta \mathbf{A}^{(t)} \|_\sigma \leq L_0$ and $\| \eta \mathbf{B}^{(t)} \|_\sigma \leq L_0$ for all $t \leq T$. Thus the preconditions of Lemma 15 hold, and in particular by item 1 of Lemma 15, it follows that

$$\| \tilde{F}^{(t^*)} \| = \| F(\mathbf{w}^{(t^*)}) + \mathbf{C}^{(t^*-1)} \cdot F(\mathbf{z}^{(t^*-1)}) \|$$
$$\leq \| F(\mathbf{w}^{(t^*)}) \| + 2 L_0^2 \| F(\mathbf{z}^{(t^*-1)}) \| \leq \delta_0 \cdot (1 + L_0 + 2 L_0^2) \leq \delta_0 \cdot (1 + 2 L_0). \tag{68}$$

Write $\delta := \delta_0(1 + 2L_0)$. By (30), we have that for any $t \in \{t^*, \ldots, T\}$,

$$\|\tilde{F}_t\|^2 \leq \prod_{t'=t^*}^{t-1} \|I - \eta \mathbf{A}^{(t')} + \mathbf{C}^{(t')}\|_\sigma^2 \cdot \delta^2. \tag{69}$$

We will prove by forwards induction (contrast with Lemma 15) that for each $t \in \{t^* - 1, \ldots, T\}$, the following hold:

1. $\|\tilde{F}^{(t+1)}\| \leq 2\delta$. (We will only need this item for $t^* - 1 \leq t \leq T - 1$.)

2. $\max\{\|F(\mathbf{z}^{(t)})\|, \|F(\mathbf{z}^{(t-1)})\|\} \leq 4\delta$.

3. $\|I - \eta \mathbf{A}^{(t)} + \mathbf{C}^{(t)}\|_\sigma^2 \leq 1 + 10025\Lambda_0^2 \eta^2 \delta^2$ if $t \geq t^*$.

The base case $t = t^* - 1$ is immediate: item 1 follows from (68), item 2 follows from (67), and item 3 states nothing for $t = t^* - 1$. We now assume that items 1 through 3 all hold for some value $t - 1 \geq t^* - 1$, and prove that they hold for $t$. We first establish that item 2 holds at time $t$, namely that $\|F(\mathbf{z}^{(t)})\| \leq 4\delta$. Since item 1 holds at time $t - 1$, we get that $\|\tilde{F}_t\| \leq 2\delta$, and so

$$\|F(\mathbf{w}^{(t)})\| = \|\tilde{F}_t - \mathbf{C}^{(t-1)}F(\mathbf{z}^{(t-1)})\| \leq \|\tilde{F}_t\| + 2L_0^2\|F(\mathbf{z}^{(t-1)})\| \leq 2\delta + 8L_0^2\delta,$$

which implies that

$$\|F(\mathbf{z}^{(t)})\| \leq \|F(\mathbf{w}^{(t)})\| + \eta\ell\|F(\mathbf{z}^{(t-1)})\| \leq 2\delta + 8L_0^2\delta + 4L_0\delta \leq 4\delta,$$

where the last inequality holds since $8L_0^2 + 4L_0 \leq 2$.

We proceed to the proof of item 1 at time $t$. By Lemma 12, we have that

$$\|\mathbf{B}^{(t)} - \mathbf{B}^{(t+1)}\|_\sigma \leq \Lambda\|\mathbf{w}^{(t)} - \mathbf{w}^{(t+1)}\| + \frac{\eta\Lambda}{2}\|F(\mathbf{z}^{(t-1)}) - F(\mathbf{z}^{(t)})\|$$

$$\leq \eta\Lambda\|F(\mathbf{z}^{(t)})\| + \frac{\eta\Lambda}{2}(\|\eta\ell F(\mathbf{z}^{(t-2)})\| + \|2\eta\ell F(\mathbf{z}^{(t-1)})\|)$$

$$\text{(item 2 at times } t, t-1) \qquad \leq 4\delta\Lambda_0 \cdot (1 + 3L_0/2) \leq 5\delta\Lambda_0 \tag{70}$$

$$\|\mathbf{B}^{(t)} - \mathbf{A}^{(t)}\|_\sigma \leq \frac{\eta\Lambda}{2}\|F(\mathbf{z}^{(t)}) - F(\mathbf{z}^{(t-1)})\|$$

$$\text{(item 2 at time } t-1) \qquad \leq 6\delta\Lambda_0 L_0 \tag{71}$$

$$\|\mathbf{B}^{(t+1)} - \mathbf{A}^{(t+1)}\|_\sigma \leq \frac{\eta\Lambda}{2}\|F(\mathbf{z}^{(t+1)}) - F(\mathbf{z}^{(t)})\|$$

$$\leq \frac{\Lambda_0}{2} \cdot (L_0\|F(\mathbf{z}^{(t)})\| + 2L_0\|F(\mathbf{z}^{(t+1)})\|)$$

$$\text{(item 2 at time } t \text{ \& Lemma 14)} \qquad \leq \frac{\Lambda_0}{2} \cdot (4\delta L_0 + 2L_0(1 + 3L_0) \cdot 4\delta)$$

$$\leq 8\Lambda_0 L_0\delta. \tag{72}$$

Recall that (35) gives

$$I - \eta\mathbf{A}^{(t)} + \mathbf{C}^{(t)} = I - \eta\mathbf{A}^{(t)} + \eta^2\mathbf{A}^{(t+1)}\mathbf{B}^{(t+1)} + \mathbf{D}^{(t+1)}.$$

Now we will apply Lemma 13 with $\mathbf{A}_1 = \eta\mathbf{A}^{(t)}, \mathbf{A}_2 = \eta\mathbf{A}^{(t+1)}, \mathbf{B} = \eta\mathbf{B}^{(t+1)}, \mathbf{D} = \mathbf{D}^{(t+1)}$, $L_0 = \eta\ell$ (which is called $L_0$ in the present proof as well). We check that all of the preconditions of the lemma hold:

- For $\mathbf{X} \in \{\eta\mathbf{A}^{(t)}, \eta\mathbf{A}^{(t+1)}, \eta\mathbf{B}^{(t+1)}\}$, $\mathbf{X} + \mathbf{X}^\top$ is PSD by Lemma 12.

- For $\mathbf{X} \in \{\eta\mathbf{A}^{(t)}, \eta\mathbf{A}^{(t+1)}, \eta\mathbf{B}^{(t+1)}\}$, $\|\mathbf{X}\|_\sigma \leq \eta\ell = L_0$ by Lemma 12, and we have $L_0 \leq 1/53$.[13]

- We may bound $\mathbf{D}^{(t+1)} + (\mathbf{D}^{(t+1)})^\top$ as follows:

$$\mathbf{D}^{(t+1)} + (\mathbf{D}^{(t+1)})^\top$$

$$\preceq 6L_0\eta^2(\mathbf{B}^{(t)})^\top\mathbf{B}^{(t)} + 4L_0\eta^2\mathbf{A}^{(t)}(\mathbf{A}^{(t)})^\top + \left(4L_0 + \frac{1}{3L_0}\right)\mathbf{C}^{(t)}(\mathbf{C}^{(t)})^\top \quad (73)$$

$$\preceq 6L_0\eta^2(\mathbf{B}^{(t)})^\top\mathbf{B}^{(t)} + 4L_0\eta^2\mathbf{A}^{(t)}(\mathbf{A}^{(t)})^\top$$
$$+ \left(\frac{1}{2L_0}\right)\cdot\left(8L_0^2\cdot\eta\mathbf{A}^{(t)}(\eta\mathbf{A}^{(t)})^\top + 30L_0^2\eta^4\Lambda^2(4\delta)^2\cdot I\right) \quad (74)$$

$$\preceq 6L_0\eta^2(\mathbf{B}^{(t)})^\top\mathbf{B}^{(t)} + 8L_0\eta^2\mathbf{A}^{(t)}(\mathbf{A}^{(t)})^\top + 240L_0\eta^2\Lambda_0^2\delta^2\cdot I$$

$$\preceq 12L_0\eta^2(\mathbf{B}^{(t+1)})^\top\mathbf{B}^{(t+1)} + 8L_0\eta^2\mathbf{A}^{(t)}(\mathbf{A}^{(t)})^\top + (300\delta^2\eta^2\Lambda_0^2 + 240L_0\eta^2\Lambda_0^2\delta^2)\cdot I. \quad (75)$$

$$\preceq 12L_0\eta^2(\mathbf{B}^{(t+1)})^\top\mathbf{B}^{(t+1)} + 8L_0\eta^2\mathbf{A}^{(t)}(\mathbf{A}^{(t)})^\top + 310\delta^2\eta^2\Lambda_0^2\cdot I.$$

where (73) follows from Lemma 16, (74) follows from item 5 of Lemma 15 and item 2 of the current induction at time $t$, and (75) follows from Lemma 18 and (70). This shows that in our application of Lemma 13 we may take $L_1 = 12L_0$. Moreover, as we will take the parameter $\delta$ in Lemma 13 to be $5\Lambda_0\eta\delta$ (see below items), we may take $K = 14L_0$ (since $14 \cdot (5\Lambda_0\eta\delta)^2 \geq 310\delta^2\eta^2\Lambda_0^2$).

- Lemma 16 gives

$$(\mathbf{D}^{(t+1)})^\top\mathbf{D}^{(t+1)} \preceq 60L_0^4\eta^2(\mathbf{B}^{(t+1)})^\top\mathbf{B}^{(t+1)},$$

so we may take $L_2 = 60L_0^4$ in our application of Lemma 13.

- We calculate that

$$12L_0 + \frac{4L_2}{L_0^2} + 5L_1 = 12L_0 + \frac{240L_0^4}{L_0^2} + 60L_0 = 72L_0 + 240L_0^2 \leq 1/2$$

holds as long as $L_0 \leq 1/150$.

- By (70), (71), and (72), we may take the parameter $\delta$ in Lemma 13 to be equal to $5\Lambda_0\eta\delta$ since $\max\{8\Lambda_0 L_0\delta, 4\delta\Lambda_0 + 12\delta\Lambda_0 L_0, 4\delta\Lambda_0 + 20\delta\Lambda_0 L_0\} \leq 5\Lambda_0\delta$.

By Lemma 13, it follows that

$$\|I - \eta\mathbf{A}^{(t)} + \mathbf{C}^{(t)}\|_\sigma^2 \leq 1 + 25\Lambda_0^2\eta^2\delta^2\cdot(400 + 14L_0) \leq 1 + 10025\Lambda_0^2\eta^2\delta^2,$$

which establishes that item 3 holds at time $t$.

Finally we show that item 1 holds at time $t$. To do so, we use (69) and the fact that $\delta^2 \leq \frac{146D^2}{\eta^2 T}$ to conclude that

$$\|\tilde{F}^{(t+1)}\|^2 \leq \delta^2\cdot\left(1 + 10025\Lambda_0^2\eta^2\delta^2\right)^T \leq \delta^2\cdot\left(1 + \frac{K_0\Lambda_0^2 D^2}{T}\right)^T \leq 4\delta^2,$$

where $K_0 = 10025\cdot 146$ and the last inequality holds as long as $K_0\Lambda_0^2 D^2 = K_0\eta^2\Lambda^2 D^2 \leq 1/2$, i.e., $\eta \leq \frac{1}{\sqrt{2K_0}\cdot\Lambda D}$; in particular, it suffices to take $\eta \leq \frac{1}{1711\cdot\Lambda D}$. This verifies that item 1 holds at time $t$, completing the inductive step.

The conclusion of Theorem 5 is an immediate conclusion of item 2 at time $T$, since $4\delta \leq 5\delta_0 = \frac{60D}{\eta\sqrt{T}}$. $\square$

## B.5 Helpful lemmas

**Lemma 17** (Young's inequality). *For square matrices $\mathbf{X}, \mathbf{Y}$, we have, for any $\epsilon > 0$,*

$$\mathbf{X}\mathbf{Y}^\top + \mathbf{Y}\mathbf{X}^\top \preceq \epsilon\mathbf{X}\mathbf{X}^\top + \frac{1}{\epsilon}\cdot\mathbf{Y}\mathbf{Y}^\top.$$

Applying the previous lemma to the cross terms in the quantity $\mathbf{X}\mathbf{X}^\top$ when using the decomposition $\mathbf{X} = \mathbf{Y} + (\mathbf{X} - \mathbf{Y})$, we obtain the following.

**Lemma 18.** *For square matrices* $\mathbf{X}, \mathbf{Y}$*, we have, for any* $\epsilon > 0$,

$$\mathbf{X}\mathbf{X}^\top \preceq (1+\epsilon) \cdot \mathbf{Y}\mathbf{Y}^\top + \left(1 + \frac{1}{\epsilon}\right) \|\mathbf{X} - \mathbf{Y}\|_\sigma^2 \cdot I.$$

*In particular, choosing* $\epsilon = 1$ *gives*

$$\mathbf{X}\mathbf{X}^\top \preceq 2\mathbf{Y}\mathbf{Y}^\top + 2\|\mathbf{X} - \mathbf{Y}\|_\sigma^2 \cdot I.$$

Lemma 19 is an immediate corollary of the two lemmas above:

**Lemma 19.** *For square matrices* $\mathbf{X}, \mathbf{Y}$*, we have*

$$\mathbf{X}\mathbf{Y}^\top + \mathbf{Y}\mathbf{X}^\top \preceq 3\mathbf{Y}\mathbf{Y}^\top + 2\|\mathbf{X} - \mathbf{Y}\|_\sigma^2 \cdot I.$$

**Lemma 20.** *For square matrices* $\mathbf{X}, \mathbf{Y}$ *such that* $\|\mathbf{Y}\|_\sigma \leq M$*, we have*

$$\mathbf{X}^\top \mathbf{Y}^\top \mathbf{Y}\mathbf{X} \preceq M^2 \mathbf{X}^\top \mathbf{X}.$$

*Proof.* For any $\mathbf{v}$, we have

$$\|\mathbf{Y}\mathbf{X}\mathbf{v}\|^2 \leq M^2 \|\mathbf{X}\mathbf{v}\|^2.$$

$\square$

**Lemma 21.** *For any square matrix* $\mathbf{X}$ *so that* $\|\mathbf{X}\|_\sigma < 1$*, we have*

$$(I - \mathbf{X})^{-1}\mathbf{X}\mathbf{X}^\top(I - \mathbf{X})^{-\top} \preceq \frac{1}{(1 - \|\mathbf{X}\|_\sigma)^2} \cdot \mathbf{X}\mathbf{X}^\top.$$

*Proof.* Using the equality (34), we have that for any $\epsilon > 0$,

$$\begin{aligned}
&(I - \mathbf{X})^{-1}\mathbf{X}\mathbf{X}^\top(I - \mathbf{X})^{-\top} \\
=&(\mathbf{X} + (I - \mathbf{X})^{-1}\mathbf{X}^2)(\mathbf{X}^\top + (\mathbf{X}^\top)^2(I - \mathbf{X})^{-\top}) \\
=&\mathbf{X}\mathbf{X}^\top + (I - \mathbf{X})^{-1}\mathbf{X}^2\mathbf{X}^\top + \mathbf{X}(\mathbf{X}^\top)^2(I - \mathbf{X})^{-\top} + (I - \mathbf{X})^{-1}\mathbf{X}^2(\mathbf{X}^\top)^2(I - \mathbf{X})^{-\top}
\end{aligned}$$

(Lemma 17) $\quad \preceq (1 + 1/\epsilon)\mathbf{X}\mathbf{X}^\top + (1 + \epsilon)(I - \mathbf{X})^{-1}\mathbf{X}^2(\mathbf{X}^\top)^2(I - \mathbf{X})^{-\top}$

(Lemma 20) $\quad \preceq (1 + 1/\epsilon)\mathbf{X}\mathbf{X}^\top + (1 + \epsilon)\|\mathbf{X}\|_\sigma^2 \cdot (I - \mathbf{X})^{-1}\mathbf{X}\mathbf{X}^\top(I - \mathbf{X})^{-\top}.$

Rearranging gives

$$(I - \mathbf{X})^{-1}\mathbf{X}\mathbf{X}^\top(I - \mathbf{X})^{-\top} \preceq \min_{\epsilon > 0 : (1+\epsilon)\|\mathbf{X}\|_\sigma^2 < 1} \frac{(1 + 1/\epsilon)\mathbf{X}\mathbf{X}^\top}{1 - (1 + \epsilon)\|\mathbf{X}\|_\sigma^2}$$

Choosing $\epsilon = \frac{1 - \|\mathbf{X}\|_\sigma}{\|\mathbf{X}\|_\sigma}$ gives the desired conclusion. $\square$

## C  Proofs for Section 4

In this section we prove Theorem 7, and as byproducts of our analysis additionally prove the results mentioned at the end of Section 4.

Recall from Section 4 that $\mathcal{F}_{n,\ell,D}^{\mathrm{bil}}$ is defined to be the set of $\ell$-Lipschitz operators $F : \mathbb{R}^n \to \mathbb{R}^n$ of the form

$$F(\mathbf{z}) = \mathbf{A}\mathbf{z} + \mathbf{b} \quad \text{where} \quad \mathbf{z} = \begin{pmatrix} \mathbf{x} \\ \mathbf{y} \end{pmatrix}, \mathbf{A} = \begin{pmatrix} \mathbf{0} & \mathbf{M} \\ -\mathbf{M}^\top & \mathbf{0} \end{pmatrix}, \mathbf{b} = \begin{pmatrix} \mathbf{b}_1 \\ -\mathbf{b}_2 \end{pmatrix}, \tag{76}$$

for which $\mathbf{A}$ is of full rank and $-\mathbf{A}^{-1}\mathbf{b} \in \mathcal{D}_D := \mathcal{B}_{\mathbb{R}^{n/2}}(\mathbf{0}, D) \times \mathcal{B}_{\mathbb{R}^{n/2}}(\mathbf{0}, D)$. Note that each $F \in \mathcal{F}_{n,\ell,D}^{\mathrm{bil}}$ can be written as the min-max gradient operator $F(\mathbf{x}, \mathbf{y}) = (\nabla_\mathbf{x} f(\mathbf{x}, \mathbf{y})^\top, -\nabla_\mathbf{y} f(\mathbf{x}, \mathbf{y})^\top)^\top$ corresponding to the function

$$f(\mathbf{x}, \mathbf{y}) = \mathbf{x}^\top \mathbf{M}\mathbf{y} + \mathbf{b}_1^\top \mathbf{x} + \mathbf{b}_2^\top \mathbf{y}. \tag{77}$$

We next note that when $F \in \mathcal{F}_{n,\ell,D}^{\mathrm{bil}}$, the $p$-SCLI updates of Definition 5 can be rewritten as follows:

**Observation 22.** *Suppose that $\mathcal{A}$ is a p-SCLI. Then there are constants $\alpha_j, \beta_j, \gamma, \delta \in \mathbb{R}$, $0 \le j \le p-1$, depending only on $\mathcal{A}$, so that for an instance $F$ of the form $F(\mathbf{z}) = \mathbf{A}\mathbf{z} + \mathbf{b}$, and an arbitrary set of p initialization poitns $\mathbf{z}^{(0)}, \ldots, \mathbf{z}^{(-p+1)} \in \mathbb{R}^n$, the iterates $\mathbf{z}^{(t)}$ of $\mathcal{A}$ satisfy*

$$\mathbf{z}^{(t)} = \sum_{j=0}^{p-1} \mathbf{C}_j(\mathbf{A})\mathbf{z}^{(t-p+j)} + \mathbf{N}(\mathbf{A})\mathbf{b}, \tag{78}$$

*for $t \ge 1$ and $\mathbf{C}_j(\mathbf{A}) = \alpha_j \mathbf{A} + \beta_j I_n$ for $0 \le j \le p-1$ and $\mathbf{N}(\mathbf{A}) = \gamma\mathbf{A} + \delta I_n$.*

In the case of OG with a constant step size $\eta$, for $F(\mathbf{z}) = \mathbf{A}\mathbf{z} + \mathbf{b}$, we may rewrite (15) as

$$\mathbf{z}^{(t)} = (I - 2\eta A)\mathbf{z}^{(t-1)} + (\eta A)\mathbf{z}^{(t-2)} - \eta b,$$

so we have $\mathbf{C}_0(\mathbf{A}) = I_n - 2\eta\mathbf{A}, \mathbf{C}_1(\mathbf{A}) = \eta\mathbf{A}, \mathbf{N}(\mathbf{A}) = -\eta I_n$.

All lower bounds we prove in this section will apply more generally to any iterative algorithm $\mathcal{A}$ whose updates are of the form (78) when restricted to instances $F(\mathbf{z}) = \mathbf{A}\mathbf{z} + \mathbf{b}$.

The remainder of this section is organized as follows. In Section C.1, we prove Theorem 7. In Section C.2 we prove Proposition 8, which is used in the proof of Theorem 7, and Proposition 9, showing that Proposition 8 is tight in a certain sense. In Section C.3 we prove a conjecture of [ASSS15], which is similar in spirit to Proposition 8 and leads to an algorithm-independent version of Theorem 7 (with a weaker quantitative bound). Finally, in Section C.4, we discuss another byproduct of our analysis, namely a lower bound for $p$-SCLIs for convex function minimization.

## C.1    $p$-SCLI lower bounds for the class $\mathcal{F}_{n,\ell,D}^{\mathrm{bil}}$

**Notation.** For a square matrix $\mathbf{A}$, let $\rho(\mathbf{A})$ be its spectral radius, i.e., the maximum magnitude of an eigenvalue of $\mathbf{A}$. For matrices $\mathbf{A}_1 \in \mathbb{R}^{n_1 \times m_1}, \mathbf{A}_2 \in \mathbb{R}^{n_2 \times m_2}$, let $\mathbf{A}_1 \otimes \mathbf{A}_2 \in \mathbb{R}^{(n_1 n_2) \times (m_1 m_2)}$ be the tensor product (also known as Kronecker product) of $\mathbf{A}_1, \mathbf{A}_2$.

We will need the following standard lemma:

**Lemma 23.** *For a square matrix $\mathbf{C}$ and all $k \in \mathbb{N}$, we have $\|\mathbf{C}^k\|_\sigma \ge \rho(\mathbf{C})^k$.*

Next we prove Theorem 7, restated below for convenience.

**Theorem 7** (restated). *Fix $\ell, D > 0$, let $\mathcal{A}$ be a p-SCLI[14], and let $\mathbf{z}^{(t)}$ denote the tth iterate of $\mathcal{A}$. Then there are constants $c_\mathcal{A}, T_\mathcal{A} > 0$ so that the following holds: For all $T \ge T_\mathcal{A}$, there is some $F \in \mathcal{F}_{n,\ell,D}^{\mathrm{bil}}$ so that for some initialization $\mathbf{z}^{(0)}, \ldots, \mathbf{z}^{(-p+1)} \in \mathcal{D}_D$ and some $T' \in \{T, T+1, \ldots, T+p-1\}$, it holds that $\mathrm{TGap}_F^{\mathcal{D}_{2D}}(\mathbf{z}^{(T')}) \ge \frac{c_\mathcal{A}\ell D^2}{\sqrt{T}}$.*

*Proof of Theorem 7.* Take $F(\mathbf{z}) = \mathbf{A}\mathbf{z} + \mathbf{b}$, where $\mathbf{A}, \mathbf{b}$ are of the form shown in (76), with $\mathbf{M} = \nu \cdot I$ for some $\nu \in (0, \ell]$. Notice that $\mathbf{A}$ therefore depends on the choice of $\nu$ (which will be specified later), but for simplicity of notation we do not explicitly write this dependence. The outline of the proof is to first eliminate some corner cases in which the iterates of $\mathcal{A}$ do not converge and then reduce the statement of Theorem 7 to that of Proposition 8. There are a few different ways to carry out this reduction: we follow the linear algebraic approach of [ASSS15], but an approach of a different flavor using elementary ideas from complex analysis is given in [Nev93, Section 3.7].

Since $F \in \mathcal{F}_{n,\ell,D}^{\mathrm{bil}}$, we have that the equilibrium $\mathbf{z}^* = (\mathbf{x}^*, \mathbf{y}^*) \in \mathcal{D}_D$ satisfies $\mathcal{B}_{\mathbb{R}^{n/2}}(\mathbf{x}^*, D) \times \mathcal{B}_{\mathbb{R}^{n/2}}(\mathbf{y}^*, D) \subset \mathcal{D}_{2D}$. Then, from [GPDO20, Eq. (22)], it follows that $\mathrm{TGap}_F^{\mathcal{D}_{2D}}(\mathbf{z}) \ge D\|F(\mathbf{z})\|$ for any $\mathbf{z} \in \mathbb{R}^n$. Therefore, to prove Theorem 7 it suffices to show the lower bound $\|F(\mathbf{z}^{(T')})\| \ge \frac{c_\mathcal{A}\ell D}{\sqrt{T}}$.

We consider the dynamics of the iterates of $\mathcal{A}$ for various choices of $\mathbf{z}^{(0)}, \ldots, \mathbf{z}^{(-p+1)} \in \mathcal{D}_D$. To do so, we define the block matrices:

$$\mathbf{C}(\mathbf{A}) := \begin{pmatrix} \mathbf{0} & I_n & \mathbf{0} & \cdots & \mathbf{0} \\ \mathbf{0} & \mathbf{0} & I_n & \mathbf{0} & \cdots \\ \vdots & \vdots & \ddots & \ddots & \vdots \\ \vdots & \vdots & \ddots & \mathbf{0} & I_n \\ \mathbf{C}_0(\mathbf{A}) & \mathbf{C}_1(\mathbf{A}) & \cdots & \mathbf{C}_{p-2}(\mathbf{A}) & \mathbf{C}_{p-1}(\mathbf{A}) \end{pmatrix}, \qquad \mathbf{U} := \begin{pmatrix} \mathbf{0} \\ \vdots \\ \mathbf{0} \\ I_n \end{pmatrix} \in \mathbb{R}^{pn \times n},$$

(79)

and the block vectors

$$\mathbf{w}^{(t)} := \begin{pmatrix} \mathbf{z}^{(t-p+1)} \\ \mathbf{z}^{(t-p+2)} \\ \vdots \\ \mathbf{z}^{(t)} \end{pmatrix}.$$

Then the updates of $\mathcal{A}$ as in (78) can be written in the following form, for $F(\mathbf{z}) = \mathbf{Az} + \mathbf{b}$:

$$\mathbf{w}^{(t+1)} = \mathbf{C}(\mathbf{A})\mathbf{w}^{(t)} + \mathbf{UN}(\mathbf{A})\mathbf{b}.$$

Hence

$$\mathbf{w}^{(t)} = \mathbf{C}(\mathbf{A})^t \cdot \mathbf{w}^{(0)} + \sum_{s=1}^{t} \mathbf{C}(\mathbf{A})^{t-s} \mathbf{UN}(\mathbf{A})\mathbf{b}.$$

(80)

Recall that Observation 22 gives us $\mathbf{C}_j(\mathbf{A}) = \alpha_j \cdot \mathbf{A} + \beta_j \cdot I_n$, and $\mathbf{N}(\mathbf{A}) = \gamma \cdot \mathbf{A} + \delta \cdot I_n$, for some real numbers $\alpha_j, \beta_j, \gamma, \delta$ where $0 \le j \le p - 1$.

We now consider several cases:

**Case 1:** $\mathbf{C}(\mathbf{A}) - I_{np}$ or $\mathbf{N}(\mathbf{A})$ is not invertible for some choice of $\nu \in (0, \ell]$ (which determines $\mathbf{A}$ as explained above). First suppose that $\mathbf{C}(\mathbf{A}) - I_{np}$ is not invertible. Note that the row-space of $\mathbf{C}(\mathbf{A}) - I_{np}$ contains the row-space of the following matrix:

$$\tilde{\mathbf{C}} := \begin{pmatrix} -I_n & I_n & \mathbf{0} & \cdots \\ -I_n & \mathbf{0} & I_n & \cdots \\ \vdots & \ddots & \ddots & \vdots \\ -I_n & \mathbf{0} & \cdots & I_n \\ -I_n + \mathbf{C}_0(\mathbf{A}) & \mathbf{C}_1(\mathbf{A}) & \cdots & \mathbf{C}_{p-1}(\mathbf{A}) \end{pmatrix}.$$

If $\mathbf{C}_0(\mathbf{A}) + \cdots + \mathbf{C}_{p-1}(\mathbf{A}) - I_n$ is full-rank, then the row-space of $\tilde{\mathbf{C}}$ additionally contains the row-space of $(I_n \ \ \mathbf{0} \ \ \cdots \ \ \mathbf{0}) \in \mathbb{R}^{n \times np}$, and thus $\tilde{\mathbf{C}}$, and so $\mathbf{C}(\mathbf{A}) - I$ would be full-rank. Thus $\mathbf{C}_0(\mathbf{A}) + \cdots + \mathbf{C}_{p-1}(\mathbf{A}) - I_n$ is not full-rank. But we can write:

$$-I_n + \sum_{j=0}^{p-1} \mathbf{C}_j(\mathbf{A}) = \left( -1 + \sum_{j=0}^{p-1} \beta_j \right) I_n + \left( \sum_{j=0}^{p-1} \alpha_j \right) \cdot \mathbf{A} = \begin{pmatrix} \left( -1 + \sum_{j=0}^{p-1} \beta_j \right) I_{n/2} & \sum_{j=0}^{p-1} \alpha_j \mathbf{M} \\ -\sum_{j=0}^{p-1} \alpha_j \mathbf{M} & \left( -1 + \sum_{j=0}^{p-1} \beta_j \right) I_{n/2} \end{pmatrix}$$

But since $\mathbf{M}$ is a nonzero multiple of the identity matrix, if the above matrix is not full-rank, it must be identically 0, i.e., $\sum_{j=0}^{p-1} \mathbf{C}_j(\mathbf{A}) = I_n$. Hence $\sum_{j=0}^{p-1} \beta_j = 1$, $\sum_{j=0}^{p-1} \alpha_j = 0$.

Thus, for *any* choice of the matrix $\mathbf{A} \in \mathbb{R}^{n \times n}$, if we choose $\mathbf{b} = \mathbf{0}$ (so that $F(\mathbf{z}) = \mathbf{Az}$), and if $\mathbf{z}^{(0)} = \cdots = \mathbf{z}^{(-p+1)} = \mathbf{z}$ for some $\mathbf{z} \in \mathbb{R}^n$, it holds that for all $t \ge 1$, the iterates $\mathbf{z}^{(t)}$ of $\mathcal{Z}$ satisfy $\mathbf{z}^{(t)} = \mathbf{z}$. We now choose $\mathbf{M} = \ell \cdot I_{n/2}$ and $\mathbf{z} = \mathbf{z}^{(0)} = D/\sqrt{n/2} \cdot \mathbf{1} \in \mathbb{R}^n$, so that $\mathbf{z}^{(0)} - \mathbf{A}^{-1}\mathbf{b} = \mathbf{z}^{(0)} \in \mathcal{D}_D$. Then for all $t \ge 0$,

$$\|F(\mathbf{z}^{(t)})\|^2 = \|F(\mathbf{z}^{(0)})\|^2 = 2\ell^2 D^2.$$

Similarly, if $\mathbf{N}(\mathbf{A})$ is not invertible for some choice of $\nu \in (0, \ell]$, then by choice of $\mathbf{A}$ we must have that $\gamma = \delta = 0$, i.e., $\mathbf{N}(\mathbf{A}) = \mathbf{0}$ for all choices of $\nu$. Thus, choosing $\mathbf{w}^{(0)} = \mathbf{0}$ and $\mathbf{b} = D/\sqrt{n/2} \cdot \mathbf{1} \in \mathcal{D}_D$, and so for all $t \ge 0$, $\|F(\mathbf{z}^{(t)})\| = \|F(\mathbf{z}^{(0)})\| = \|\mathbf{b}\| = \sqrt{2}D$. Thus in this case we get the lower bound for $T \ge T_{\mathcal{A}} := \ell^2$.

**Cases 2 & 3.** In the remaining cases $\mathbf{C}(\mathbf{A}) - I_{np}$ and $\mathbf{N}(\mathbf{A})$ are invertible for all $\nu \in (0, \ell]$. Hence we can rewrite (80) as:

$$\mathbf{w}^{(t)} = \mathbf{C}(\mathbf{A})^t \cdot \mathbf{w}^{(0)} + (\mathbf{C}(\mathbf{A}) - I_{np})^{-1}(\mathbf{C}(\mathbf{A})^t - I_{np})\mathbf{U}\mathbf{N}(\mathbf{A})\mathbf{b}. \tag{81}$$

We consider three further sub-cases:

**Case 2.** $\rho(\mathbf{C}(\mathbf{A})) \geq 1$ for some $\nu \in (0, \ell]$. Fix such a $\nu$ (and thus $\mathbf{A}$). Since $\mathbf{C}(\mathbf{A})$ is invertible, we must in fact have $\rho(\mathbf{C}(\mathbf{A})) > 1$; write $\rho_0 := \rho(\mathbf{C}(\mathbf{A}))$. Again we choose $\mathbf{b} = \mathbf{0}$, so that $\mathbf{w}^{(t)} = \mathbf{C}(\mathbf{A})^t \cdot \mathbf{w}^{(0)}$, and so $(I_p \otimes \mathbf{A})\mathbf{w}^{(t)} = \mathbf{C}(\mathbf{A})^t \cdot (I_p \otimes \mathbf{A})\mathbf{w}^{(0)}$. By Lemma 23 we have that $\|\mathbf{C}(\mathbf{A})^t\|_\sigma \geq \rho_0^t$. Let $\tilde{\mathbf{w}}^{(0)} := ((\tilde{\mathbf{z}}^{(-p+1)})^\top, \dots, (\tilde{\mathbf{z}}^{(0)})^\top)^\top$ be a singular vector of $\mathbf{C}(\mathbf{A})^t$ corresponding to a singular value which is at least $\rho_0^t$. By appropriately scaling $\tilde{\mathbf{w}}^{(0)}$, we may ensure that $\tilde{\mathbf{z}}^{(-p+1)}, \dots, \tilde{\mathbf{z}}^{(0)} \in \mathcal{D}_D$ and $\|\tilde{\mathbf{w}}^{(0)}\| \geq D$. Moreover, we have that $\|(I_p \otimes \mathbf{A})\mathbf{w}^{(t)}\| = \nu\|\mathbf{w}^{(t)}\| \geq \nu\rho_0^t D$. This quantity can be made arbitrarily large by taking $t$ to be arbitrarily large (as $\rho_0 > 1$), and thus in this case $\|F(\mathbf{z}^{(t)})\| = \|\mathbf{A}\mathbf{z}^{(t)}\|$ fails to converge to 0 since $\|(I_p \otimes \mathbf{A})\mathbf{w}^{(t)}\| \to \infty$ as $t \to \infty$.

**Case 3.** $\rho(\mathbf{C}(\mathbf{A})) < 1$; in this case we have

$$\lim_{t \to \infty} \mathbf{U}^\top \mathbf{w}^{(t)} = -\mathbf{U}^\top(\mathbf{C}(\mathbf{A}) - I_{np})^{-1}\mathbf{U}\mathbf{N}(\mathbf{A})\mathbf{b}.$$

Note that $\mathbf{U}^\top(\mathbf{C}(\mathbf{A}) - I_{np})^{-1}\mathbf{U}$ is the lower $n \times n$-submatrix of the matrix $(\mathbf{C}(\mathbf{A}) - I_{np})^{-1}$, and therefore it must be the inverse of the Schur complement of the upper $(p-1)n \times (p-1)n$-submatrix of $\mathbf{C}(\mathbf{A}) - I_{np}$. Thus $\mathbf{U}^\top(\mathbf{C}(\mathbf{A}) - I_{np})^{-1}\mathbf{U}$ is invertible, and since $\mathbf{N}(\mathbf{A})$ is as well, we may define $\mathbf{B}(\mathbf{A}) := -\left(\mathbf{U}^\top(\mathbf{C}(\mathbf{A}) - I_{np})^{-1}\mathbf{U}\mathbf{N}(\mathbf{A})\right)^{-1}$. Hence $\mathbf{U}^\top(\mathbf{C}(\mathbf{A}) - I_{np})^{-1}\mathbf{U} = -\mathbf{B}(\mathbf{A})^{-1}\mathbf{N}(\mathbf{A})^{-1}$. As shown in [ASSS15, Eqs. (68) – (70)], this implies that $\sum_{j=0}^{p-1} \mathbf{C}_j(\mathbf{A}) = I_n + \mathbf{N}(\mathbf{A})\mathbf{B}(\mathbf{A})$, which can be written as:

$$\left(\sum_{j=0}^{p-1} \alpha_j\right)\mathbf{A} + \left(\sum_{j=0}^{p-1} \beta_j\right)I_n = I + (\gamma\mathbf{A} + \delta I_n) \cdot \mathbf{B}(\mathbf{A}). \tag{82}$$

Let $\mathbf{1}_p \in \mathbb{R}^p$ be the $p$-vector of ones. The fact that $\mathbf{N}(\mathbf{A})\mathbf{B}(\mathbf{A}) = \sum_{j=0}^{p-1} \mathbf{C}_j(\mathbf{A}) - I_n$ and definition of $\mathbf{U}$ gives

$$\mathbf{U}\mathbf{N}(\mathbf{A})\mathbf{B}(\mathbf{A}) = (\mathbf{C}(\mathbf{A}) - I_{pn})\begin{pmatrix} I_n \\ \vdots \\ I_n \end{pmatrix} = (\mathbf{C}(\mathbf{A}) - I_{pn})(\mathbf{1}_p \otimes I_n) \;\Rightarrow\; (\mathbf{C}(\mathbf{A}) - I_{pn})^{-1}\mathbf{U}\mathbf{N}(\mathbf{A})\mathbf{B}(\mathbf{A}) = \mathbf{1}_p \otimes I_n.$$

It then follows from (81) and the fact that $\mathbf{N}(\mathbf{A}), \mathbf{C}(\mathbf{A})$ commute with $\mathbf{A}$ that

$$(I_p \otimes \mathbf{A})\mathbf{w}^{(t)} + (\mathbf{1}_p \otimes \mathbf{b})$$
$$= (I_p \otimes \mathbf{A})\mathbf{C}(\mathbf{A})^t\mathbf{w}^{(0)} + (I_p \otimes \mathbf{A})(\mathbf{C}(\mathbf{A})^t - I_{np})(\mathbf{C}(\mathbf{A}) - I_{pn})^{-1}\mathbf{U}\mathbf{N}(\mathbf{A})\mathbf{B}(\mathbf{A})\mathbf{B}(\mathbf{A})^{-1}\mathbf{b} + (\mathbf{1}_p \otimes \mathbf{b})$$
$$= (I_p \otimes \mathbf{A})\mathbf{C}(\mathbf{A})^t\mathbf{w}^{(0)} + (\mathbf{C}(\mathbf{A})^t - I_{np})(I_p \otimes \mathbf{A})(\mathbf{1}_p \otimes \mathbf{B}(\mathbf{A})^{-1}\mathbf{b}) + (\mathbf{1}_p \otimes \mathbf{b})$$
$$= (I_p \otimes \mathbf{A})\mathbf{C}(\mathbf{A})^t\mathbf{w}^{(0)} + \mathbf{C}(\mathbf{A})^t(\mathbf{1}_p \otimes \mathbf{A}\mathbf{B}(\mathbf{A})^{-1}\mathbf{b}) + \mathbf{1}_p \otimes (I_n - \mathbf{A}\mathbf{B}(\mathbf{A})^{-1})\mathbf{b}. \tag{83}$$

**Case 3a.** $\sum_{j=0}^{p-1} \beta_j \neq 1$. Taking $\nu \to 0$ (i.e., $\mathbf{A} \to \mathbf{0}$) in (82), we see that $\delta \neq 0$, and moreover $\lim_{\mathbf{A} \to \mathbf{0}} \mathbf{B}(\mathbf{A}) = \delta^{-1}(\sum_{j=0}^{p-1} \beta_j - 1)I_n \neq \mathbf{0}$. Thus, there must be some $\nu_0 \in (0, \ell]$ so that $\mathbf{B}(\mathbf{A}) \neq \mathbf{A}$, and so for this choice of $\nu = \nu_0$, by (83), for an arbitrary choice of $\mathbf{w}^{(0)}$ and for some choice of $\mathbf{b}$ not in the nullspace of $I_n - \mathbf{A}\mathbf{B}(\mathbf{A})^{-1}$ with $\|\mathbf{b}\| = \nu_0 D/\sqrt{n/2} \cdot \mathbf{1}$, the following holds: for some constants $T_0 \in \mathbb{N}, c_0 > 0$, for all $t \geq T_0$, we have $\|(I_p \otimes \mathbf{A})\mathbf{w}^{(t)} + (\mathbf{1}_p \otimes \mathbf{b})\| \geq c_0$. This suffices to prove the desired lower bound on $\|F(\mathbf{z}^{(t)})\|$ (in particular, the constant $T_0$ determines $T_\mathcal{A}$ in the theorem statement).

**Case 3b.** $\sum_{j=0}^{p-1} \beta_j = 1$. This case contains the case in which the iterates $\mathbf{z}^{(t)}$ of the $p$-SCLI converge to the true solution $-\mathbf{A}^{-1}\mathbf{b}$ for all $\mathbf{A}, \mathbf{b}$, and is thus the main nontrivial case (in particular, it is the case in which we use Proposition 8).

We now choose $\mathbf{b} = \mathbf{0} \in \mathbb{R}^n$, and so $(I \otimes \mathbf{A})\mathbf{w}^{(t)} = \mathbf{C}(\mathbf{A})^t \mathbf{A}\mathbf{w}^{(0)}$ (we use here that $\mathbf{C}_j(\mathbf{A})$ all commute with $\mathbf{A}$). [ASSS15, Lemma 14] gives that the characteristic polynomial of $\mathbf{C}(\mathbf{A})$ is given by

$$\chi_{\mathbf{C}(\mathbf{A})}(\lambda) = (-1)^{pn} \det \left( \lambda^p I_n - \sum_{j=0}^{p-1} \lambda^j \mathbf{C}_j(\mathbf{A}) \right).$$

Recall that the assumption of linear coefficient matrices gives us that $\mathbf{C}_j(\mathbf{A}) = \alpha_j \cdot \mathbf{A} + \beta_j \cdot I_n$, where $\mathbf{A}$ is defined as in (76), depending on some matrix $\mathbf{M}$. Recall our choice of $\mathbf{M} = \nu \cdot I_{n/2}$, for some $\nu \in (0, \ell)$, to be specified below. Now define $q(\lambda) := \lambda^p - \sum_{j=0}^{p-1} \beta_j \lambda^j$ and $r(\lambda) := \sum_{j=0}^{p-1} \alpha_j \lambda^j$. Then

$$\lambda^p I_n - \sum_{j=0}^{p-1} \lambda^j \mathbf{C}_j(\mathbf{A}) = q(\lambda) \cdot I_n - r(\lambda) \cdot \mathbf{A} = \begin{pmatrix} q(\lambda) \cdot I_{n/2} & \nu r(\lambda) \cdot I_{n/2} \\ -\nu r(\lambda) \cdot I_{n/2} & q(\lambda) \cdot I_{n/2} \end{pmatrix} = \begin{pmatrix} q(\lambda) & \nu r(\lambda) \\ -\nu r(\lambda) & q(\lambda) \end{pmatrix} \otimes I_{n/2}.$$

By the formula for the determinant of a tensor product of matrices,

$$\chi_{\mathbf{C}(\mathbf{A})}(\lambda) = (-1)^{pn} \cdot (q(\lambda)^2 + \nu^2 r(\lambda)^2)^{n/2},$$

and so the spectral radius of $\mathbf{C}(\mathbf{A})$ is given by $\rho(\mathbf{C}(\mathbf{A})) = \rho(q(\lambda)^2 + \nu^2 r(\lambda)^2)$. Since $\sum_{j=0}^{p-1} \beta_j = 1$, we have that $q(1)^2 = 0$; moreover, $\lambda \mapsto q(\lambda)^2$ is a degree-$2p$ monic polynomial, while $\lambda \mapsto -r(\lambda)^2$ is a degree-$(2(p-1))$ (and thus also degree-$(2p-1)$) polynomial. Thus, by Proposition 8, we get that there are some constants $\mu_{\mathcal{A}}, C_{\mathcal{A}} > 0$ (depending on the algorithm $\mathcal{A}$) so that for any $\mu \in (0, \mu_{\mathcal{A}})$, there is some $\nu \in [\mu, \ell]$ so that $\rho(q(\lambda)^2 + \nu^2 r(\lambda)^2) \geq 1 - C_{\mathcal{A}} \cdot \mu^2/\ell^2$. Let $T_{\mathcal{A}}$ be so that $\ell/(2\sqrt{T_{\mathcal{A}}}) < \mu_{\mathcal{A}}$. Now for any $T \geq T_{\mathcal{A}}$, we may choose $\mu = \ell/(2\sqrt{T})$, and set $\nu \in [\ell/(2\sqrt{T}), \ell]$ accordingly per Proposition 8. By Lemma 23, we have that, for $T \geq T_{\mathcal{A}}$,

$$\|\mathbf{C}(\mathbf{A})^T\|_\sigma \geq \rho(\mathbf{C}(\mathbf{A}))^T \geq (1 - C_{\mathcal{A}}/(4T))^T \geq \exp(-C_{\mathcal{A}}).$$

Set $c_{\mathcal{A}} = \exp(-C_{\mathcal{A}})$. Choose $\mathbf{w}^{(0)} = ((\mathbf{z}^{(-p+1)})^\top, \dots, (\mathbf{z}^{(0)})^\top)^\top \in \mathbb{R}^{np}$ so that it is a (right) singular vector of $\mathbf{C}(\mathbf{A})^T$ corresponding to a singular value of magnitude at least $c_{\mathcal{A}}$. By scaling $\mathbf{w}^{(0)}$ appropriately, we may ensure that $\mathbf{z}^{(0)}, \dots, \mathbf{z}^{(-p+1)} \in \mathcal{D}_D$, and that $\|\mathbf{w}^{(0)}\| \geq D$. It follows that

$$\|(I_p \otimes \mathbf{A})\mathbf{w}^{(T)}\|^2 = \|(I_p \otimes \mathbf{A})\mathbf{C}(\mathbf{A})^T \mathbf{w}^{(0)}\|^2 \geq c_{\mathcal{A}} \nu^2 D^2 \geq \frac{c_{\mathcal{A}} \ell^2 D^2}{T}.$$

Thus, for some $T' \in \{T, T-1, \dots, T-p+1\}$, we have that $\|F(\mathbf{z}^{(T')})\| = \|\mathbf{A}\mathbf{z}^{(T')}\| \geq \sqrt{\frac{c_{\mathcal{A}} \ell^2 D^2}{pT'}}$, which establishes the desired lower bound on iteration complexity. □

## C.2 Proof of Propositions 8 and 9

In this section we prove Propositions 8 and 9.

**Proposition 8** (restated). *Suppose $q(z)$ is a degree-$p$ monic real polynomial such that $q(1) = 0$, $r(z)$ is a polynomial of degree $p-1$, and $\ell > 0$. Then there is a constant $C_0 > 0$, depending only on $q(z), r(z)$ and $\ell$, and some $\mu_0 \in (0, \ell)$, so that for any $\mu \in (0, \mu_0)$,*

$$\sup_{\nu \in [\mu, \ell]} \rho(q(z) - \nu \cdot r(z)) \geq 1 - C_0 \cdot \frac{\mu}{\ell}.$$

*Proof.* Let $\Delta \subset \mathbb{C}$ be the unit disk in the complex plane centered at 0 and of radius 1. Set $R(z)$ to be the rational function $R(z) := \frac{q(z)}{r(z)}$. Our goal is to find some $\mu_0$ so that for any $\mu < \mu_0$, we have

$$[\mu, \ell] \cap \{R(z) : |z| \geq 1 - C_0 \cdot \mu/\ell\} \neq \emptyset. \tag{84}$$

We may assume $r(1) \neq 0$ (if instead $r(1) = 0$, then $q(1) - \nu \cdot r(1) = 0$ for all $\nu$, and the proof is complete). Hence $R(1) = 0$, and $R$ is nonconstant. Since $R(z)$ is holomorphic in a neighborhood of 1, there are neighborhoods $U \ni 1$ and $V \ni 0$, with $R(U) = V$, together with conformal mappings $a : \Delta \to U$ with $a(0) = 1$, and $b : V \to \Delta$ with $b(0) = 0$, which extend to continuous functions on

$\bar{\Delta}, \bar{V}$, respectively, so that the mapping $\tilde{R} : \Delta \to \Delta$, defined by $\tilde{R} = b \circ R \circ a$, satisfies $\tilde{R}(w) = w^k$ for some $k \geq 1$.

By Cauchy's integral formula, there is a positive constant $A_0$, depending only on the function $R(\cdot)$, so that for $w \in \Delta$, we have that

$$|a(w) - (1 + a'(0) \cdot w)| \leq A_0 \cdot |w|^2$$

and for $z \in V$, we have that

$$|b(z) - b'(0) \cdot z| \leq A_0 \cdot |z|^2.$$

By choosing $\mu_0 > 0$ to be sufficiently small, we may ensure that $[0, \mu_0] \subset V$. Now fix any $\mu \in (0, \mu_0)$. We consider several cases:

**Case 1.** $k = 1$. Let $w_0 = b(\mu)$, so that

$$|w_0| \leq |b'(0)| \cdot \mu + A_0 \cdot \mu^2 \leq A_1 \cdot \mu \tag{85}$$

for some constant $A_1 > 0$. We have that $R(a(w_0)) = \mu$ by definition of $a(z)$. Moreover,

$$|a(w_0)| \geq |1 + a'(0) \cdot w_0| - A_0 \cdot |w_0|^2 \geq 1 - |a'(0)| \cdot (|b'(0)| \cdot \mu + A_0 \cdot \mu^2) - A_0 \cdot A_1^2 \mu^2,$$

and thus as long as $C_0$ is chosen sufficiently large as a function of $|a'(0)|, |b'(0)|, A_0, A_1, \ell$, we have $|a(w_0)| \geq 1 - C_0 \cdot \mu/\ell$, and $R(a(w_0)) = \mu$, and thus (84) is satisfied in this case.

**Case 2.** $k = 2$. Again let $w_0 = b(\mu)$, so that (85) holds. Let $u_0 \in \Delta$ be a square root of $w_0$, i.e., $u_0^2 = (-u_0)^2 = w_0$. Then $R(a(u_0)) = R(a(-u_0)) = \mu$. It must be the case that either $a'(0) \cdot u_0$ or $-a'(0) \cdot u_0$ has a non-negative real part; suppose without loss of generality that it is $a'(0) \cdot u_0$ (if not, then replace $u_0$ with $-u_0$). Then

$$|a(u_0)| \geq |1 + a'(0) \cdot u_0| - A_0 \cdot |u_0|^2 \geq \sqrt{1 + |a'(0) \cdot u_0|^2} - A_0 \cdot |w_0| \geq \sqrt{1} - A_0 A_1 \mu,$$

and thus as long as $C_0$ is chosen sufficiently large as a function of $A_0, A_1, \ell$, we have that $|a(u_0)| \geq 1 - C_0 \cdot \mu/\ell$ and $R(a(u_0)) = \mu$, and again (84) is satisfied in this case.

**Case 3.** $k \geq 3$. In this case we have that $|R(1 - z)| \leq O(|z|^3)$ as $z \to 0$, so there are some constants $\mu_0, C > 0$ so that for $\mu \in (0, \mu_0)$ we have that any root $z$ of $z \mapsto q(z) - \mu \cdot r(z)$ must satisfy $|z - 1| \geq C \sqrt[3]{\mu/\ell}$. Theorem 25 (in the following section) implies that $\sup_{\nu \in [\mu, \ell]} \rho(q(z) - \nu \cdot r(z)) \geq 1 - 3\sqrt{\mu/\ell}$ for all $\mu \in [0, \ell]$. By making $\mu_0$ smaller if necessary we may assume without loss that for any $\mu \in [0, \mu_0]$, it holds that $3\sqrt{\mu/\ell} < C \sqrt[3]{\mu/\ell}$. If it holds that $\sup_{\nu \in [\mu_0, \ell]} \rho(q(z) - \nu \cdot r(z)) \geq 1$, then the lemma is established for this case. Otherwise, there is some $\mu' \in (0, \mu_0)$ so that for some $\nu \in [\mu', \mu_0]$ we have $\rho(q(z) - \nu \cdot r(z)) \geq 1 - 3\sqrt{\mu'/\ell}$. But since $\mu' \leq \nu \leq \mu_0$ we also have

$$|\rho(q(z) - \nu \cdot r(z)) - 1| \geq C\sqrt[3]{\nu/\ell} > 3\sqrt{\nu/\ell} \geq 3\sqrt{\mu'/\ell},$$

and so it must be the case that $\rho(q(z) - \nu \cdot r(z)) \geq 1 + 3\sqrt{\mu'/\ell} \geq 1$, which establishes the lemma in this case.

We remark also that the case $k \geq 3$ can be dealt with directly, without appealing to Theorem 25: again let $w_0 = b(\mu)$, so that (85) holds. Then there exists some $k$th root $u_0 \in \Delta$ of $w_0$ so that $a'(0) \cdot u_0 = re^{i\theta}$ for some $\theta \in [-\pi/3, \pi/3]$ and $r > 0$. Then

$$|a(u_0)| \geq |1 + a'(0) \cdot u_0| - A_0 \cdot |u_0|^2 \geq \frac{1}{\sqrt{3}}|a'(0)| \cdot |u_0| + 1 - A_0 \cdot |u_0|^2 \geq 1$$

for sufficiently small $u_0$ (which can be made arbitrarily small by taking $\mu \downarrow 0$). $\qquad\square$

**Proposition 9** (restated). *For any constant $C_0 > 0$ and $\mu_0 \in (0, \ell)$, there is some $\mu \in (0, \mu_0)$ and polynomials $q(z), r(z)$ so that $\sup_{\nu \in [\mu, \ell]} \rho(q(z) - \nu \cdot r(z)) < 1 - C_0 \cdot \mu$. Moreover, the choice of the polynomials is given by*

$$q(z) = \ell(z - \alpha)(z - 1), \qquad r(z) = -(1 + \alpha)z + \alpha \qquad for \qquad \alpha := \frac{\sqrt{\ell} - \sqrt{\mu}}{\sqrt{\ell} + \sqrt{\mu}}. \tag{86}$$

*Proof of Proposition 9.* The proof of this proposition involves similar calculations as were done in [ASSS15, Section 5.2], but we spell them out in detail for completeness.

Fix $C_0 > 0, \mu_0 \in (0, \ell)$. We will show that for some $\mu \in (0, \mu_0)$, we have that $\rho(q(z) - \nu \cdot r(z)) < 1 - C_0 \cdot \mu$ for all $\nu \in [\mu, \ell]$, for the choice of $q(z), r(z), \alpha$ in (86).

Fix any $\nu \in [\mu, \ell]$. Solving $q(z) - \nu \cdot r(z) = 0$ gives

$$z = \frac{(\alpha + 1)(1 - \nu/\ell) \pm \sqrt{(\alpha + 1)^2 (1 - \nu/\ell)^2 - 4\alpha}}{2}. \tag{87}$$

Let us write $\alpha = \frac{\sqrt{\ell} - \sqrt{\mu}}{\sqrt{\ell} + \sqrt{\mu}} = 1 - 2\epsilon$ for some $\epsilon \in [\sqrt{\mu/\ell}, 2\sqrt{\mu/\ell}]$. Note that, since $\nu \geq \mu$,

$$(\alpha + 1)^2 (1 - \nu/\ell)^2 - 4\alpha \leq (\alpha + 1)^2 (1 - \mu/\ell)^2 - 4\alpha = 4((1 - \sqrt{\mu/\ell})^2 - \alpha) < 0,$$

so the values of $z$ in (87) have absolute value equal to $\sqrt{\alpha} \leq 1 - \epsilon \leq 1 - \sqrt{\mu/\ell}$ for any $\nu \in [\mu, \ell]$. For sufficiently small $\mu$, we have $\sqrt{\mu/\ell} > C_0 \mu$, and thus $1 - \sqrt{\mu/\ell} < 1 - C_0 \mu$. $\qquad \square$

The polynomials in (86) are closely related to Nesterov's accelerated gradient descent (AGD); we discuss this connection further in Remark 8.

## C.3 Proof of a conjecture of [ASSS15]

In this section we prove the following conjecture:

**Conjecture 24** ([ASSS15])**.** *Suppose $q(z)$ is a degree-$p$ monic real polynomial such that $q(1) = 0$. Then for any polynomial $r(z)$ of degree $p - 1$ and for any $0 < \mu < \ell$, there exists $\nu \in [\mu, \ell]$ so that*

$$\rho(q(z) - \nu \cdot r(z)) \geq \frac{\sqrt{\ell/\mu} - 1}{\sqrt{\ell/\mu} + 1}. \tag{88}$$

**Theorem 25.** *Conjecture 24 is true.*

We are not aware of any reference in the literature directly claiming to prove the statement of Conjecture 24. However, we will show two distinct proofs of Conjecture 24: the first is an indirect proof showing how Conjecture 24 may be derived indirectly as a consequence of prior works ([Nev93, AS16]), and the second is a direct proof using basic principles from complex analysis.

Before continuing, we introduce some further notation.

**Notation.** For a polynomial $s(z)$, write $\rho(s)$ to be the spectral radius of $s$, i.e., $\rho(s) = \max\{|z| : s(z) = 0\}$ is the maximum magnitude of a root of $s$. Let $\hat{\mathbb{C}} = \mathbb{C} \cup \{\infty\}$ denote the Riemann sphere. For $z \in \mathbb{C}, r > 0$, let $D(z, r) := \{w \in \mathbb{C} : |w - z| < r\}$ denote the (open) disk of radius $r$ centered at $z$. Set $\Delta = D(0, 1)$ and $\mathbb{H} := \{z \in \mathbb{C} : \Im(z) > 0\}$ to be the upper half-plane (here $\Im(z)$ denotes the imaginary part of $z$). We refer the reader to [Ahl79] for further background on complex analysis.

*Indirect proof of Theorem 25 using prior works.* We first make the simplifying assumption that there is no $\nu \in [\mu, \ell]$ so that $q(z) - \nu \cdot r(z) = z^p$. (We remove this assumption at the end of the proof.) Let us write $q(z) = z^p - q_{p-1} z^{p-1} - \cdots - q_1 z - q_0, r(z) = r_0 + r_1 z + \cdots + r_{p-1} z^{p-1}$. We have that $q_0 + \cdots + q_{p-1} = 1$ since $q(1) = 0$. Similar to the proof of Theorem 7, define, for $\nu \in [\mu, \ell]$,

$$\mathbf{C}(\nu) := \begin{pmatrix} 0 & 1 & 0 & \cdots & 0 \\ 0 & 0 & 1 & 0 & \cdots \\ \vdots & \vdots & \ddots & \ddots & \vdots \\ \vdots & \vdots & \ddots & 0 & 1 \\ C_0(\nu) & C_1(\nu) & \cdots & C_{p-2}(\nu) & C_{p-1}(\nu) \end{pmatrix},$$

where $C_j(\nu) = q_j + r_j \nu$ for $0 \leq j \leq p - 1$. By our initial simplifying assumption, there is no $\nu \in [\mu, \ell]$ so that $C_0(\nu) = \cdots = C_{p-1}(\nu) = 0$. Then by [ASSS15, Lemma 14], we have that

$$\rho(\mathbf{C}(\nu)) = \rho\left(z^p - \sum_{j=0}^{p-1} C_j(\nu) z^j\right) = \rho(q(z) - \nu \cdot r(z)). \tag{89}$$

Let $\mathbf{e} := \frac{1}{\sqrt{p}}(1, 1, \ldots, 1)^\top \in \mathbb{R}^p$. Note that $\mathbf{e}^\top \mathbf{C}(\nu)^t \mathbf{e}$ is a polynomial in $\nu$, which we write as $p_t(\nu)$, of degree at most $t$. It is also immediate that $p_t(0) = 1$ for all $t$. Moreover, $p_t$ satisfies

$$|p_t(\nu)| \leq |\mathbf{e}^\top \mathbf{C}(\nu)^t \mathbf{e}| \leq \|\mathbf{c}(\nu)^t \mathbf{e}\| \leq \|\mathbf{C}(\nu)^t\|_\sigma. \tag{90}$$

Next we will need the following lemma:

**Lemma 26.** *It holds that*

$$\sup_{\nu \in [\mu, \ell]} \rho(\mathbf{C}(\nu)) = \sup_{\nu \in [\mu, \ell]} \liminf_{t \to \infty} \|\mathbf{C}(\nu)^t\|_\sigma^{1/t} \geq \liminf_{t \to \infty} \sup_{\nu \in [\mu, \ell]} \|\mathbf{C}(\nu)^t\|_\sigma^{1/t}. \tag{91}$$

Notice that the opposite direction of the inequality in (91) holds trivially, and thus we have equality. Notice also that the first equality in (91) follows by Gelfand's formula.

*Proof of Lemma 26.* Note that if at least one of $C_0(\nu), \ldots, C_{p-1}(\nu)$ is nonzero, then $\mathbf{C}(\nu)^p \neq \mathbf{0}$: this is the case since there is some vector $\mathbf{v} \in \mathbb{R}^p$ so that $\langle \mathbf{v}, (C_0(\nu), \ldots, C_{p-1}(\nu)) \rangle \neq 0$, and the first entry of $\mathbf{C}(\nu)^p \mathbf{v}$ is $\langle \mathbf{v}, (C_0(\nu), \ldots, C_{p-1}(\nu)) \rangle$. Since $[\mu, \ell]$ is compact, it follows that the function $\nu \mapsto \frac{\|\mathbf{C}(\nu)\|^p}{\|\mathbf{C}(\nu)^p\|}$ is bounded for $\nu \in [\mu, \ell]$. Let $S := \sup_{\nu \in [\mu, \ell]} \frac{\|\mathbf{C}(\nu)\|^p}{\|\mathbf{C}(\nu)^p\|}$, $\sigma := \max\left\{1/2, \frac{\ln(p-1)}{\ln(p)}\right\}$, and $A_p = 2^p$. Then [Koz09, Theorem 1] gives that for all $\nu \in [\mu, \ell]$ and $t \geq 1$, we have

$$\|\mathbf{C}(\nu)^t\|^{1/t} \leq \rho(\mathbf{C}(\nu)) \cdot A_p^{A_p \cdot t^{\sigma-1}} \cdot \left(\frac{\|\mathbf{C}(\nu)\|^p}{\|\mathbf{C}(\nu)^p\|}\right)^{A_p \cdot t^{\sigma-1}}$$

$$\leq \rho(\mathbf{C}(\nu)) \cdot A_p^{A_p \cdot t^{\sigma-1}} \cdot S^{A_p \cdot t^{\sigma-1}}.$$

Since $\sigma < 1$, it follows that

$$\liminf_{t \to \infty} \sup_{\nu \in [\mu, \ell]} \|\mathbf{C}(\nu)^t\|^{1/t} \leq \liminf_{t \to \infty} \sup_{\nu \in [\mu, \ell]} \rho(\mathbf{C}(\nu)) \cdot A_p^{A_p \cdot t^{\sigma-1}} \cdot S^{A_p \cdot t^{\sigma-1}} = \sup_{\nu \in [\mu, \ell]} \rho(\mathbf{C}(\nu)).$$

$\square$

By (90) and Lemma 26, we have

$$\liminf_{t \to \infty} \sup_{\nu \in [\mu, \ell]} |p_t(\nu)|^{1/t} = \liminf_{t \to \infty} \sup_{\nu \in [\mu, \ell]} |\mathbf{e}^\top \mathbf{C}(\nu)^t \mathbf{e}|^{1/t}$$

$$\leq \liminf_{t \to \infty} \sup_{\nu \in [\mu, \ell]} \|\mathbf{C}(\nu)^t \mathbf{e}\|^{1/t}$$

$$\leq \liminf_{t \to \infty} \sup_{\nu \in [\mu, \ell]} \|\mathbf{C}(\nu)^t\|_\sigma^{1/t}$$

$$\leq \sup_{\nu \in [\mu, \ell]} \rho(\mathbf{C}(\nu)). \tag{92}$$

(We use Lemma 26 in (92).) Let $\mathcal{S}_t$ denote the set of polynomials $s_t$ with complex coefficients of degree at most $t$ such that $s_t(0) = 1$. (Note in particular that the polynomials $p_t$ defined above belong to $\mathcal{S}_t$ for each $t$.) It follows from Theorem 3.6.3, and Example 3.8.3 of [Nev93] that

$$\inf_{t > 0} \inf_{s_t \in \mathcal{S}_t} \sup_{\nu \in [\mu, \ell]} |s_t(\nu)|^{1/t} = \frac{\sqrt{\ell/\mu} - 1}{\sqrt{\ell/\mu} + 1}. \tag{93}$$

(In more detail, the quantity on the left-hand-side of (93), which is called the *optimal reduction factor* of the region $[\mu, \ell]$ in [Nev93] and denoted by $\eta_{[\mu, \ell]}$ therein, is shown in [Nev93, Theorem 3.6.3] to be equal to $e^{-G(0)}$, where $G : \mathbb{C} - [\mu, \ell] \to \mathbb{R}$ is the Green's function for the region $\mathbb{C} - [\mu, \ell]$. Then [Nev93, Example 3.8.3] explicitly computes the Green's function and shows that $e^{-G(0)}$ is the quantity on the right-hand-side of (93)).

Combining (89), (92), and (93), we see that

$$\sup_{\nu \in [\mu, \ell]} \rho(q(z) - \nu \cdot r(z)) = \sup_{\nu \in [\mu, \ell]} \rho(\mathbf{C}(\nu)) \geq \inf_{t > 0} \inf_{s_t \in \mathcal{S}_t} \sup_{\nu \in [\mu, \ell]} |s_t(\nu)|^{1/t} = \frac{\sqrt{\ell/\mu} - 1}{\sqrt{\ell/\mu} + 1}.$$

Finally, we deal with the case that for some $\nu \in [\mu, \ell]$, we have $q(z) - \nu \cdot r(z) = z^p$. Since the roots of a polynomial are continuous functions of its coefficients and a continuous function defined on a compact set is uniformly continuous, for any $\epsilon > 0$, there is some $\delta > 0$ so that for any polynomial $\tilde{r}(z) = \tilde{r}_0 + \cdots + \tilde{r}_p z^{p-1}$ with $|\tilde{r}_j - r_j| \leq \delta$ for each $j$, we have that $|\rho(q(z) - \nu \cdot r(z)) - \rho(q(z) - \nu \cdot \tilde{r}(z))| \leq \epsilon$ for all $\nu \in [\mu, \ell]$. Such a polynomial $\tilde{r}$ may be found so that $q(z) - \nu \cdot \tilde{r}(z) \neq z^p$ for all $\nu \in [\mu, \ell]$, and so by the proof above we have

$$\sup_{\nu \in [\mu, \ell]} \rho(q(z) - \nu \cdot r(z)) \geq \sup_{\nu \in [\mu, \ell]} \rho(q(z) - \nu \cdot \tilde{r}(z)) - \epsilon \geq \frac{\sqrt{\ell/\mu} - 1}{\sqrt{\ell/\mu} + 1} - \epsilon.$$

The desired conclusion follows by taking $\epsilon \downarrow 0$, thus completing the proof of Theorem 25.

We remark that an alternative approach to establishing (93) without appealing to the heavy machinery of Green's functions is to use [AS16, Lemma 2] directly, which shows that

$$\inf_{s_t \in \mathcal{S}_t} \sup_{\nu \in [\mu, \ell]} |s_t(\nu)| \geq \left( \frac{\sqrt{\ell/\nu} - 1}{\sqrt{\ell/\nu} + 1} \right)^t.$$

$\square$

The approach to proving Conjecture 24 described above is unsatisfying in that it first passes a statement about polynomials (namely, Conjecture 24) to a statement about matrices (namely, about $\liminf_t \sup_{\nu \in [\mu, \ell]} \|\mathbf{C}(\nu)^t\|_\sigma^{1/t}$), relying on a nontrivial uniform version of Gelfand's formula ([Koz09]), before passing back to a statement about polynomials and using either [AS16] or [Nev93] to establish (93). It is natural to wonder whether there is a *direct* proof of Conjecture 24 which operates on the polynomials $q(z), r(z)$ directly, without bounding the optimal reduction factor in (93) and constructing the matrices $\mathbf{C}(\nu)$. We next give such a direct proof of Conjecture 24, which follows from basic facts from complex analysis.

*Direct proof of Theorem 25.* Fix some polynomials $q, r$ satisfying the conditions of Conjecture 24. Choose $\delta \in \mathbb{R}$ so that $\max_{\nu \in [\mu, \ell]} \rho(q(z) - \nu \cdot r(z)) = 1 - \delta$. Notice that the maximum exists since the roots of a polynomial are a continuous function of its coefficients. Our goal is to show that $\delta \leq 1 - \frac{\sqrt{\ell/\mu} - 1}{\sqrt{\ell/\mu} + 1}$. Define the rational function $R : \hat{\mathbb{C}} \to \hat{\mathbb{C}}$ by $R(z) = \frac{q(z)}{r(z)}$. If, for some $z_0$ with $|z_0| > 1 - \delta$, $R(z_0) =: \nu \in [\mu, \ell]$, then we have $q(z_0) - \nu \cdot r(z_0) = 0$, and so $\rho(q(z) - \nu \cdot r(z)) \geq |z_0| > 1 - \delta$, a contradiction. Hence the restriction of $R$ to $\hat{\mathbb{C}} - \overline{D(0, 1 - \delta)}$ is in fact a holomorphic function to the Riemann surface $\hat{\mathbb{C}} - [\mu, \ell]$, i.e., $R : \hat{\mathbb{C}} - \overline{D(0, 1 - \delta)} \to \hat{\mathbb{C}} - [\mu, \ell]$. (Recall that $\overline{D(0, 1 - \delta)}$ denotes the closed disc of radius $1 - \delta$ centered at 0.) We next need the following standard lemma:

**Lemma 27.** *There is a holomorphic map* $G : \hat{\mathbb{C}} - [\mu, \ell] \to \Delta$ *from* $\hat{\mathbb{C}} - [\mu, \ell]$ *to the unit disk* $\Delta$, *so that* $G(0) = \frac{1 - \sqrt{\ell/\mu}}{1 + \sqrt{\ell/\mu}}$ *and* $G(\infty) = 0$.[15]

For completeness we prove Lemma 27 below; we first complete the proof of Theorem 25 assuming Lemma 27.

Notice that the mapping $z \mapsto \frac{1}{z}$ maps $D\left(0, \frac{1}{1-\delta}\right)$ to $\hat{\mathbb{C}} - \overline{D(0, 1 - \delta)}$. Thus we may define $\tilde{R} : D\left(0, \frac{1}{1-\delta}\right) \to \hat{\mathbb{C}} - [\mu, \ell]$ by $\tilde{R}(z) = R\left(\frac{1}{z}\right)$, which is holomorphic since $R$ is. Now define the function $H : \Delta \to \Delta$ by

$$H(z) = G\left( \tilde{R}\left( \frac{1}{1-\delta} \cdot z \right) \right),$$

which is well-defined since $\frac{1}{1-\delta} \cdot z \in D\left(0, \frac{1}{1-\delta}\right)$ for $z \in \Delta$. Since $H$ is a composition of the holomorphic functions $z \mapsto \frac{1}{1-\delta} \cdot z$, $\tilde{R}$, and $G$, $H$ is itself holomorphic. Note that

$$H(0) = G(\tilde{R}(0)) = G(R(\infty)) = G(\infty) = 0 \tag{94}$$

$$H(1-\delta) = G(\tilde{R}(1)) = G(R(1)) = G(0) = \frac{1 - \sqrt{\ell/\mu}}{1 + \sqrt{\ell/\mu}}. \tag{95}$$

where to derive (94) we used that $R(\infty) = \infty$ since $q(z)$ is monic of degree $p$ and $r(z)$ is of degree $p-1$, and to derive (95) we used that $R(1) = 0$ since $q(1) = 0$ by assumption.

Next we recall the Schwarz lemma from elementary complex analysis:

**Lemma 28** (Schwarz). *A holomorphic function $f : \Delta \to \Delta$ with $f(0) = 0$ satisfies $|f(z)| \leq |z|$ for all $z \in \Delta$.*

Since $H : \Delta \to \Delta$ is holomorphic, satisfies $H(0) = 0$ (by (94)), (95) together with Lemma 28 gives us that

$$|H(1-\delta)| = \frac{\sqrt{\ell/\mu} - 1}{\sqrt{\ell/\mu} + 1} \leq 1 - \delta.$$

In particular, $\delta \leq 1 - \frac{\sqrt{\ell/\mu}-1}{\sqrt{\ell/\mu}+1}$, which completes the proof.

$\square$

Now we prove Lemma 27 for completeness.

*Proof of Lemma 27.* We will take

$$G(w) := \frac{\sqrt{w-\mu} - i\sqrt{\ell-w}}{\sqrt{w-\mu} + i\sqrt{\ell-w}},$$

where the choice of the branch of the square root will be explained below. In particular, $G$ is obtained as the composition of maps $G = G_5 \circ G_4 \circ G_3 \circ G_2 \circ G_1$, where $G_1, \ldots, G_5$ are defined by:

$$
\begin{aligned}
G_1 &: \hat{\mathbb{C}} - [\mu, \ell] \to \hat{\mathbb{C}} - [0,1], & w &\mapsto \frac{\ell - w}{\ell - \mu} \\
G_2 &: \hat{\mathbb{C}} - [0,1] \to \hat{\mathbb{C}} - [1,\infty], & w &\mapsto 1/w \\
G_3 &: \hat{\mathbb{C}} - [1,\infty] \to \hat{\mathbb{C}} - [0,\infty], & w &\mapsto w - 1 \\
G_4 &: \hat{\mathbb{C}} - [0,\infty] \to \mathbb{H}, & w &\mapsto \sqrt{w} \\
G_5 &: \mathbb{H} \to \Delta, & w &\mapsto \frac{w - i}{w + i},
\end{aligned}
\tag{96}
$$

where the choice of the branch of the square root in (96) is given by $G_4(re^{i\theta}) = \sqrt{r}e^{i\theta/2}$ for $r > 0, \theta \in (0, 2\pi)$. It is clear that each of $G_1, \ldots, G_5$ are holomorphic functions between their respective Riemann surfaces, and thus $G : \hat{\mathbb{C}} - [\mu, \ell] \to \Delta$ is holomorphic.

To verify the values of $G(0), G(\infty)$, note that $G_3(G_2(G_1(0))) = -\mu/\ell$ and $G_3(G_2(G_1(\infty)) = -1$. By the choice of the branch of the square root defining $G_4$, we have that $G_4(G_3(G_2(G_1(0)))) = i\sqrt{\mu/\ell}$ and $G_4(G_3(G_2(G_1(\infty))) = i$. It follows that $G(\infty) = 0$ and $G(0) = \frac{1-\sqrt{\ell/\mu}}{\sqrt{\ell/\mu}+1}$. $\square$

Theorem 25 leads to an algorithm-independent version of Theorem 7. We need the following definition: a $p$-SCLI in the form (78) with $\mathbf{C}_j(\mathbf{A}) = \alpha_j \mathbf{A} + \beta_j I_n$ is called *consistent* ([ASSS15]) if $\sum_{j=0}^{p-1} \beta_j = 1$. It is known that if the iterates of $\mathcal{A}$ converge for all $\mathbf{b} \in \mathbb{R}^n$, then $\mathcal{A}$ is consistent; hence consistent $p$-SCLIs represent all "useful" ones.

**Proposition 29.** *Let $\mathcal{A}$ be a consistent $p$-SCLI and let $\mathbf{z}^{(t)}$ denote the $t$th iterate of $\mathcal{A}$. Then for all $T \in \mathbb{N}$, there is some $F \in \mathcal{F}_{n,\ell,D}^{\mathrm{bil}}$ so that for some initialization $\mathbf{z}^{(0)}, \ldots, \mathbf{z}^{(-p+1)} \in \mathcal{D}_D$ and some $T' \in \{T, T-1, \ldots, T-p+1\}$, it holds that $\mathrm{TGap}_F^{\mathcal{D}_{2D}}(\mathbf{z}^{(T')}) \geq \frac{\ell D^2}{\sqrt{20pT}}$.*

*Proof.* The proof of Proposition 29 mirrors nearly exactly the proof of Theorem 7, except we need only consider Case 3b by consistency. Moreover, the only difference to Case 3b is the following: instead of applying Proposition 8, we apply Theorem 25 (i.e., Conjecture 24) with $\mu = \ell/2T$. Then, we may choose $\mu \in [\ell/2T, \ell]$ accordingly per the statement of Conjecture 24 to conclude that

$$\rho(\mathbf{C}(\mathbf{A}))^T \geq \left(\frac{2T-1}{2T+1}\right)^T \geq 1/5.$$

Thus it follows in the same way as in the proof of Theorem 7 that for some $T' \in \{T, T-1, \ldots, T-p+1\}$ we have that $\|F(\mathbf{z}^{(T')})\| \geq \sqrt{\frac{\nu^2 D^2}{5pT}} \geq \sqrt{\frac{\ell^2 D^2}{20pT}}$. $\qquad\square$

The conclusion of Proposition 29 is known even for non-stationary $p$-CLIs and without the superfluous $1/\sqrt{p}$ factor (e.g., it follows from Proposition 5 in [ASM$^+$20]), but our proof is new since it involves Theorem 25, which does not seem to have been previously known in the literature. We are hopeful that Theorem 25 may have further consequences for proving lower bounds for optimization algorithms, such as in the stochastic setting.

### C.4 Byproduct: Lower bound for convex function minimization

In this section we prove an (algorithm-dependent) lower bound of $\Omega(1/T)$ on the rate of convergence for $p$-SCLIs for convex function minimization. This statement was claimed to be proven by [AS16, Corollary 1], but in fact their results only give a linear lower bound for the strongly convex case (and not the sublinear bound of $\Omega(1/T)$ we obtain here): in particular, Corollary 1 of [AS16] is a corollary of Theorem 2 of [AS16], which should be adjusted to state that the error after $T$ iterations cannot be upper bounded by $O\left((1 - (\mu/L)^\alpha)^T\right)$, for any $\alpha < 1$.[16] This weaker version of [AS16, Theorem 2] does not imply [AS16, Corollary 1].

In this section, we show that Proposition 8 can be used to correct the above issue in [AS16]. We first introduce the function class of "hard" functions, analogously to $\mathcal{F}_{n,\ell,D}^{\mathrm{bil}}$. Let $\mathcal{F}_{n,\ell,D}^{\mathrm{quad}}$ be the class of $\ell$-smooth[17] functions $f : \mathbb{R}^n \to \mathbb{R}$ of the form

$$f(\mathbf{x}) = \frac{1}{2}\mathbf{x}^\top \mathbf{S}\mathbf{x} + \mathbf{b}^\top \mathbf{x},$$

for which $\mathbf{S} \in \mathbb{R}^{n \times n}$ is a positive definite matrix and $\mathbf{x}^* := -\mathbf{S}^{-1}\mathbf{b}$ has norm $\|\mathbf{x}^*\| \leq D$. We prove the following lower bound for $p$-SCLI algorithms using functions from $\mathcal{F}_{n,\ell,D}^{\mathrm{quad}}$

**Proposition 30.** *Let $\mathcal{A}$ be a $p$-SCLI, and let $\mathbf{x}^{(t)}$ denote the $t$th iterate of $\mathcal{A}$. Then there are constants $c_\mathcal{A}, T_\mathcal{A} > 0$ so that the following holds: for all $T \geq T_\mathcal{A}$, there is some $f \in \mathcal{F}_{n,\ell,D}^{\mathrm{quad}}$ so that for some initialization $\mathbf{x}^{(0)}, \ldots, \mathbf{x}^{(-p+1)} \in \mathcal{B}(\mathbf{0}, D)$ and some $T' \in \{T, T+1, \ldots, T+p-1\}$, it holds that $f(\mathbf{x}^{(T)}) - f(\mathbf{x}^*) \geq \frac{c_\mathcal{A} \ell D^2}{T}$.*

*Proof of Proposition 30.* Note that for any $\mathbf{x} \in \mathbb{R}^n$, we have that

$$f(\mathbf{x}) - f(\mathbf{x}^*) = \frac{1}{2}\mathbf{x}^\top \mathbf{S}\mathbf{x} + \mathbf{b}^\top \mathbf{x} + \frac{1}{2}\mathbf{b}^\top \mathbf{S}^{-1}\mathbf{b} = \frac{1}{2}(\mathbf{S}\mathbf{x} + \mathbf{b})^\top \mathbf{S}^{-1}(\mathbf{S}\mathbf{x} + \mathbf{b}).$$

Define, for each $t \geq 0$,

$$\mathbf{w}^{(t)} := \begin{pmatrix} \mathbf{x}^{(t-p+1)} \\ \mathbf{x}^{(t-p+2)} \\ \vdots \\ \mathbf{x}^{(t)} \end{pmatrix}.$$

We will choose $\mathbf{S} = \nu \cdot I_n$, for some $\nu \in (0, \ell]$ to be chosen later. Thus $f(\mathbf{x}) - f(\mathbf{x}^*) = \frac{1}{2\nu} \|\mathbf{S}\mathbf{x} + \mathbf{b}\|^2$. Next we proceed exactly as in the proof of Theorem 7, with $\mathbf{S}$ taking the role of $\mathbf{A}$ there. In particular, we define $\mathbf{C}(\mathbf{S})$ exactly as in (79), where $\mathbf{C}_j(\mathbf{S}) = \alpha_j \cdot \mathbf{A} + \beta_j \cdot I_n$, $\mathbf{N}(\mathbf{S}) = \gamma \cdot \mathbf{S} + \delta \cdot I_n$, where $\alpha_j, \beta_j, \gamma, \delta \in \mathbb{R}$ are the constants associated with the $p$-SCLI $\mathcal{A}$. Cases 1, 2, and 3a of the proof (namely, the ones in which the algorithm does not converge) proceed in exactly the same way and we omit the details.

To deal with Case 3b (i.e., the case that $\sum_{j=0}^{p-1} \beta_j = 1$), we choose $\mathbf{b} = \mathbf{0} \in \mathbb{R}^n$, and (83) gives us that $(I_p \otimes \mathbf{S})\mathbf{w}^{(t)} = \mathbf{C}(\mathbf{S})^t \mathbf{S} \mathbf{w}^{(0)}$. Moreover, it follows from [ASSS15, Lemma 14] that

$$\rho(\mathbf{C}(\mathbf{S})) = \rho(q(z) - \nu \cdot r(z)).$$

By Proposition 8, there are some constants $\mu_{\mathcal{A}}, C_{\mathcal{A}} > 0$ so that for any $\mu \in (0, \mu_{\mathcal{A}})$, there is some $\nu \in [\mu, \ell]$ so that $\rho(q(z) - \nu \cdot r(z)) \geq 1 - C_{\mathcal{A}} \cdot \mu/\ell$. Letting $T_{\mathcal{A}}$ be so that $\ell/(4T_{\mathcal{A}}) < \mu_{\mathcal{A}}$, as long as $T \geq T_{\mathcal{A}}$, we may choose $\mu = \ell/(4T)$, and set $\nu \in [\ell/(4T), \ell]$ accordingly per Proposition 8. By Lemma 23, we have that for $T \geq T_{\mathcal{A}}$,

$$\|\mathbf{C}(\mathbf{S})^T\|_\sigma \geq \rho(\mathbf{C}(\mathbf{S}))^T \geq (1 - C_{\mathcal{A}}/(4T))^T \geq \exp(-C_{\mathcal{A}}).$$

Set $c_{\mathcal{A}} = \exp(-C_{\mathcal{A}})$. Choose $\mathbf{w}^{(0)} = ((\mathbf{x}^{(-p)})^\top, \ldots, (\mathbf{x}^{(0)})^\top)^\top \in \mathbb{R}^{np}$ so that it is a right singular vector of $\mathbf{C}(\mathbf{S})^T$ corresponding to a singular value of magnitude at least $c_{\mathcal{A}}$. By scaling $\mathbf{w}^{(0)}$ appropriately, we may ensure that $\|\mathbf{x}^{(-p+1)}\|, \ldots, \|\mathbf{x}^{(0)}\| \leq D$, and that $\|\mathbf{w}^{(0)}\| \geq D$. It follows that

$$\sum_{j=0}^{p-1} \left( f(\mathbf{x}^{(T-j)}) - f(\mathbf{x}^*) \right) = \frac{1}{2\nu} \|(I_p \otimes \mathbf{S})\mathbf{w}^{(T)}\|^2 = \frac{\nu}{2} \|\mathbf{C}(\mathbf{S})^t \mathbf{w}^{(0)}\|^2 \geq \frac{\nu D^2 c_{\mathcal{A}}}{2} \geq \frac{\ell D^2 c_{\mathcal{A}}}{8T}.$$

By replacing $T$ with $T + p - 1$ and decreasing $c_{\mathcal{A}}$, the conclusion of Proposition 30 follows. $\square$

**Remark 8.** As in Theorem 7, the lower bound in Proposition 30 involves an algorithm-dependent constant $c_{\mathcal{A}}$ due to the reliance on Proposition 8. We remark that the iterates $\mathbf{x}^{(t)}$ of gradient descent satisfy $f(\mathbf{x}^{(t)}) - f(\mathbf{x}^*) \leq O(\ell D^2/T)$ for any $\ell$-smooth convex function $f$, so Proposition 8 is tight up to the algorithm-dependent constant $c_{\mathcal{A}}$. Nesterov's AGD improves the rate of gradient descent to $O(\ell D^2/T^2)$, but is non-stationary (i.e., requires a changing step size). The polynomials in Proposition 9 (i.e., (9)) showing the necessity of an algorithm-dependent constant in Proposition 8 correspond under the reduction outlined in the proof of Proposition 30 to running Nesterov's AGD with a fixed learning rate. We do not know if such an algorithm (for an appropriate choice of the arbitrary but fixed learning rate) can lead to an arbitrarily large constant factor speedup over the rate $O(\ell D^2/T)$ of gradient descent. We believe this is an interesting direction for future work.