[Reviews · NeurIPS 2020]

Review 1

Summary and Contributions: This paper studies unconstrained OGDA in multi-player games. It shows that for monotone and smooth (first-order + second-order) games, the last-iterate convergence rate is 1/sqrt{t}, and shows a matching lower bound for all p-SCLI algorithms, which OGDA belongs to.

Strengths: The results are important for machine learning and multi-player game communities. It shows the last-iterate convergence for OGDA under a rather mild assumption, generalizing many previous works. One advantage over many previous results is that the convergence rate does not depend on some problem-dependent quantity like the smallest eigenvalue of some matrices.

Weaknesses: Although this work provides some breakthrough on the last-iterate convergence property of OGDA, the analysis of the main result is very long and lacks intuition. It will be good if you can provide more high-level explanation about the "potential function", perhaps with its physical meaning, and give more intuition about why it won't grow/grows slowly. Some simple experiments showing the behavior of the potential function, and/or comparison between the behaviors of OGDA and EG will also make the paper much more interesting.

Correctness: I only checked some part for the upper bounds, and did not spot errors.

Clarity: The writing is good. This paper well explains how the technique is related to previous techniques used for analyzing extragradient algorithms, and how to addresses the additional challenges. It also did a good job in first explaining a special case for matrix games then going to the more general case, making the content more accessible.

Relation to Prior Work: Yes (see the "Clarity section").

Reproducibility: Yes

Additional Feedback:


Review 2

Summary and Contributions: In the paper, the authors derive O(1/\sqrt{T}) last-iterate convergence rate for optimistic gradient method in unconstrained smooth monotone games. Moreover, they also show that this upper bound is tight by proving a lower bound result for all p-SLCI algorithms. Accompanying the analysis for the lower bound, several results on the spectral radius of a family of polynomials are proved.

Strengths: The problem of whether we can derive last-iterate non-asymptotic convergence rate for extragradient-like methods in general monotone games is an important problem that needs to be answered to better understand the dynamics of these methods. This paper is largely built upon [GPDO20] but instead investigates the optimistic gradient algorithm and provides a partial answer to the above question. Nonetheless, this adaptation from EG to OG seems to be nontrivial and requires the developments of new techniques. In addition, the lower bound is also more general and therefore I believe the contribution of this paper is enough for publication.

Weaknesses: The major limitation of the analysis, shared by [GPDO20], is the requirement of Lipschitz Jacobian and the impossibility of working with a constrained set. Nonetheless, given that this is a first result of this type, I think it is sensible to work with more restricted assumptions. Otherwise, I believe the authors should elaborate more on the lower bound result, and in particular present it along with the known results for last-iterate convergence in bilinear games to provide a better intuition of the meaning of the lower bound. See the part "Relation to prior work" for more details.

Correctness: I did not go through all the proofs but the results and the methods are reasonable and in agreement with existing work.

Clarity: The paper is generally well-written and some key ingredients of the proofs are provided in the main text.

Relation to Prior Work: While the paper properly refers to most of the works working on this subject, I notice that the paper does not mention the linear last-iterate convergence of OG in zero-sum bilinear games (proved by several papers already cited in this work). I think this is worth mentioning since it is somehow in contrast to the lower bound result proved by the paper. In fact, in Theorem 7 it is shown that for a given T we can always find a bilinear (affine) problem such that at time T the last iterate of the algorithm performs poorly (and so that the bound cannot be improved with further assumption/characterization). On the other hand, when we fix an instance of this family, the last iterate eventually becomes a better candidate that the averaged iterate. These results are complementary and I think they should be presented side by side to give the readers a more complete picture on the behavior of the method.

Reproducibility: Yes

Additional Feedback:


Review 3

Summary and Contributions: Very interesting work! The authors showed 1/sqrt{T} last-iterate convergence for monotone games using extra-gradient, where convergence is not guaranteed at all if normal gradient-descent or mirror-descent (the most widely used no-regret algorithm in this literature) is used. The proof also used an interesting adaptive potential function technique, which appears novel to me. This work injects a fresh light into the multi-agent online learning literature, where finite-time last-iterate convergence is extremely scarce: only very recently has there been such results on strongly monotone games (very strong structural requirement) and cocoercive games (pretty restrictive too). Besides these two works, at the generality of monotone games, existing works at best provide convergence guarantees in the ergodic sense (and not very illuminating from a game perspective).

Strengths: Strong and interesting result, interesting technique, lower bound completes the picture. Great work!

Weaknesses: Missing the related reference "Mirror Descent Learning in Continuous Games/CDC 2017" It has no finite-time rate, but has last-iterate convergence for a broad class of games.

Correctness: Yes Post-rebuttal updates: looks good!

Clarity: Yes, the paper is well written. The ideas are clearly presented in the main part of the paper. I enjoyed reading the paper.

Relation to Prior Work: Yes, the prior work has been clearly discussed and the originality of the current work is clearly highlighted.

Reproducibility: Yes

Additional Feedback: This is certainly one of the results that lies at the frontier of the field of multi-agent no-regret learning. Last-iterate convergences in general are hard to obtain, even for qualitative convergence. A majority of prior work at the intersection of economics and ML in this area has focused on ergodic convergence (i.e. the convergence of the time-average of the joint action of all players). In the past few years, a growing literature has devoted attention to last-iterate convergence. Even at this point, such qualitative last-iterate convergence results still have a big room to grow. In this background, quantitative (rather than qualitative) last-iterate convergence rates are even more under-explored (and more valuable of course). I'm certainly pleased to see a result like this.

[Author Response · NeurIPS 2020]

1 We thank the reviewers for their helpful comments. Please find below our responses.

2 **Reviewer 1:** Thank you for your helpful feedback. We will add additional discussion to the paper to give more intuition
3 for the potential function, and we will add some experiments illustrating its behavior. (In footnote 6 we give an example
4 showing that a natural alternative to our potential, namely consecutive "gradient" norms $\|(F_{\mathcal{G}}(\mathbf{z}^{(t)}), F_{\mathcal{G}}(\mathbf{z}^{(t-1)}))\|$, will
5 not suffice as a potential function.)

6 **Reviewer 2:** Thank you for your helpful feedback. We will update the discussion around Theorem 7 to state the
7 complementary fact (and refer to the appropriate references) that for a fixed problem instance the last iterate has linear
8 convergence and so eventually becomes better than the averaged iterate.

9 **Reviewer 3:** Thank you for your helpful feedback. We will add a reference to the CDC 2017 paper.

[Meta-Review · NeurIPS 2020]

This paper examines the "last-iterate" convergence rate of optimistic gradient descent with perfect gradient feedback in smooth, unconstrained monotone games. Without strong monotonicity, the closest results in the literature are the recent papers [LZMJ20] and [GPDO20], where the authors prove an $O(1/\sqrt{T})$ convergence rate for cocoercive and merely monotone games respectively (with perfect gradient input in both cases). The current paper extends the results of [GPDO20] to the optimistic gradient algorithm which, in contrast to extra-gradient, does not require an intermediate "gradient sample". The reviewers all agreed that this contribution is interesting, and the paper itself is well written. As a result, there was a quick consensus to accept the paper. After my own reading of the paper, I concur with the reviewers' assessment; however, I would also like the authors to clarify how their work compares to [GDPO20]. This paper relies heavily on the techniques of [GPDO20] but, in the introduction, [GPDO20] is only mentioned as an afterthought and the motivation to focus on "optimistic gradient" instead of "extra-gradient" is left unqualified - for example, when is extra-gradient regretful? In the main body of the paper, the authors explain quite clearly which techniques are adapted from [GPDO20] and how -- a version of these explanations should also appear in the introduction (the extra page available should be more than sufficient for this). I would also recommend making a clear distinction between methods that require perfect gradient feedback and those that do not, as well as those that examine gradient versus extra-gradient methods. The paper currently mixes these contributions together; tabulating them would make the paper's contribution clearer and provide valuable context. Finally, a minor point: several references in the bibliography seem to be incomplete – a quick run through DBLP should be enough to fix this.